# WaterDrum: Watermark-based Data-centric Unlearning Metric

**Xinyang Lu\***[1], **Xinyuan Niu\***[1,2]**, Gregory Kang Ruey Lau\***[1,3]**,**
**Nhung Bui**[1]**, Rachael Hwee Ling Sim**[1]**, John Russell Himawan**[1]**, Fanyu Wen**[1]**,**
**Chuan-Sheng Foo**[2]**, See-Kiong Ng**[1]**, Bryan Kian Hsiang Low**[1]

[1]Department of Computer Science, National University of Singapore, Singapore 117417
[2]Centre for Frontier AI Research, A*STAR, Singapore 138632
[3]CNRS@CREATE, 1 Create Way, #08-01 Create Tower, Singapore 138602
`{xinyang.lu,xinyuan,gregorylau,btcnhung,rachael.sim}@u.nus.edu,`
`{johnr.himawan,wenfanyu}@u.nus.edu, foo_chuan_sheng@a-star.edu.sg`
`seekiong@nus.edu.sg, lowkh@comp.nus.edu.sg`

## Abstract

Large language model (LLM) unlearning is critical in real-world applications where it is necessary to efficiently remove the influence of private, copyrighted, or harmful data from some users. Existing utility-centric unlearning metrics (based on model utility) may fail to accurately evaluate the *extent of unlearning* in realistic settings such as when the forget and retain sets have semantically similar content and/or retraining the model from scratch on the retain set is impractical. This paper presents the first data-centric unlearning metric for LLMs called `WaterDrum` that exploits robust text watermarking to overcome these limitations. We introduce new benchmark datasets (with different levels of data similarity) for LLM unlearning that can be used to rigorously evaluate unlearning algorithms via `WaterDrum`. Our code is available on Github and our new benchmark datasets are released on HuggingFace.

## 1 Introduction

The capabilities of large language models (LLMs) have drastically improved in recent years, prompting increased efforts to deploy LLMs in real-world applications. However, accompanying this push for practical LLM deployment are growing concerns around data issues regarding LLMs that may threaten to derail such developments, especially since LLMs typically require large amounts of training data. Data owners have raised intellectual property (IP) infringement concerns: For example, New York Times has sued OpenAI over its LLM's use of their copyrighted work (Grynbaum & Mac, 2023). Many jurisdictions are also paying increased scrutiny over *data privacy* concerns, e.g., with regulations such as GDPR (2016) and California Consumer Privacy Act (CCPA, 2018) mandating the "right to be forgotten" that allow data owners to request the erasure of their data from the trained models. Furthermore, it is not uncommon for public data to become outdated or be found erroneous/ harmful, e.g., the retraction of public scientific papers with fabricated data (Hu et al., 2024).

These data concerns have sparked considerable research efforts on LLM unlearning algorithms, which aim to efficiently remove the influence of a subset of the model's original training data (called the *forget set*) while avoiding the prohibitively expensive alternative of retraining the LLM from scratch on the *retain set*. However, due to the size and complexity of LLMs, existing unlearning algorithms cannot yet achieve perfect unlearning like retraining: They may not fully remove the influence of all data in the forget set, and may also inadvertently remove the influence of data in the retain set that should be preserved (Maini et al., 2024; Shi et al., 2025). *How can we measure the extent to which these algorithms have unlearned a given set of data?* Existing works have largely proposed utility-centric unlearning metrics that evaluate unlearning based on model utility (performance) indicators, like the perplexity or accuracy on downstream tasks. After unlearning, the model utility indicators related to the forget set are expected to worsen. We provide an overview of existing utility,

---

*Equal contribution.

membership inference attack, and image and classification watermark-based unlearning metrics in App. A.1 and position our work with respect to other LLM unlearning evaluation works in App. A.2.

However, *are the utility-centric metrics effective in the face of practical challenges with real-world datasets?* One such challenge is that the forget and retain sets usually have semantically similar content. As existing benchmark datasets (Li et al., 2024b; Maini et al., 2024; Shi et al., 2025) do not explicitly consider a higher level of data similarity, we first propose **(a)** a new benchmark dataset called `WaterDrum-Ax` that includes data from multiple data owners and contains duplicates with different levels of data similarity for a more practical evaluation of the unlearning metrics and algorithms (Sec. 2). Using `WaterDrum-Ax`, we observe that utility-centric metrics fall short, because to evaluate the success of unlearning, they need to reference the retrained LLMs (on the retain set), which are prohibitively costly to obtain in practice. Also, expecting a worse utility on the forget set after unlearning ignores the LLMs' ability to generalize from retain set (Liu et al., 2024).

In this work, we consider the above limitations in **(b)** defining clear desiderata that an effective and practical unlearning metric should satisfy to enable *direct interpretation* (Sec. 3). Next, we **(c)** propose a novel *data-centric* metric to *continuously* evaluate the success/*extent of LLM unlearning* instead, which we call **Water**mark-based **D**ata-cent**r**ic **U**nlearning **M**etric (`WaterDrum`) that satisfies these desiderata. `WaterDrum` is based on a robust text watermarking framework that is capable of verifying multiple data owners' watermarks in the text outputs of the LLM when fine-tuned on their watermarked text data (Sec. 4). Our key insight is that using watermarked data creates a clear counterfactual: A model not trained on watermarked data would not contain the watermark signal. In Sec. 5, we **(d)** empirically show that our proposed metric `WaterDrum` significantly outperforms existing ones at satisfying our desiderata. We **(e)** also benchmark unlearning algorithms using `WaterDrum` to reveal their strengths and weaknesses.

## 2 PROBLEM FORMULATION

We consider the setting of a collection $\mathcal{T}$ of data owners where each data owner $i$ has a dataset $\mathcal{D}_i$. These datasets may contain similar data instances (e.g., news articles on the same event from different news agencies or arXiv paper abstracts from the same academic subject category but different authors, as illustrated in App. B.4). The model owner aggregates their data $\mathcal{D}_{\mathcal{T}} := \bigcup_{i \in \mathcal{T}} \mathcal{D}_i$ for training an LLM $\varphi_{\mathcal{T}}$ to be deployed as a service. We consider the unlearning scenario where a subset $\mathcal{F} \subset \mathcal{T}$ of data owners requests to remove the influence of their to-be-erased data $\mathcal{D}_{\mathcal{F}} := \bigcup_{i \in \mathcal{F}} \mathcal{D}_i$ (called the *forget set*) from the LLM due to concerns about privacy, IP protection, or erroneous content.

Ideally, the model owner would retrain a new model $\varphi_{\mathcal{R}}$ on the remaining set of data $\mathcal{D}_{\mathcal{R} := \mathcal{T} \setminus \mathcal{F}}$ (called the *retain set*) to comply with these unlearning requests. However, full retraining is impractical in reality due to the prohibitive computational cost, especially when $\mathcal{D}_{\mathcal{R}}$ is large. Instead, the model owner would resort to using some *unlearning algorithm*, which modifies the original model $\varphi_{\mathcal{T}}$ based on $\mathcal{D}_{\mathcal{F}}$ to an *unlearned model* $\widetilde{\varphi}$ that approximates $\varphi_{\mathcal{R}}$. Such an unlearned model may not have perfectly unlearned the forget set, so it can be intuitively viewed as retaining the influence of some (possibly unknown) subset of the forget set $\mathcal{D}_{\mathbb{O}} \subseteq \mathcal{D}_{\mathcal{F}}$ and hence still be effectively influenced by its **approximate retain set** $\mathcal{D}_{\mathcal{R}} \bigcup \mathcal{D}_{\mathbb{O}}$. Note that $\mathcal{D}_{\mathbb{O}}$ might not correspond exactly to the union of $\mathcal{D}_i$'s over some subset of data owners in $\mathcal{F}$ and can possibly include only a subset of data points from each $\mathcal{D}_i$. The best unlearned models should have $|\mathcal{D}_{\mathbb{O}}|$ and its influence to be as small as possible.

The model owner should allow each data owner $i \in \mathcal{F}$ to evaluate the extent to which its data $\mathcal{D}_i$ has been unlearned, and would usually only grant them **query access to the LLM**. Let each data point $d$ be used to form a text query $q_d$. For example, $q_d$ can be a formatted prompt to an LLM for QA or completion tasks. Then, both the model owner and data owner $i$ can rely on some LLM $\varphi_{\bullet}$'s text output $\varphi_{\bullet}(q_d)$ to compute an *unlearning metric $M$* that quantifies the extent to which $i$'s data remains present. We define an unlearning metric $M$ where $M(\varphi_{\bullet}(q_d), i)$ measures the influence of data $\mathcal{D}_i$ from owner $i$ (i.e., second input to $M$) detectable in the LLM's text output $\varphi_{\bullet}(q_d)$ to query $q_d$.

The unlearning metric should also be able to measure the influence of data from a set of owners; for example, $M(\varphi_{\bullet}(q_d), \mathcal{F})$ measures the influence of the forget set $\mathcal{D}_{\mathcal{F}}$ detectable in the LLM's text output. Usually, we set the influence as the extent to which data point $d_i$ from some owner $i \in \mathcal{F}$ remains present in the LLM's text output $\varphi_{\bullet}(q_{d_i})$ to its query $q_{d_i}$, i.e., $M(\varphi_{\bullet}(q_{d_i}), \mathcal{F}) = M(\varphi_{\bullet}(q_{d_i}), i)$. Often, we measure the influence of $\mathcal{D}_{\mathcal{F}}$ detectable in the LLM's text outputs to an aggregate of queries formed by owners' data (e.g., $\mathcal{D}_{\mathcal{F}}$). With a slight abuse of notation, we denote

such an *aggregate unlearning metric* as $M_{d \in \mathcal{D}_\mathcal{F}}(\varphi_\bullet(q_d), \mathcal{F})$, which can be, for example, the uniform average over all $d \in \mathcal{D}_\mathcal{F}$: $\sum_{d \in \mathcal{D}_\mathcal{F}} M(\varphi_\bullet(q_d), \mathcal{F})/|\mathcal{D}_\mathcal{F}| = \sum_{i \in \mathcal{F}, d_i \in \mathcal{D}_i} M(\varphi_\bullet(q_{d_i}), \mathcal{F})/|\mathcal{D}_\mathcal{F}|$.

**Existing datasets to benchmark unlearning algorithms do not reflect practical challenges:** Existing works have proposed to evaluate unlearning algorithms and metrics using benchmark datasets like TOFU (Maini et al., 2024), MUSE (Shi et al., 2025), and WMDP (Li et al., 2024b) with the following properties: (a) The forget set $\mathcal{D}_\mathcal{F}$ and retain set $\mathcal{D}_\mathcal{R}$ are fixed. In contrast, in practice, multiple owners can independently decide whether to erase their data, which requires evaluation on **multiple forget-retain splits**. (b) $\mathcal{D}_\mathcal{F}$ and $\mathcal{D}_\mathcal{R}$ are disjoint (i.e., the queries formed by $\mathcal{D}_\mathcal{F}$ are related only to $\mathcal{D}_\mathcal{F}$ and are unrelated to queries formed by $\mathcal{D}_\mathcal{R}$) and the unlearning algorithms perform poorly if dependencies between both sets are introduced (Thaker et al., 2024). In contrast, real-world datasets may **contain more similar data and different levels of similarity across $\mathcal{D}_\mathcal{F}$ and $\mathcal{D}_\mathcal{R}$**.

To address these limitations, we introduce a complementary unlearning benchmark dataset called `WaterDrum-Ax` that comprises arXiv paper abstracts across various academic subject categories published after the release of the Llama-2 model. In particular, to address (a) and (b) above, `WaterDrum-Ax` includes (a) abstracts from the 20 most popular academic subject categories to represent 20 different data owners that can be freely assigned to define $\mathcal{D}_\mathcal{F}$ and $\mathcal{D}_\mathcal{R}$; and (b) different levels of data similarity ranging from exact duplicates to paraphrased versions of the abstracts that can be used across $\mathcal{D}_\mathcal{F}$ and $\mathcal{D}_\mathcal{R}$. Overall, `WaterDrum-Ax` contains 400 abstracts for each category, aggregating to a total of 8000 data points in `WaterDrum-Ax`. These abstracts have an average length of 260 tokens, which is considerably longer than that of TOFU (Maini et al., 2024) (59 tokens).

The `WaterDrum-Ax` benchmark dataset can be used to (i) evaluate unlearning metrics based on the desiderata introduced in Sec. 3, and (ii) evaluate unlearning algorithms using effective and practical metrics identified in (i). The empirical evaluations in Sec. 5 focus on (i) but include some preliminary results on (ii) in Sec. 5.1. We leave more systematic investigations of (ii) to future work.

**Existing unlearning metrics are ineffective in the face of practical challenges:** Here, we discuss some existing definitions of the unlearning metric $M$ and their limitations; see App. A.1 for a deeper introduction of utility-centric and other unlearning metrics. *Utility-centric* unlearning metrics have evaluated the effectiveness of unlearning based on model utility (performance) indicators, such as verbatim memorization, perplexity, or accuracy on downstream tasks. Performance indicators $P$ have compared the unlearned LLM $\widetilde{\varphi}$'s text outputs to queries (e.g., $\widetilde{\varphi}(q_d)$ for all $d \in \mathcal{D}_\mathcal{F}$) to the original text data (e.g., $\mathcal{D}_\mathcal{F}$). For instance, ROUGE-L (Maini et al., 2024) compares the output phrasing/longest common subsequence of $\widetilde{\varphi}(q_d)$ to the training text data point $d$. As another example, some membership inference attack (MIA) based unlearning metrics (Shokri et al., 2017), such as that of Shi et al. (2024) for LLMs, are utility-centric as they may depend on the log-likelihood of tokens of the original text data.

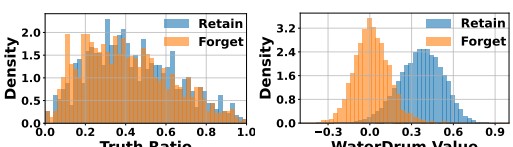

Figure 1: Histograms of utility-centric Truth Ratio metric vs. `WaterDrum` values under the 'semantic duplicate' setting of data similarity for `WaterDrum-TOFU` dataset (Table 2 in Sec. 5) where the individual metric values $M(\varphi_\mathcal{R}(q_d), \mathcal{F})$ for each $d \in \mathcal{D}_\mathcal{F}$ are in orange and $M(\varphi_\mathcal{R}(q_d), \mathcal{R})$ for each $d \in \mathcal{D}_\mathcal{R}$ are in blue. The Truth Ratio metric values cannot be interpreted on their own as there is no value on the horizontal axis where we can confidently conclude that the LLM's text output $\varphi_\mathcal{R}(q_d)$ to query $q_d$ is more likely to be formed by any $d \in \mathcal{D}_\mathcal{F}$. In contrast, `WaterDrum` values can be interpreted on their own: Values $< 0.2$ are more likely to be associated with forget set $\mathcal{D}_\mathcal{F}$.

Our key observation is that **utility-centric metric values**, such as $P(\widetilde{\varphi}(q_d), \mathcal{F})$ for each $d \in \mathcal{D}_\mathcal{F}$ and their aggregate (e.g., average of ROUGE-L), **cannot be interpreted on their own**: For example, Table 3 of Shi et al. (2025) compares the aggregate metric value (e.g., KnowMem) on the unlearned LLM $\widetilde{\varphi}$ with that on the retrained LLM $\varphi_\mathcal{R}$ (trained on the retain set) to evaluate the extent of unlearning. Ideally, the aggregate (over all $d \in \mathcal{D}_\mathcal{F}$) of $P(\widetilde{\varphi}(q_d), \mathcal{F})$ should be equal to that of $P(\varphi_\mathcal{R}(q_d), \mathcal{F})$. Similarly, the work of Maini et al. (2024) also compares the distribution of the metric values (e.g., Truth Ratio, ROUGE-L) on the unlearned LLM $\widetilde{\varphi}$ with that on the retrained LLM $\varphi_\mathcal{R}$ to evaluate the extent of unlearning.

This raises a critical issue: In practice, the retrained LLM $\varphi_\mathcal{R}$ is usually not available (Sec. 2). In fact, the aim of unlearning algorithms (and metrics) is to produce an unlearned LLM $\widetilde{\varphi}$ that most

closely approximates $\varphi_{\mathcal{R}}$. Regarding our main research question, **is there an (aggregate) unlearning metric (over all $d \in \mathcal{D}_{\mathcal{F}}$) whose values can be interpreted on their own to measure the extent of unlearning $\mathcal{D}_{\mathcal{F}}$ *without* referencing a retrained LLM?**

It can be observed from Fig. 1 that **the answer is no for utility-centric metrics**, especially when there are similar data in the retain and forget sets: For any Truth Ratio metric value (e.g., 0.2), the LLM's text output to a query is equally likely to be formed by a data point in the retain set vs. that in the forget set. There is no value of $\kappa$ where we can confidently conclude that the LLM's text output $\varphi_{\mathcal{R}}(q_d)$ to query $q_d$ is more likely to be formed by any $d \in \mathcal{D}_{\mathcal{F}}$ when $P(\varphi_{\mathcal{R}}(q_d), \mathcal{F}) < \kappa$. Let $d_i \simeq d_j$ denote that text data points $d_i$ and $d_j$ have a large *similarity score $SS(d_i, d_j)$*, e.g., computed using some semantic text similarity (STS) score. A likely explanation is that when similar text data points $d_f$ and $d_r$ are present in the respective forget and retain sets, (i) any unlearned LLM $\widetilde{\varphi}$ (e.g., retrained LLM $\varphi_{\mathcal{R}}$) tends to generate similar text outputs to queries formed by both sets, i.e., $\widetilde{\varphi}(q_{d_f}) \simeq \widetilde{\varphi}(q_{d_r})$, as empirically verified in App. G.2. As performance indicators largely depend on direct comparisons with the LLM's text outputs, their metric values are also similar. (ii) Expecting poor predictions on the forget set overlooks the generalization capability of LLMs (Liu et al., 2024).

## 3 UNLEARNING METRIC DESIDERATA

The goal of our work here is to come up with an alternative effective and practical unlearning metric whose values can be interpreted on their own *without* referencing a retrained LLM. **What desiderata must such an unlearning metric satisfy?** We define a few non-exhaustive desiderata in this section.

Intuitively, based on Fig. 1, we would want the LLM's text outputs to queries formed by the text data points in the forget vs. retain sets to have (i) *separable metric values* and (ii) aggregate (e.g., average) metric values to be easily interpreted (e.g., 0 for perfect unlearning) (iii) without referencing the retrained model. (i) and (ii) correspond to our desiderata **D1** and **D2**, respectively. We include practical constraint (iii) as **D3**. The unlearning metric $M$ should satisfy the following desiderata:

EFFECTIVENESS. First, the metric must effectively measure the extent to which an unlearning algorithm has not unlearned the forget set (so, the resulting unlearned LLM $\widetilde{\varphi}$ would still be influenced by its unknown approximate retain set, as discussed in Sec. 2). To achieve this, we will now define effectiveness desiderata that utilize LLMs retrained on the retain set (and varying known subsets of the forget set) as retraining is a perfect unlearning algorithm:[1]

**D1 Separability.** The metric should detect/classify whether an owner's data still influences an unlearned LLM. Specifically, when evaluating the retrained LLM $\varphi_{\mathcal{R}}$ (i.e., achieved by perfect unlearning), the metric should, with high probability, **give higher values when measured on its text outputs to queries formed by the retain set $\mathcal{D}_{\mathcal{R}}$** (which influences $\varphi_{\mathcal{R}}$) **than queries formed by the forget set $\mathcal{D}_{\mathcal{F}}$** (which does not). That is, for any randomly selected text data points $d_r \in \mathcal{D}_r \subseteq \mathcal{D}_{\mathcal{R}}$ from owner $r$ and $d_f \in \mathcal{D}_f \subseteq \mathcal{D}_{\mathcal{F}}$ from owner $f$, the probability

$$\mathbb{P}\left[M(\varphi_{\mathcal{R}}(q_{d_r}), r) > M(\varphi_{\mathcal{R}}(q_{d_f}), f)\right] \approx 1 . \tag{1}$$

Separability, which is defined by the left-hand side expression of Eq. (1) (or, equivalently, AUROC), implies that some threshold $\kappa$ exists such that for any text data point $d_i \in \mathcal{D}_i \subseteq \mathcal{D}_{\mathcal{T}}$ from owner $i$, a large value $M(\varphi_{\mathcal{R}}(q_{d_i}), i) > \kappa$ indicates that $d_i$ is likely to be in the retain set $\mathcal{D}_{\mathcal{R}}$; varying $\kappa$ yields the ROC curve. Similarly, when considering an unlearned LLM $\widetilde{\varphi}$, a large value $M(\widetilde{\varphi}(q_{d_i}), i)$ indicates that $d_i$ is likely to be in the approximate retain set (Sec. 2). In other words, the metric should serve as a good classifier for whether an owner's data still influences the LLM and is hence in the approximate retain set: A higher AUROC indicates a better separability of data that influences the LLM vs. not (Fawcett, 2006). App. B.1 gives a further intuitive discussion on **D1**.

**D2 Calibration.** In Sec. 1, we have highlighted that existing unlearning algorithms cannot yet achieve perfect unlearning. Thus, our unlearning metric should be **calibrated to the extent of imperfect unlearning**. For example, we can simulate different extents of imperfect unlearning by retraining with different sizes of subsets of the forget set. Specifically, the aggregate metric (in expectation) should be proportional to the size $k$ of the random subset $\mathcal{D}_{\bullet}$ of the forget set that is used together with the retain set $\mathcal{D}_{\mathcal{R}}$ to retrain the LLM $\widehat{\varphi}$:

---

[1]Note that these retrained LLMs are only used to justify our effectiveness desiderata for evaluating the unlearning metrics. In practice, the metrics should be used to evaluate imperfect unlearning algorithms without referencing the retrained LLMs, as discussed in **D3**(a).

$$\mathbb{E}_{\mathcal{D}_\bullet \subseteq \mathcal{D}_\mathcal{F} : |\mathcal{D}_\bullet| = k} \left[ M_{d \in \mathcal{D}_\mathcal{F}}(\widehat{\varphi}(q_d), \mathcal{F}) \right] \quad \propto \quad k / |\mathcal{D}_\mathcal{F}| \tag{2}$$

where $\mathcal{D}_\bullet$ is defined in a similar way as $\mathcal{D}_\mathbb{O}$ in Sec. 2 except that it is known. Eq. (2) implies that a perfectly unlearned LLM like $\varphi_\mathcal{R}$ should have $M_{d \in \mathcal{D}_\mathcal{F}}(\varphi_\mathcal{R}(q_d), \mathcal{F}) = 0$ since $k = 0$. So, when evaluating unlearning algorithms, we identify successful perfect unlearning of the forget set by looking for $M_{d \in \mathcal{D}_\mathcal{F}}(\widetilde{\varphi}(q_d), \mathcal{F}) \approx 0$. The value of the aggregate metric can also be interpreted as the extent to which the forget set has not been unlearned. This enables the unlearning metric to go *beyond being just a binary indicator* of whether an entire forget set has been unlearned to a meaningful *continuous measure* of unlearning. App. B.2 provides a further intuitive discussion on **D2**.

PRACTICALITY. To be a viable metric for deployment, the metric must also satisfy the following feasibility and robustness desiderata that account for challenges faced in common real-life scenarios:

**D3 Feasibility.** (a) When the metric is used to evaluate an unlearning algorithm and produce $M(\widetilde{\varphi}(q_d), \mathcal{F})$ or the aggregate $M_{d \in \mathcal{D}_\mathcal{F}}(\widetilde{\varphi}(q_d), \mathcal{F})$ on the unlearned LLM $\widetilde{\varphi}$, it **should not require the retrained LLM $\varphi_\mathcal{R}$ to interpret/measure the extent of imperfect unlearning**. The premise of unlearning is that retraining the LLM on the retain set is prohibitively expensive. Hence, metrics cannot depend on $\varphi_\mathcal{R}$ in practice. (b) To additionally enable data owners with only query access to the LLM to evaluate unlearning, the metric **should only depend on the queried text outputs** instead of full access to the weights or token probabilities of the unlearned model $\widetilde{\varphi}$.

**D4 Robustness to similar data.** The effectiveness desiderata **D1**-**D2** should hold for any $\mathcal{D}_\mathcal{R}$ and $\mathcal{D}_\mathcal{F}$, including typical scenarios where $\mathcal{D}_\mathcal{R}$ and $\mathcal{D}_\mathcal{F}$ have similar data points (e.g., news agencies have different news articles reporting on the same event, as illustrated in App. B.4).

EXISTING METRICS DO NOT SATISFY DESIDERATA. Continuing the discussion from Sec. 2, Table 1 compares our `WaterDrum` and existing metrics based on the proposed desiderata in Sec. 3. As other utility-centric metrics may not satisfy **D1** and **D2** under **D3** and **D4**, their values cannot directly interpret/measure the extent of imperfect unlearning without a retrained LLM and using the `WaterDrum-Ax` dataset (Sec. 2).

## 4 WATERMARKING FRAMEWORK

Instead of relying on utility-centric metrics that indirectly infer unlearning via model performance, we propose a novel *data-centric* metric that **directly tracks the influence of data by actively embedding data-specific signals detectable in the LLM's text outputs**. These data signals are embedded by watermarking the training data and preserved by the LLM. We discuss how `WaterDrum` differs from existing watermark-based metrics for image classification tasks in App. A.1 and give an introduction of text watermarking in App. A.3. We will start by outlining desiderata required by a watermarking framework (and its verification operator) to meet our unlearning metric desiderata in Sec. 3.

Table 1: Comparison of unlearning metrics based on the proposed desiderata (Sec. 3). We enforce **D3**, so metrics cannot rely on the retrained LLM. **D1** and **D2** consider the setting of no data similarity.

|  | D1 | D2 | D4 |
|---|---|---|---|
| ROUGE (Maini et al., 2024) | ✓ | ✗ | ✗ |
| Truth Ratio (Maini et al., 2024) | ✗ | ✗ | ✗ |
| KnowMem (Shi et al., 2025) | ✗ | ✗ | ✗ |
| MIA (Shi et al., 2024) | ✗ | ✗ | ✗ |
| `WaterDrum` (ours) | ✓ | ✓ | ✓ |

WATERMARKING DESIDERATA. Our watermarking framework assigns each data owner $i$ a watermark key $\mu_i$. It comprises (a) a **watermarking operator** $\mathcal{W}(d_i, \mu_i) \rightarrow d_i'$ that takes in any text data point $d_i \in \mathcal{D}_i$ from owner $i$ and watermarks it with the key $\mu_i$ to produce a unique corresponding text data point $d_i'$, and (b) a **verification operator** $\mathcal{V}(g', \mu_i)$ that takes in any text data $g'$ (e.g., LLM's text output) and a watermark key $\mu_i$ and provides a score reflecting the likelihood of $g'$ containing the watermark $\mu_i$. To satisfy our unlearning metric desiderata in Sec. 3, the watermark and verification operators used will need to satisfy the following desiderata:[2]

**W0 Fidelity.** The watermarking should have minimal impact on the semantic similarity of the original data, i.e., $d \simeq \mathcal{W}(d, \mu)$ for any watermark key $\mu$ and data $d \in \mathcal{D}_\mathcal{T}$. While this does not directly impact the unlearning metric desiderata, **W0** ensures that the watermarking process preserves the value of the data and model for the model owner and the metric can be deployed in practice.

---

[2]When evaluating unlearning algorithms (Sec. 5.1), the model owner can perform the watermarking and verification. In real-world deployment, the data owners do so instead.

Figure 2: (Left) Unlike existing utility-centric metrics, `WaterDrum` satisfies the unlearning metric desiderata in Sec. 3. `WaterDrum` is robust to similar data by embedding orthogonal data-specific signals in the LLM's text outputs that are **W1** verifiable. (Right) An overview of the watermarking, training, unlearning, and verification processes of `WaterDrum`.

**W1 Verifiability.** (a) The watermark should be verifiable if and only if the watermarked content is present in the LLM. In our setting, this means the retrained LLM should not contain the watermark of an owner $f$ in $\mathcal{F}$ who requests to erase its data from the LLM, i.e., $\mathcal{V}(\varphi_{\mathcal{R}}(q_{d_f}), \mu_f) = 0$. In contrast, an LLM that has been trained on owner $r$'s data $\mathcal{D}_r \subseteq \mathcal{D}_{\mathcal{R}}$ should be verifiable with watermark key $\mu_r$, i.e., $\mathcal{V}(\varphi_{\mathcal{R}}(q_{d_r}), \mu_r) \gg 0$ for all $d_r \in \mathcal{D}_r$. (b) If every text data point in $\mathcal{D}_{\mathcal{F}}$ is watermarked with the same key $\mu_{\mathcal{F}}$, the average of $\mathcal{V}(\widehat{\varphi}(q_{d_f}), \mu_{\mathcal{F}})$ over all $d_f \in \mathcal{D}_{\mathcal{F}}$ for model $\widehat{\varphi}$ retrained on $\mathcal{D}_{\mathcal{R}} \bigcup \mathcal{D}_{\bullet}$ should be proportional to the size of the data $\mathcal{D}_{\bullet} \subseteq \mathcal{D}_{\mathcal{F}}$. (a) supports **D1** as $\mathcal{V}(\varphi_{\mathcal{R}}(q_{d_i}), \mu_i)$ can be used to classify whether an owner's data influences a perfectly unlearned LLM: A value near 0 or much larger than 0, respectively, indicates that owner $i$ likely has no influence or some influence on the unlearned LLM. Together, (a) and (b) support **D2** as the value is 0 in the case of a perfectly unlearned LLM like $\varphi_{\mathcal{R}}$ and the average value is proportional to the extent of imperfect unlearning.

**W2 Overlap verifiability.** The verifiability desideratum **W1** is satisfied despite the presence of other watermarks (e.g., $\mu_r$ from another owner $r$) in the data for training the LLM. This allows for multiple watermarks to be verified from the text outputs of the same LLM.

We also need desiderata on the watermarking process to meet the rest of unlearning metric desiderata:

**W3 Query access constraint.** Data owners should verify their watermarks with only query access to the LLM. This supports **D3** with feasible & efficient evaluation of the extent of imperfect unlearning.

**W4 Unique key.** Each data owner $i$'s watermark key $\mu_i$ should be unique. When an owner requests to erase its data from the LLM, the corresponding forget set would have a different watermark from that associated with the retain set, thus supporting **D1**. The unique keys also ensure that similar or even identical data from different owners would have different watermarks, which supports **D4**.

Fig. 2(left) shows how a watermarking framework satisfying these desiderata satisfies the unlearning metric desiderata in Sec. 3. Concretely, we define a metric $M'$ using the verification operator:

$$M'(\varphi_{\bullet}(q_d), i) := \mathcal{V}(\varphi_{\bullet}(q_d), \mu_i) . \tag{3}$$

OVERVIEW OF WATERDRUM. Can any watermarking framework be adapted to satisfy our proposed watermarking desiderata above? Here, we propose the first data-centric unlearning metric called `WaterDrum` built on top of our adaptation of the training-free, scalable, and robust `Waterfall` framework (Lau et al., 2024) that can successfully and efficiently verify multiple data owners' watermarks in the text outputs of the LLM when trained on their watermarked text data. While we will mainly use `Waterfall` to demonstrate the effectiveness and practicality of `WaterDrum`, other watermarking methods satisfying our desiderata can be adopted as well. We provide a comprehensive overview of watermarking methods (e.g., (Dathathri et al., 2024; Kirchenbauer et al., 2023; Kuditipudi et al., 2024)) in App. A.3 and discuss their adaptations and empirical performance in Sec. 5.2.

Specifically, we adopt the watermarking $\mathcal{W}(\cdot, \mu)$ and verification $\mathcal{V}(\cdot, \mu)$ operators as defined in `Waterfall` (respectively, Algorithms 1 and 2) and summarized in App. C.1 due to lack of space. `Waterfall`'s watermarking and verification operators satisfy the watermarking desiderata **W0**, **W1**(a), and **W2**, as elaborated and demonstrated in (Lau et al., 2024). We have empirically verified that `Waterfall` satisfies **W0** in App. G.1 and **W1**(b) on calibration in Sec. 5. The rest of the

watermarking desiderata can be satisfied by an appropriate design of the unlearning and verification processes, which we illustrate in Fig. 2(right) and present below:

**P1 Watermarking setup.** Each data owner $i$ first watermarks its data $\mathcal{D}_i$ with a unique key $\mu_i$ to generate a watermarked dataset $\mathcal{D}'_i := \{d'_i := \mathcal{W}(d_i, \mu_i)\}_{d_i \in \mathcal{D}_i}$. Then, the model owner aggregates their watermarked data $\mathcal{D}'_{\mathcal{T}} := \bigcup_{i \in \mathcal{T}} \mathcal{D}'_i$, trains an LLM $\varphi'_{\mathcal{T}}$ on it, and offers to clients (including data owners) query access to the trained LLM.

**P2 Unlearning.** A subset of data owners $\mathcal{F}$ requests for their data $\mathcal{D}'_{\mathcal{F}} := \bigcup_{i \in \mathcal{F}} \mathcal{D}'_i$ to be erased from the LLM $\varphi'_{\mathcal{T}}$. The model owner will claim to have performed the unlearning and offer query access to the resulting unlearned LLM $\widetilde{\varphi}'$.

**P3 Unlearning verification.** The verification operator plays the role of an unlearning metric in WaterDrum, as per Eq. (3). Each data owner $i$ in $\mathcal{F}$ can query the unlearned LLM $\widetilde{\varphi}'$ with queries $q_{d'}$ based on $d' \in \mathcal{D}'_i$ and apply the verification operator $\mathcal{V}(\widetilde{\varphi}'(q_{d'}), \mu_i)$ to measure the extent to which its data remains present in the text outputs $\widetilde{\varphi}'(q_{d'})$ and hence has not been unlearned.

Note that computing the WaterDrum value in Eq. (3) applied during **P3** only requires query access to the LLM, hence satisfying **W3**. Watermarking desideratum **W4** is also satisfied by the setup in **P1** and the fact that the model owner never requires the data owners' keys, which is also the case in **P2**.

*Remark* 1. Using watermarked data is both **(i) beneficial and important for identifying practical and effective unlearning metrics** and **(ii) reasonable going forward**. (i) In Table 1, our watermarked data-based WaterDrum is the *only* metric that satisfies all the unlearning metric desiderata. (ii) There are a few important reasons: (a) data owners with IP or privacy rights (Sec. 1) can require the model owner to use the watermarked version of their released data instead; (b) data owners can watermark their unreleased data, which will be used to train (and may be more relevant for) future LLMs, and (c) the adoption of text watermarking is expected to grow and match the prevalence of image watermarking. In App. C.4, we elaborate on these practical considerations and other benefits (e.g., computationally lightweight, no change to existing ML pipelines), beyond meeting the desiderata, for deploying WaterDrum. So, the benefits of using watermarked data in WaterDrum to evaluate unlearning algorithms, such as not needing to reference a retrained LLM (unlike utility-centric metrics), outweigh the slight inconvenience and cost.

*Remark* 2. If (i) the model owner tries to reduce its LLM's metric value without directly performing unlearning or copyright its LLM via watermarking methods for a model owner (Kirchenbauer et al., 2023) or (ii) data owners try to report inflated metric values to understate the unlearning by the model owner, is WaterDrum still an effective unlearning metric? The answer is yes if (i) the watermarking framework is designed to be *resilient*, the watermark keys are *private* to the data owners, and (ii) a trusted third party validates the metric values. We discuss these questions and additional requirements in App. D and show that WaterDrum also satisfies them due to the properties of Waterfall.

## 5 EXPERIMENTS AND DISCUSSION

**Experimental setup.** In this section, we empirically compare WaterDrum with other commonly used unlearning metrics: ROUGE-L (Lin, 2004; Maini et al., 2024), Truth Ratio (Maini et al., 2024), KnowMem (Shi et al., 2025), and MIA (Shi et al., 2024)). We use the Llama-2-7B (Touvron et al., 2023) as the base LLM. For WaterDrum, the LLM is fine-tuned on the watermarked dataset $\mathcal{D}'_{\mathcal{T}}$ in WaterDrum-Ax (Sec. 2) or WaterDrum-TOFU derived from TOFU (Maini et al., 2024). For other metrics, the LLM is instead fine-tuned on their unwatermarked version $\mathcal{D}_{\mathcal{T}}$. To ease comparison, all metrics are scaled to 1.0 when evaluated on the original model $\varphi_{\mathcal{T}}$ before unlearning. We use 1 category from WaterDrum-Ax and 10% data from WaterDrum-TOFU as forget sets. We further evaluate the metrics with other LLMs (Li et al., 2023b) as base models (App. F.3). Although watermarking with Waterfall is only essential for WaterDrum, App. G.1 shows that it does not degrade the LLM's performance and App. F.4 shows that other metrics still do not satisfy some desiderata when the LLM is fine-tuned on $\mathcal{D}'_{\mathcal{T}}$ instead. To ease notation, in the rest of this paper, we will use $d_\bullet$, $\mathcal{D}_\bullet$, $q_\bullet$, $\varphi_\bullet$, and $\widetilde{\varphi}$ in place of $d'_\bullet$, $\mathcal{D}'_\bullet$, $q'_\bullet$, $\varphi'_\bullet$, and $\widetilde{\varphi}'$ (i.e., those associated with the watermarked data used by WaterDrum), respectively. App. E gives additional details on the datasets, other models used, unlearning metrics, inference parameters, queries, and implementation.

Table 2: AUROC ($\pm$ across 3 seeds) of various unlearning metrics under different levels of data similarity for the `WaterDrum-TOFU` and `WaterDrum-Ax` datasets. `WaterDrum`'s AUROC remains near 1.0 even when similar data exists.

| Data Similarity | WaterDrum-TOFU | | | WaterDrum-Ax | | |
|---|---|---|---|---|---|---|
| | ROUGE | Truth Ratio | WaterDrum | ROUGE | KnowMem | WaterDrum |
| Exact Duplicate | 0.510±0.007 | 0.508±0.008 | **0.926±0.027** | 0.334±0.005 | 0.492±0.005 | **0.957±0.008** |
| Semantic Duplicate | 0.798±0.001 | 0.472±0.054 | **0.954±0.001** | **0.960±0.002** | 0.450±0.007 | **0.963±0.001** |
| No Duplicate | **0.908±0.005** | 0.747±0.011 | **0.928±0.026** | **0.974±0.001** | 0.491±0.008 | **0.965±0.002** |

We will evaluate `WaterDrum` and the baseline metrics in experimental settings that mimic the real-life scenarios described in the PRACTICALITY DESIDERATA **D3** and **D4** (Sec. 3). Then, under these settings, we assess the effectiveness of various metrics based on **D1** and **D2** by considering how they evaluate the perfect unlearning algorithm—retraining the base LLM on only the retain set to obtain $\varphi_{\mathcal{R}}$, which is guaranteed to contain no influence of forget set $\mathcal{D}_{\mathcal{F}}$ by construction.

**Feasibility D3.** To satisfy **D3**(a), the metrics should not require referencing the retrained LLM $\varphi_{\mathcal{R}}$ to interpret/measure the extent of imperfect unlearning (Sec. 3). For example, when assessing **D2**, we enforce the metric values not to use (e.g., subtract) $M_{d \in \mathcal{D}_{\mathcal{F}}}(\varphi_{\mathcal{R}}(q_d), \mathcal{F})$. To satisfy **D3**(b), the metric should not require logit access, but for evaluation, we allow the use of logits only to compute MIA.

**Robustness to similar data D4.** Let $\mathcal{D}_i \simeq \mathcal{D}_j$ denote sets where for any $d_i \in \mathcal{D}_i$, there is a corresponding $d_j \in \mathcal{D}_j$ such that $d_i \simeq d_j$. We establish the settings to assess the robustness of the unlearning metrics to similar data by injecting a small amount of data $\mathcal{D}_s \simeq \mathcal{D}_{\mathcal{F}}$ into $\mathcal{D}_{\mathcal{R}}$, i.e., the retain set is augmented ($\mathcal{D}_{\mathcal{R}}^s \coloneqq \mathcal{D}_s \bigcup \mathcal{D}_{\mathcal{R}}$) with some data points that are similar to $\mathcal{D}_{\mathcal{F}}$. We consider two such settings: (a) **exact duplicate**: data points in $\mathcal{D}_s$ are exact copies of those in $\mathcal{D}_{\mathcal{F}}$ (i.e., $\mathcal{D}_s = \mathcal{D}_{\mathcal{F}}$), and (b) **semantic duplicate**: data points in $\mathcal{D}_s$ are paraphrased versions of those in $\mathcal{D}_{\mathcal{F}}$ (i.e., $\mathcal{D}_s \simeq \mathcal{D}_{\mathcal{F}}$). In addition, we consider the case where (c) **no duplicate** of any data point in $\mathcal{D}_{\mathcal{F}}$ is used to augment $\mathcal{D}_{\mathcal{R}}$ (i.e., $\mathcal{D}_s = \emptyset$). Additional implementation details are in App. E.4.

SEPARABILITY DESIDERATUM **D1**. To assess whether the unlearning metrics satisfy **D1**, note that the left-hand side expression $\mathbb{P}[M(\varphi_{\mathcal{R}}(q_{d_r}), r) > M(\varphi_{\mathcal{R}}(q_{d_f}), f)]$ in Eq. (1) corresponds to the definition of the AUROC of the metric $M$ in measuring the separability of the retain set $\mathcal{D}_{\mathcal{R}}$ that influences the retrained LLM $\varphi_{\mathcal{R}}$ vs. the forget set $\mathcal{D}_{\mathcal{F}}$ that does not (Fawcett, 2006). Hence, we can compute the AUROC of various unlearning metrics with the retrained LLM $\varphi_{\mathcal{R}}$ (i.e., a perfectly unlearned LLM) and assess if they have AUROC $\approx 1$. We exclude MIA from this experiment as it only considers the separability of the forget set vs. holdout set instead of the retain set (App. E.3).

Table 2 shows the AUROC of the various unlearning metrics under different levels of data similarity for the `WaterDrum-TOFU` dataset.[3] Notably, `WaterDrum` is the only metric that consistently achieves AUROC $> 0.9$ and close to 1, hence satisfying **D1**. In contrast, the other metrics' performances degrade significantly under the 'exact and semantic duplicate' settings; for the former, their AUROCs drop to around $0.5$, so the other metrics are no better than random assignment in the separability of $\mathcal{D}_{\mathcal{R}}$ vs. $\mathcal{D}_{\mathcal{F}}$. Furthermore, Truth Ratio only achieves an AUROC of around $0.75$ under the conventional 'no duplicate' setting, hence not satisfying **D1** even in this case.

The results on the `WaterDrum-Ax` dataset in Table 2 show similar trends with `WaterDrum` consistently performing well and KnowMem performing poorly in all settings. ROUGE performs poorly under the 'exact duplicate' setting where only $5\%$ of the augmented retain set are exact copies of those in the forget set. It performs well for the 'semantic duplicate' setting as the mean ROUGE-L recall score between $\mathcal{D}_s$ and $\mathcal{D}_{\mathcal{F}}$ is low ($\approx 0.65$), which implies that the text data is already heavily paraphrased such that the 'semantic duplicate' setting is effectively closer to the 'no duplicate' one for ROUGE. However, the mean semantic text similarity (STS) score of $\mathcal{D}_s$ and $\mathcal{D}_{\mathcal{F}}$ remains high (i.e., 0.94). Milder forms of perturbation for this dataset would likely make the degradation of its performance on **D1** more apparent.

---

[3] Truth Ratio is only applicable to question answering (QA) datasets for which `WaterDrum-Ax` is not. Since `WaterDrum-TOFU` is already a QA dataset, there is no need to consider KnowMem that generates QA pairs for evaluation using the ROUGE-L recall score.

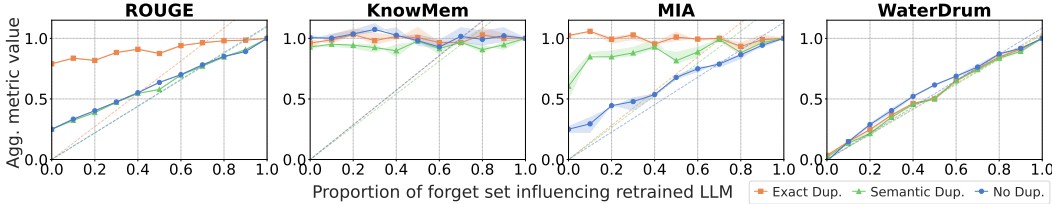

Figure 3: Calibration curves for various unlearning metrics w.r.t. proportion $k/|\mathcal{D}_{\mathcal{F}}|$ of forget set influencing retrained LLM (solid) and their best-fit lines (see associated $R^2$ in Table 3) through origin (dotted) under different levels of data similarity for `WaterDrum-Ax` dataset. Only `WaterDrum` is well-calibrated and satisfies **D2** with its best-fit lines closely following its aggregate metric values.

Table 3: $R^2$ values for the best-fit lines (dotted in Fig. 3) of various unlearning metrics under different levels of data similarity for `WaterDrum-Ax` dataset. `WaterDrum` achieves the highest $R^2$ values that are close to 1 and is hence a well-calibrated metric.

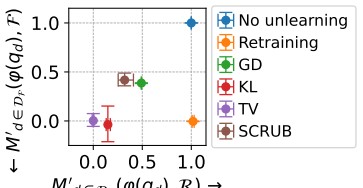

Figure 4: Benchmarking the unlearning algorithms with `WaterDrum`.

| Data Similarity | ROUGE | KnowMem | MIA | WaterDrum |
|---|---|---|---|---|
| Exact Duplicate | -37.47 | -498.1 | -285.6 | **0.987** |
| Semantic Duplicate | 0.693 | -276.5 | -14.52 | **0.991** |
| No Duplicate | 0.650 | -252.9 | 0.677 | **0.963** |

CALIBRATION DESIDERATUM **D2**. Next, we assess whether the unlearning metrics meet the calibration desideratum, as defined in Eq. (2). Failing to meet this desideratum implies that the metrics cannot measure the extent to which the forget set $\mathcal{D}_{\mathcal{F}}$ has not been unlearned (i.e., for imperfect unlearning). To evaluate this, we first retrain LLMs on $\mathcal{D}_{\mathcal{R}} \bigcup \mathcal{D}_{\ominus}$ by varying the size $k$ of the subset $\mathcal{D}_{\ominus} \subseteq \mathcal{D}_{\mathcal{F}}$. Then, we compute the unlearning metrics for each retrained LLM and plot calibration curves showing how the metrics vary with $k$. To quantify how well a metric satisfies Eq. (2), we can compute the $R^2$ value for its best-fit line through the origin since a calibrated metric (in expectation) should be proportional to $k/|\mathcal{D}_{\mathcal{F}}|$ and have $M_{d \in \mathcal{D}_{\mathcal{F}}}(\varphi_{\mathcal{R}}(q_d), \mathcal{F}) = 0$ at $k = 0$. Consequently, an $R^2$ value close to 1 implies that the metric is well-calibrated, while a large negative value occurs when the metric produces similar (instead of proportional) values for varying $k$.

Fig. 3 and Table 3 show, respectively, the calibration curves for the various unlearning metrics and the $R^2$ values for the corresponding best-fit lines under different levels of data similarity for the `WaterDrum-Ax` dataset. The results show that `WaterDrum` is the only well-calibrated metric across all settings that can represent the proportion $k/|\mathcal{D}_{\mathcal{F}}|$ of the forget set still influencing the unlearned LLM (i.e., the extent of imperfect unlearning). Comparatively, other metrics perform poorly across *all settings*, including the conventional 'no duplicate' setting: They cannot be used to determine when $\mathcal{D}_{\mathcal{F}}$ has been perfectly unlearned as $M_{d \in \mathcal{D}_{\mathcal{F}}}(\varphi_{\mathcal{R}}(q_d), \mathcal{F}) \neq 0$ (i.e., calibration curves do not pass through the origin in Fig. 3). Thus, they do not satisfy **D2** when enforcing **D3**.

The results demonstrate the strong reliance of the baseline unlearning metrics on access to the retrained LLM $\varphi_{\mathcal{R}}$. Without knowing the reference value on the perfectly unlearned LLM (i.e., $\varphi_{\mathcal{R}}$), these baselines fail to quantify the extent of imperfect unlearning or even evaluate the success of unlearning. This reliance is impractical as retraining the LLM on the retain set is prohibitively expensive, which motivates the need for unlearning algorithms. Fig. 14 and Table 13 in App. H.2.1 show similar results for the `WaterDrum-TOFU` dataset where all baselines fail to meet the calibration desideratum across all settings, including the 'no duplicate' setting. App. H.1.1 gives more results.

### 5.1 BENCHMARKING UNLEARNING ALGORITHMS ON NEW `WATERDRUM-AX` DATASET

Finally, Fig. 4 illustrates how `WaterDrum` can be used to benchmark unlearning algorithms using `WaterDrum-Ax` via an evaluation plot of $M'_{d \in \mathcal{D}_{\mathcal{F}}}(\widetilde{\varphi}(q_d), \mathcal{F})$ vs. $M'_{d \in \mathcal{D}_{\mathcal{R}}}(\widetilde{\varphi}(q_d), \mathcal{R})$. This measures the aggregate `WaterDrum` values of the respective watermarked forget set $\mathcal{D}_{\mathcal{F}}$ vs. retain set $\mathcal{D}_{\mathcal{R}}$ on the unlearned LLM $\widetilde{\varphi}$ (Sec. 4). The original LLM $\varphi_{\mathcal{T}}$, which trains on both $\mathcal{D}_{\mathcal{F}}$ and $\mathcal{D}_{\mathcal{R}}$ (i.e.,

Table 4: Comparison of `WaterDrum` implemented with various adapted watermarking methods.

| Watermarking Method | D1 Separability (AUROC) | D2 Calibration ($R^2$) | Verification time | Need GPU? |
|---|---|---|---|---|
| Waterfall | **0.965** | 0.963 | **0.015s** | **No** |
| Adapted KGW | 0.871 | **0.996** | 0.336s | Yes |
| Adapted Synth-ID | 0.549 | -16.951 | 0.386s | Yes |
| Adapted EXP-edit | 0.789 | -17.079 | 165.5s | **No** |

no unlearning), is at the top right corner, while the retrained LLM $\varphi_{\mathcal{R}}$, which only trains on $\mathcal{D}_{\mathcal{R}}$ (i.e., perfect unlearning), is at the bottom right corner. It is expected that perfect unlearning would achieve an approximately similar aggregate `WaterDrum` value of $\mathcal{D}_{\mathcal{R}}$ as no unlearning since the retain set still influences both the retrained and original LLMs. In this plot, if an unlearning algorithm can produce an unlearned LLM $\widetilde{\varphi}$ with aggregate `WaterDrum` values closer to that achieved by retraining, then its $\widetilde{\varphi}$ is better at both unlearning $\mathcal{D}_{\mathcal{F}}$ from $\varphi_{\mathcal{T}}$ and retaining the influence of $\mathcal{D}_{\mathcal{R}}$.

Fig. 4 shows results for unlearning algorithms such as Gradient Descent (GD) on $\mathcal{D}_{\mathcal{R}}$ from $\varphi_{\mathcal{T}}$, KL Minimization (KL) (Maini et al., 2024), Task Vector (TV) (Ilharco et al., 2023), SCRUB (Kurmanji et al., 2024), details of which are in App. E.6. It can be observed that they achieve aggregate `WaterDrum` values still far from that achieved by retraining: KL and TV can produce unlearned models that unlearn the forget set very well but cannot preserve the influence of the retain set much, the latter of which compromises their overall utility. GD and SCRUB can produce unlearned models that preserve some influence of the retain set but do not unlearn the forget set well. App. H.3 gives preliminary results for the cases with multiple data owners and different levels of data similarity.

## 5.2 IMPLEMENTING WATERDRUM WITH OTHER ADAPTED WATERMARKING METHODS

For the results above, we use the `Waterfall` framework in our implementation of `WaterDrum` because it satisfies our watermarking desiderata without excessive adaptation, as detailed in App. C.2. As suggested in Sec. 4, `WaterDrum` is designed to work with other watermarking methods that satisfy the watermarking desiderata. So, we will assess the performance of `WaterDrum` implemented with other adapted watermarking methods (see details in App. E.7) like KGW (Kirchenbauer et al., 2023), Synth-ID (Dathathri et al., 2024), and EXP-edit (Kuditipudi et al., 2024).

Table 4 shows results of `WaterDrum` implemented with `Waterfall`, the adapted KGW, Synth-ID, and EXP-edit in **D1** and **D2** as well as the average time needed to verify the watermark in a single text output of the LLM and whether GPU is needed for verification. It can be observed that the other adapted watermarking methods need either GPU or a few orders of magnitude more time than `Waterfall` to verify whether the LLM's text output contains the watermark. It is promising to see that `WaterDrum` with the adapted KGW performs very well in **D2** with an $R^2$ value close to 1. However, it performs poorly in **D1** as it does not satisfy **W2** by nature. This is because unlike `Waterfall`, KGW is designed to embed the watermark of a *single* model owner into the LLM's newly generated text outputs during inference time instead of embedding different watermarks of *multiple* data owners into their existing text data (App. A.3). Thus, the adapted KGW is suited for scenarios where satisfying **D2** is more important than **D1** (e.g., there is only one data owner, hence less of a need for **W2**) and it is possible to use significantly more verification time and GPU resources. The adapted Synth-ID and EXP-edit do not perform well in both **D1** and **D2** likely because they are not designed to satisfy **W1** and **W2**. In general, it is still better to use `Waterfall` given its better performance in both **D1** and **D2**.

## 6 CONCLUSION

Our work here has (a) defined clear desiderata that an effective and practical unlearning metric should satisfy to enable *direct interpretation* and *continuously* measure the *extent of unlearning* (Sec. 3), (b) proposed a novel data-centric LLM unlearning metric, `WaterDrum`, based on robust text watermarking that, unlike existing metrics, satisfies these desiderata (Secs. 4 and 5), and (c) introduced a new `WaterDrum-Ax` dataset to be used with `WaterDrum` to benchmark unlearning algorithms (Sec. 5.1). App. I discusses other questions (e.g., limitations) a reader may have.

## ACKNOWLEDGMENTS

This research is supported by the National Research Foundation, Singapore under its National Large Language Models Funding Initiative (AISG Award No: AISG-NMLP-2024-001). This research is supported by the National Research Foundation, Singapore under its AI Singapore Programme (AISG Award No: AISG3-RP-2022-029). Xinyuan Niu is supported by the Centre for Frontier AI Research of Agency for Science, Technology and Research (A*STAR). This research is supported by the National Research Foundation, Singapore under its AI Singapore Programme (AISG Award No: AISG2-PhD/2023-01-039J). This research is part of the programme DesCartes and is supported by the National Research Foundation, Prime Minister's Office, Singapore under its Campus for Research Excellence and Technological Enterprise (CREATE) programme. We would like to acknowledge that computational work involved in this research work is partially supported by NUS IT's Research Computing group under grant number NUSREC-HPC-00001.

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

# A    RELATED WORKS

## A.1    UNLEARNING METRICS

Unlearning algorithms are often evaluated based on their (a) unlearning effectiveness, (b) utility preservation, and (c) unlearning efficiency (Li et al., 2024a). We first briefly discuss (b) and (c). (b) Utility preservation refers to how well the LLM maintains its performance and usability after unlearning, and can be measured with performance indicators (e.g., perplexity, accuracy) on the retain set or various downstream tasks (Chang et al., 2024b). These performance indicators can include those used for (a) below, but evaluated on the retain set instead of the forget set. (c) Efficiency of an unlearning algorithm can be assessed based on how much time and resources it saves compared to retraining from scratch (Li et al., 2024a; Nguyen et al., 2022). See Sec. 4 of (Liu et al., 2025) for a deeper discussion about other unlearning effectiveness, utility preservation, efficiency, and scalability metrics. (c) is not the focus of this work. `WaterDrum` is designed to evaluate (a) unlearning effectiveness (Sec. 3) but may also evaluate (b) utility preservation on the retain set (Sec. 5.1).

**(a) Unlearning effectiveness metrics.** Broadly, unlearning effectiveness refers to how well the influence of the forget set is being removed from the LLM. There are a few classes of such metrics:

- **Utility-based metrics** are a form of utility-centric metrics that expect the model utility (performance indicators), when evaluated on the forget set, to worsen after unlearning. Utility-based LLM unlearning metrics include ROUGE-L (Lin, 2004), Truth Ratio (Maini et al., 2024), and KnowMem (Shi et al., 2025). Their definitions can be found in App. E.3 and the disadvantages of utility-centric metrics are already described in Sec. 2.

- **Membership inference attack (MIA)-based metrics** expect the ability or probability to infer the membership of a data point in the forget set to reduce significantly after unlearning. Some MIA-based metrics are also utility-centric as membership inference may depend on performance indicators such as perplexity and the log-likelihood of tokens in the text data (Shi et al., 2024). However, MIAs (Shokri et al., 2017) have demonstrated limited success against LLMs (Duan et al., 2024) and their performance is adversely affected by the presence of similar data in the forget and retain sets.

- **Watermark-based metrics** embed signals in the forget set and expect the values of these signals to decrease after unlearning (Li et al., 2024a). **Our `WaterDrum` falls under this class but is the first metric that can be applied to LLMs. Existing watermark-based unlearning metrics are designed to only work for image datasets and classification models.** For example, the work of Guo et al. (2023) has embedded invisible backdoors in images with incorrect target labels and the success of unlearning is measured by a drop in the success rate of backdoor attacks. The work of Sommer et al. (2022) has introduced a probabilistic verification framework for backdoors, in which users modified their data prior to submission. **We will highlight the key differences of our work here**: (i) These works rely on label-based predictions and face challenges such as generalization effects, conflicting backdoor patterns, or backdoor defences. In contrast, our work focuses on adapting watermarking to LLMs where longer and more complex output sequences provide richer signals for unlearning verification. (ii) In these works, the model utility is compromised even before unlearning, especially when the forget set is large. In contrast, our `WaterDrum` has minimal impact on the model utility because it is based on the robust watermarking framework called `Waterfall` that satisfies desideratum **W0**, as shown in App. G.1. (iii) Most importantly, existing watermark- and backdoor attack-based metrics are limited to image data and cannot be directly applied as unlearning metrics to text data due to additional challenges such as in preserving data fidelity (Guo et al., 2023; Sommer et al., 2022).

Unlearning metrics can also be classified based on whether they are **retraining-based** or **non-retraining-based**. Retraining is commonly viewed as the gold standard in classical unlearning settings (Bourtoule et al., 2021; Cao & Yang, 2015; Golatkar et al., 2020). This has led to various evaluation metrics that assert how closely an unlearned model approximates a retrained one, such as via matching performance on the forget set (Chundawat et al., 2023b; Golatkar et al., 2020) or measuring distances in weights and activations (Chundawat et al., 2023a; Golatkar et al., 2021; Tarun et al., 2023). However, retraining LLMs is often infeasible due to the scale of model parameters

and the volume of pretraining data. In addition, retraining-based metrics contradict the premise of unlearning that emphasizes the unavailability of a retrained LLM.

Therefore, non-retraining-based metrics are now more important and aligned with the growing trend of commercial LLMs that only provide black-box access. The work of Chundawat et al. (2023a) has proposed the ZRF score that captures the randomness in LLM predictions as an indicator of unlearning, while the work of Becker & Liebig (2022) has proposed to utilize a model's epistemic uncertainty. The work of Yao et al. (2024) has proposed that a surrogate subset with the same distribution as the forget set can be employed to approximate the performance of the retrained LLM. However, these metrics often **overlook the LLM's ability to generalize from pretraining or the remaining retain set**. To address this, synthetic datasets, such as the TOFU dataset (Maini et al., 2024), are carefully crafted to ensure a sufficient separation between the forget and retain sets. Nonetheless, **such a separation (i.e., a low level of data similarity) is rarely achievable in real-world scenarios. In this work, we address these limitations by proposing a non-retraining-based metric that works despite a greater level of data similarity between the forget and retain sets and the generalization ability of LLMs. Additionally, our metric would work for multiple unlearning requests.** Specifically, we propose to use watermarking (Guo et al., 2023; Sommer et al., 2022) to handle potential data similarities due to its ability to make each data point uniquely identifiable.

Note that our `WaterDrum` focuses on the unlearning scenario for data owners (Sec. 2), which is also known as "data influence removal" in (Liu et al., 2025). On the other hand, there exists another body of works in concept unlearning (i.e., also known as "knowledge unlearning" in (Jang et al., 2023; Si et al., 2023)) whose algorithms aim to remove certain knowledge from an LLM. The distinction between them is important when the forget set might contain concepts that are also present in the retain set, such as in our 'exact and semantic duplicate' settings. In Sec. 2, we have described our problem setting where even if one copy of the forget set is to be unlearned, the concepts should remain in the LLM when other copies with similar concepts exist in the retain set. This is in contrast to knowledge unlearning where concepts from the forget set are to be removed no matter whether they are also present in the retain set. In the latter case, unlearning metrics that directly assess the model's ability to answer those concepts would better measure the success of unlearning, e.g., KnowMem (Shi et al., 2025).

## A.2 EXISTING LLM UNLEARNING EVALUATION WORKS

The works of Maini et al. (2024); Shi et al. (2025) have proposed new unlearning metrics and benchmark datasets. The work of Li et al. (2024b) has proposed a multiple choice question benchmark dataset called WMDP to evaluate the LLM's knowledge in biosecurity, cybersecurity, and chemical security. This benchmark dataset is different from TOFU, MUSE, and ours in nature because it is specifically for knowledge editing and only contains test data instead of training data. The work of Wang et al. (2025) has suggested that an unlearning metric should be robust against (unchanged by) red teaming scenarios (such as recovering knowledge by jail-breaking, probing, relearning), and unlearning algorithms should be compared when they produce unlearned models with the same utility/performance on the retain set that is realized by mixing the parameters of the LLM before and after unlearning. The work of Wu et al. (2024) has proposed a new perspective of fact unlearning and an accompanying synthetic dataset. **In contrast, we propose a novel set of unlearning metric desiderata, which is satisfied by `WaterDrum`, to address realistic settings, such as when the forget and retain sets have semantically similar content and when retraining is impractical. Our desiderata are not intended to be exhaustive and can complement that of existing LLM unlearning evaluation works.** The work of Lynch et al. (2024) has proposed a suite of adversarial metrics to resurface forget set-related knowledge that exists in the unlearned LLMs, such as jailbreaking prompts, relearning (via fine-tuning and in-context learning), and latent knowledge extraction. While these metrics employ text similarity to the forget set in adversarial scenarios to evaluate the success of unlearning, watermarking uses the signal carried in the LLM's text outputs to detect the influence of the forget set.

## A.3 TEXT WATERMARKING FOR LLMS

Watermarking is an extensively studied technique for copyright protection, fingerprinting, and authentication (Liu et al., 2024; Wan et al., 2022). Watermarking consists of two main stages:

embedding and detection where the watermark must remain imperceptible and robust against removal attacks (Wan et al., 2022). Unlike digital images where continuous signals facilitate imperceptible watermark embedding, text watermarking is more difficult due to its discrete nature and susceptibility to text modifications (Liu et al., 2024).

Existing works on text watermarking for LLMs can be classified into two modes: **(1) watermarking for a model owner** and **(2) watermarking for data owners**. These modes differ based on who uses them (i.e., model owner vs. data owners), input (i.e., generic LLM query vs. existing source text data), and scalability (i.e., supporting a single model owner vs. multiple data owners). Most existing works fall under **(1) watermarking for a model owner** and its purpose is for the LLM owner and users of the LLM to distinguish if some text output is generated by the LLM. A *single* model owner can embed a watermark into the LLM's newly generated text outputs during inference time by changing the sampling distribution during the autoregressive generation process (Dathathri et al., 2024; Kirchenbauer et al., 2023; Kuditipudi et al., 2024), watermarking the LLM's weights (Li et al., 2023a), and post-hoc watermarking the LLM's generated text outputs (Chang et al., 2024a; Hao et al., 2025).

Our work aligns better with the other mode instead, i.e., **(2) watermarking for multiple data owners**. In this mode, data owners want to independently verify if their text data still influence the LLM. This involves *multiple* data owners watermarking their existing text data (e.g., news agencies watermarking their news articles) to trace their downstream usage, such as in LLM training. Therefore, the primary concerns of (2) differ from (1)—there is a need for stricter preservation of the semantic content of the text (i.e., the fidelity desideratum **W0**) and scalability to support a large number of data owners concurrently. (2) can be further categorized into (2a) non-LLM-based methods or (2b) LLM-based methods. Existing watermarking methods in (2a), such as inserting Unicode characters (Lu et al., 2025; Por et al., 2012; Sato et al., 2023) or synonym replacement (Qiang et al., 2023; Yang et al., 2022; Yoo et al., 2023), are often easily detectable and susceptible to word replacement (Lau et al., 2024). On the other hand, syntactic-based watermarking methods are often constrained by the limited choices of syntactic structures and require prior linguistic knowledge (Wan et al., 2022). Recently, LLMs have emerged as a promising watermarking tool as they can generate natural-looking text and improve watermarking robustness. The work of Lau et al. (2024) falls under (2b) by proposing a robust text watermarking framework called `Waterfall` that is capable of embedding watermarks across data from multiple data owners (while preserving the semantic content of the original text) and also achieving watermarking robustness such that watermarks in the training data for the LLMs remain detectable in the LLMs' text outputs. The watermarked training data are then used to train other LLMs. **Our work builds on top of our adaptation of the `Waterfall` framework (Lau et al., 2024) to develop our unlearning metric, i.e., `WaterDrum`. We have also considered adapting (1) watermarking methods for a model owner for mode (2) and in our implementation of `WaterDrum`, as discussed in Sec. 5.2.**

## B   FURTHER DISCUSSION ON UNLEARNING METRIC DESIDERATA (SEC. 3)

### B.1   SEPARABILITY DESIDERATUM **D1**

A separable unlearning metric (i.e., **D1**) should be a good classifier of whether an owner's data still influences an unlearned LLM, in particular, the retrained LLM $\varphi_{\mathcal{R}}$ (i.e., achieved by perfect unlearning) trained only on $\mathcal{D}_{\mathcal{R}}$. To illustrate the difference between a separable and non-separable metric, we provide a toy example in Fig. 5(left). With a separable metric, an optimal threshold $\kappa^*$ can be chosen where false positive and false negative classifications are minimal, as is the case for `WaterDrum` shown in Fig. 5 (top right). However, for non-separable metrics, any $\kappa$ chosen would result in similar true and false positive rates, as is the case for the utility-centric Truth Ratio metric shown in Fig. 5 (bottom right).

### B.2   CALIBRATION DESIDERATUM **D2**

Ideally, perfect unlearning will completely remove the influence of the forget set. However, in practice, imperfect unlearning may be inevitable due to the size and complexity of LLMs. This is because (a) perfect unlearning involving retraining from scratch is prohibitively expensive and impractical, and (b) perfect unlearning on LLMs is not yet achievable with the current approximate

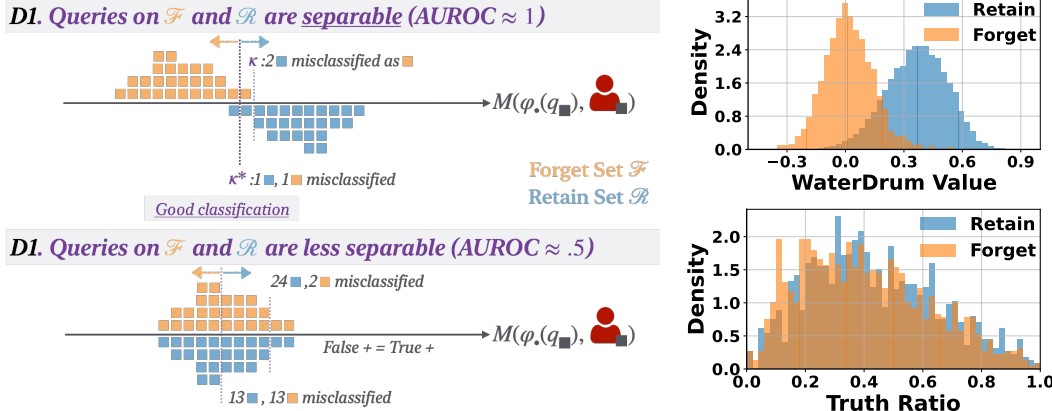

Figure 5: (Left) A toy example is provided to illustrate the intuition of the separability desiderata **D1**. Each ■ represents an LLM's text output to a query formed by a data point. Different $\kappa$'s correspond to different decision boundaries. In the top diagram, the metric and $\kappa^*$ can clearly separate the LLM's text outputs to queries formed by the forget set vs. the retain set. In the bottom diagram, there is no $\kappa$ that can clearly separate them and the true and false positive rates are always the same. (Right) Histograms of our `WaterDrum` vs. the utility-centric Truth Ratio metric values under the 'semantic duplicate' setting of data similarity for `WaterDrum-TOFU` dataset (Table 2 in Sec. 5) where `WaterDrum` exhibits a clear separability over the Truth Ratio metric; see the caption of Fig. 1 for a detailed description.

unlearning algorithms without significantly harming model utility/performance (e.g., on the retain set). In Sec. 5.1, we demonstrate that all unlearning algorithms only achieve imperfect unlearning, except when the LLM is destroyed (i.e., it is influenced by neither the forget nor the retain sets) or when a new LLM is retrained from scratch.

With the calibration desideratum **D2**, characterization of imperfect unlearning becomes possible. **D2** enables an unlearning metric to go *beyond being just a binary indicator* of whether the entire forget set has been unlearned to being a meaningful *continuous measure* of the extent to which a forget set $\mathcal{D}_{\mathcal{F}}$ has been unlearned:

- The proposed linear proportional form (i.e., Eq. (2)) of **D2** captures the goal that the unlearning metric can be interpreted on its own and indicates the proportion of $\mathcal{D}_{\mathcal{F}}$ that remains unlearned, while being given only a single calibration data point (i.e., the aggregate unlearning metric value of the forget set on the *original* LLM) available before unlearning. This contrasts with existing utility-centric metrics, which require another calibration data point (i.e., the aggregate unlearning metric value of the forget set on the *retrained* LLM) and hence violate **D3**(a), as discussed in Sec. 3.

- Our experiments (Fig. 3 and Table 3) show that `WaterDrum` can satisfy **D2**, enabling this intuitive interpretation when LLMs are retrained with the retain set $\mathcal{D}_{\mathcal{R}}$ and varying proportions of the forget set $k/|\mathcal{D}_{\mathcal{F}}|$.

Fig. 6 provides an intuitive illustration of the calibration desideratum where the metric measures the extent of imperfect unlearning. **D2** is practically useful in the following use cases:

1. Deployment: In practice, model owners may only achieve imperfect unlearning of the forget set to some extent while preserving the utility/performance of their LLM for customers. A calibrated continuous unlearning metric value satisfying **D2** can serve as an objective proxy for negotiations with data owners on the required extent of unlearning and corresponding monetary compensation. The negotiated target extent can then guide the actual implementation of unlearning, e.g., by selecting the most suitable unlearning algorithm (since different algorithms achieve different forget-retain performance trade-offs, as shown in Fig. 4) or guide the tuning of hyperparameters for a given algorithm.

2. Evaluation and development: For research and development, a calibrated metric satisfying **D2** enables evaluation beyond binary success/failure and instead continuously measures the extent of imperfect unlearning of the forget set. This supports a more realistic and fine-grained assessment of unlearning algorithms.

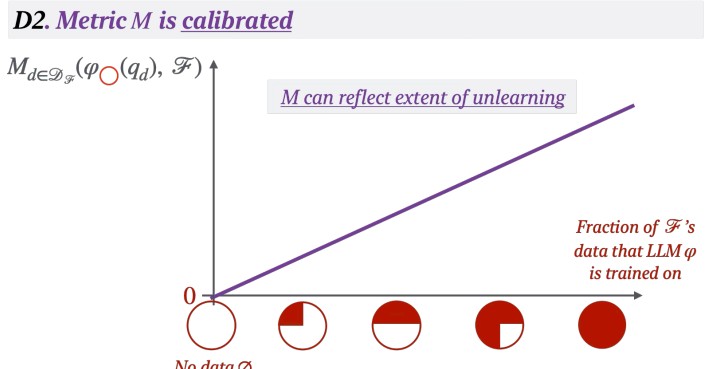

Figure 6: A calibrated metric should continuously measure the extent of imperfect unlearning. On the horizontal axis, we simulate using different sized proportions of the red owner's dataset. **D2** requires the metric to have a value of $0$ when the dataset is not used (y-intercept) and a larger value when a larger proportion is used. As shown in Fig. 3 in Sec. 5, `WaterDrum` is well-calibrated while other metrics are not.

### B.3 FEASIBILITY DESIDERATUM **D3**

In Sec. 2, we describe how utility-centric metric values, such as ROUGE-L, cannot be interpreted on their own. Instead, their interpretation requires referencing the aggregate metric value of the forget set on the retrained LLM. For example, we can compute the difference between the aggregate metric values on the unlearned vs. retrained LLMs. For a perfectly unlearned LLM (e.g., the retrained LLM), the difference should become $0$. Note that although the difference satisfies our calibration desideratum **D2**, it violates **D3**(a).

What happens when the aggregate metric value on the retrained LLM is unknown? Any aggregate metric value (e.g., ROUGE-L recall score) would not be informative; it is impossible for the model owner to know whether the aggregate metric value indicates perfect unlearning or only imperfect unlearning, or to know how far the unlearned LLM is from perfect unlearning. In Sec. 5, we do not reference the retrained LLM. The baseline metrics computed using the unlearned LLM only do not satisfy **D2**, as seen in Fig. 3.

### B.4 ROBUSTNESS TO SIMILAR DATA DESIDERATUM **D4**

In Secs. 2 and 3, we suggest that it is common for data owners to have semantically similar instances. Here, we provide concrete examples. Consider a real-life scenario where two news agencies, Reuters and The Straits Times (i.e., the data owners), produce semantically similar news articles, as shown in Fig. 7a. These two articles from different data owners exhibit a high semantic similarity with an STS score of $0.90$. In this case, only one agency may request unlearning. As another example in the `WaterDrum-Ax` dataset, Fig. 7b shows that the two arXiv paper abstracts from the same *Materials Science* category but different authors (i.e., the data owners) are also semantically similar with an STS score of $0.88$. In this example, only one group of authors may request unlearning.

## C FURTHER DISCUSSION ON WATERFALL

### C.1 OVERVIEW OF WATERFALL

`Waterfall` (Lau et al., 2024) embeds watermark signals in text by paraphrasing the text while preserving its original meaning. For example, "The cat caught the rat" can be watermarked and

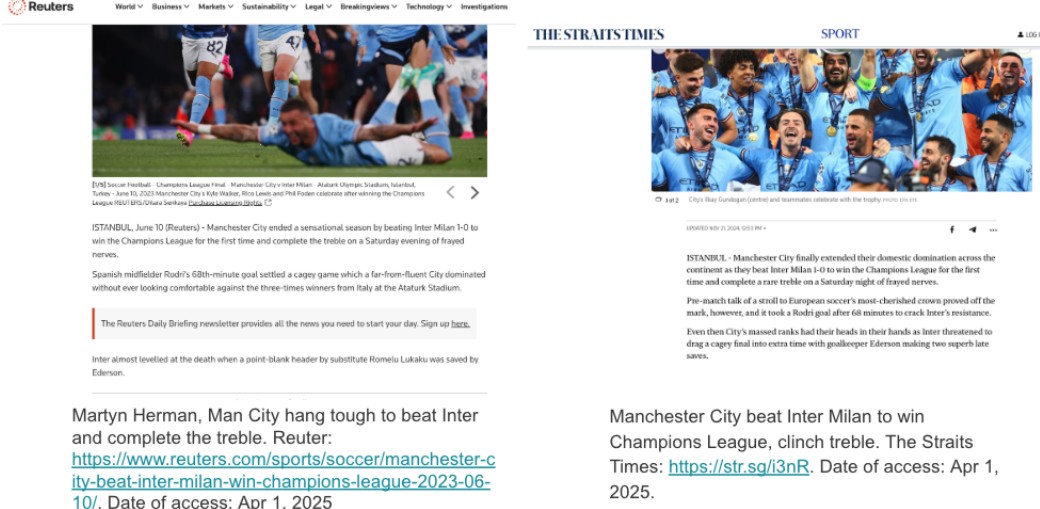

(a) The news agencies, Reuters and The Straits Times, both produce news articles reporting on the same soccer match and hence have a high semantic similarity with STS = 0.90.

**ABSTRACT**

We present the median surface brightness profiles of diffuse Ly$\alpha$ haloes (LAHs) around star-forming galaxies by stacking 155 spectro-scopically confirmed Ly$\alpha$ emitters (LAEs) at $3 < z < 4$ in the MUSE Extremely Deep Field (MXDF) with a median Ly$\alpha$ luminosity of $L_{Ly\alpha} \approx 10^{41.1}$ erg s$^{-1}$. After correcting for a systematic surface brightness offset we identified in the data cube, we detect extended Ly$\alpha$ emission out to a distance of $\approx$270 kpc. The median Ly$\alpha$ surface-brightness profile shows a power-law decrease in the inner 20 kpc and a possible flattening trend at a greater distance. This shape is similar for LAEs with different Ly$\alpha$ luminosities, but the normalisation of the surface-brightness profile increases with luminosity. At distances over 50 kpc, we observe a strong overlap of adjacent LAHs, and the Ly$\alpha$ surface brightness is dominated by the LAHs of nearby LAEs. We find no clear evidence of redshift evolution of the observed Ly$\alpha$ profiles when comparing with samples at $4 < z < 5$ and $5 < z < 6$. Our results are consistent with a scenario in which the inner 20 kpc of the LAH is powered by star formation in the central galaxy, while the LAH beyond a radius of 50 kpc is dominated by photons from surrounding galaxies.

**ABSTRACT**

The extended Ly$\alpha$ haloes (LAHs) have been found to be prevalent around high-redshift star-forming galaxies. However, the origin of the LAHs is still a subject of debate. Spatially resolved analysis of Ly$\alpha$ profiles provides an important diagnostic. We analyse the average spatial extent and spectral variation of the circumgalactic LAHs by stacking a sample of 155 Ly$\alpha$ emitters (LAEs) at redshift $3 < z < 4$ in the MUSE Extremely Deep Field. Our analysis reveals that, with respect to the Ly$\alpha$ line of the target LAE, the peak of the Ly$\alpha$ line at large distances becomes increasingly more blueshifted up to a projected distance of 60 kpc ($\approx 3\times$ virial radius), with a velocity offset of $\approx 250$ km/s. This trend is evident in both the mean and median stacks, suggesting that it is a general property of our LAE sample, which typically has a Ly$\alpha$ luminosity $\approx 10^{41.1}$ erg s$^{-1}$. However, due to the absence of systemic redshift data, it remains unclear whether the Ly$\alpha$ line peak at large projected distances is less redshifted compared to the inner regions or truly blueshifted with respect to the systemic velocity. We explore various scenarios to explain the large-scale kinematics of the Ly$\alpha$ line.

(b) In the `WaterDrum-Ax` dataset, the two arXiv paper abstracts from the same *Materials Science* category but different authors both present similar content and hence have a high semantic similarity with STS = 0.88.

Figure 7: Examples of high semantic text similarity (STS) in different domains.

paraphrased as "The rat was captured by the cat" while preserving the same meaning and not affecting downstream uses of the text (e.g., when used for LLM training). We will briefly describe the watermarking and verification operators below and refer the reader to the work of Lau et al. (2024) for more details. The watermarking and verification processes of `Waterfall` do not require the training of any LLMs.

**Watermarking Operator.** `Waterfall` uses an off-the-shelf LLM as a paraphraser, which preserves the text's original meaning, and embeds token-level watermark perturbation signals to the LLM's logits while generating the paraphrased text. Specifically, the token-level perturbations depend on the data owner's key (ID) and preceding tokens such that, on average, the perturbation is equivalent to adding random noise to the LLM's logits. Technically, at each token generation step, this involves (i) a permutation of the vocabulary space ordering based on the ID and preceding tokens, combined

with (ii) ID-dependent orthogonal perturbation functions that allow desirable properties such as the ability to add multiple watermarks in the same text. The watermarking operator is summarized in Algorithm 1 in (Lau et al., 2024).

**Verification Operator.**    With knowledge of the data owner's key (ID), it becomes possible to find the 'right' permutation to verify whether there has indeed been any signal embedded into the text or not. Technically, this involves simply accessing the correct vocabulary token space permutation. If a watermark has been embedded, doing a dot product in this permuted token space would yield a signal. Else, on average, only noise will be present, hence no signal will be detected. The verification operator is summarized in Algorithm 2 in (Lau et al., 2024). It does not involve any LLM inference or training and can be run on a CPU.

We discuss practical deployment details and computational cost of `Waterfall` in App. C.4.

## C.2  WATERFALL AND WATERMARKING DESIDERATA

`Waterfall` satisfies the watermarking desiderata required for `WaterDrum`, as stated in Sec. 4. Specifically,

- **W0 Fidelity**: `Waterfall` (Lau et al., 2024) is designed to satisfy a *fidelity* desideratum and ensure that the watermarked text is semantically similar to its unwatermarked version. Fidelity is ensured by using an LLM as a paraphraser and adding watermark signals to the LLM logits when generating tokens (such that, on average, the perturbation is equivalent to adding random noise).
  **Evidence.** App. H.3 in (Lau et al., 2024) shows that the LLMs fine-tuned using watermarked vs. unwatermarked datasets have minimal difference in *fidelity*. In App. G.1, we also empirically verify that the watermarking process has minimal impact on the LLM's performance (e.g., Truth Ratio).

- **W1 Verifiability**: (a) `Waterfall` would produce a high verification score if the watermarked text and the correct corresponding watermark key are inputs to the verification operator (Algorithm 2), and a score with an expected value of $0$ otherwise (e.g., if the wrong watermark key or unwatermarked text is used). Intuitively, this is because the verification score is the dot product of the watermark signal and the average cumulative token distribution, which is almost uniform noise without the right watermark key and watermarked text. (b) `Waterfall`'s verification score can be aggregated by taking the uniform average over all $d \in \mathcal{D}_{\mathcal{F}}$. When the LLM is trained on a larger subset of the forget set, more LLM outputs will contain the watermark, such that the aggregate metric value increases proportionally.
  **Evidence.** (a) Sec. $4.3$ in (Lau et al., 2024) has shown that `Waterfall` is verifiable in the LLM fine-tuned on watermarked text with AUROC of $1.0$ when evaluated on 100 queries of 100 generated tokens each. We empirically verify this with the `WaterDrum-TOFU` and `WaterDrum-Ax` datasets in Sec. 5 (under 'Separability desideratum **D1**'). (b) We further show empirically that the verification score of `Waterfall` is also proportional to the size of the subset of the forget set in Sec. 5 (under 'Calibration desideratum **D2**').

- **W2 Overlap verifiability**: `Waterfall` (Lau et al., 2024) is designed such that different watermark keys correspond to different permutations and perturbations of the logits in the LLM paraphraser. With pseudorandom permutations of the LLM logits and watermark signals (added to the LLM logits) that are defined with *orthogonal* functions, different watermarks are less likely to interfere with one another.
  **Evidence.** Sec. $4.3$ in (Lau et al., 2024) showed that `Waterfall` remains verifiable in an LLM when the training data has texts with up to 100 different watermarks. We empirically verify **W2** with the `WaterDrum-Ax` dataset in Sec. 5 (under 'Separability desideratum **D1**').

- **W3 Query access constraint**: Algorithm 2 in (Lau et al., 2024) for performing watermark verification only requires the suspected text (i.e., text output of the LLM), and does not require any other access to the LLM which generates the text.

- **W4 Unique key**: `Waterfall` (Lau et al., 2024) is designed to satisfy a *scalability* desideratum since it has a theoretical maximum of $10^{130274}$ unique watermark keys due to its

vocabulary permutations and orthogonal perturbations. This is in contrast to the maximum of $10^{10}$ for other text watermarking frameworks (Lau et al., 2024). This allows different data owners to have different unique keys, and it is extremely unlikely that different owners end up with the same key by random chance.

**Evidence.** App. E.8 in (Lau et al., 2024) empirically shows that the watermark is verifiable for up to $100,000$ random unique watermark keys.

Furthermore, `Waterfall` also satisfies the additional desiderata described in App. D:

- **W5 Privacy**: Algorithm 2 in (Lau et al., 2024) requires the private watermark key for watermark verification. As the watermark is embedded in the phrasing of the text, `Waterfall`'s watermark key cannot be directly or easily extracted from the watermarked text.

  **Evidence.** The work of Lau et al. (2024) has compared `Waterfall` with other text watermarking frameworks in Sec. 4.1 (Robust verifiability) and App. F.3. Other text watermarking frameworks (Lu et al., 2025; Qiang et al., 2023; Sato et al., 2023; Yoo et al., 2023), unlike `Waterfall`, have watermark keys that can be directly extracted from the text, causing them to be easily exploited and fail the following desideratum **W6**. We further discuss this in App. D.

- **W6 Resilience**: `Waterfall`'s robustness relies on its unique token-level embedding process such that completely removing its signal will require so many token changes in the text that would likely destroy its original meaning.

  **Evidence.** Sec. 4.1 (Robust verifiability) in (Lau et al., 2024) has evaluated `Waterfall` on a range of word-level and passage-level modifications, as well as additional rounds of text watermarking and LLM-aided attacks, and has shown that `Waterfall` remains verifiable after those attacks unlike other text watermarking frameworks. We also further demonstrate that `WaterDrum` with `Waterfall` is resilient to the model owner intercepting queries and attacks using watermarking methods for a model owner in Apps. D.1 and D.3, respectively.

In general, any watermarking method that satisfies our watermarking desiderata can be used within our `WaterDrum`. To the best of our knowledge, `Waterfall` is the only text watermarking method for data owners that satisfies our requirements as of now, and is thus used in our paper. However, as discussed in Sec. 5.2, it is possible to adapt other watermarking methods for a model owner to satisfy our requirements to some extent and we expect more watermarking methods to work with future developments in text watermarking.

### C.3 WATERMARKING OF DATASETS WITH WATERFALL

Watermarking and verification of the training text data have been performed with `Waterfall` (Lau et al., 2024) using the default configuration of the code available on https://github.com/aoi3142/Waterfall. When creating our `WaterDrum-Ax` and `WaterDrum-TOFU` datasets, the data is watermarked using the default LLM `meta-llama/Llama-3.1-8B-Instruct` with watermark strength $\kappa = 2^4$ and perturbation key $k_p = 1$. In App. G.3, we show that after LLMs are fine-tuned using text data watermarked with `Waterfall`, the watermark can be verified in the text outputs generated from these LLMs with low false positive and high true positive rates using just 59 generated tokens.

To create the watermarked `WaterDrum-Ax` and `WaterDrum-TOFU` datasets, we consider the scenario where each data owner $i \in \mathcal{T}$ with ID $i$ for $i = 0, 1, 2, \ldots, |\mathcal{T}| - 1$ has its own unique watermark key $\mu_i$. This ensures that each data owner is able to uniquely verify and evaluate the influence of its own data on the LLM when fine-tuned on its data and after any possible unlearning has been done on the fine-tuned LLM. For simplicity, we set $\mu_i$ to be $i$ when watermarking our datasets with `Waterfall`. For the experimental settings where duplicate data is considered (see Sec. 5, specifically, under 'Robustness to similar data **D4**'), each data owner $f \in \mathcal{F}$ whose data $\mathcal{D}_f$ is in the forget set $\mathcal{D}_\mathcal{F}$ and duplicated would have another data owner $j \in \mathcal{T}$ (where $j \neq f$) owning

---

[4]Note that this $\kappa$ is the watermark strength, as defined in `Waterfall` (Lau et al., 2024), and not the same as the separability threshold defined in the separability desideratum **D1**.

the duplicate of owner $f$'s data $\mathcal{D}_f$ and watermarking this duplicate with its watermark key $\mu_j$. The practical motivation for data owners with similar data is discussed in App. B.4.

To create the `WaterDrum-Ax` dataset, we consider $|\mathcal{T}| = 20$ unique data owners where each $i$-th category among the 20 categories of paper abstracts belongs to a single data owner $i$ and is watermarked with its watermark key $\mu_i = i$ for $i = 0, 1, \ldots, 19$. This emulates the setting where a model owner is aggregating data from 20 sources of academic publications where each source centers on a single academic discipline. To construct a forget set that consists of data from $n$ data owners, the data from the last $n$ of the $|\mathcal{T}|$ data owners are used as the forget set, while the data from the first $|\mathcal{T}| - n$ data owners forms the retain set. For instance, when unlearning the data from 1 data owner, the data from data owner $i = 19$ is the forget set. When unlearning the data from 5 data owners, the data from data owners $i = 15, 16, 17, 18, 19$ forms the forget set. For the experimental settings where duplicate data is considered as discussed above, the next data owner $j = (f + 1) \bmod |\mathcal{T}|$ owns the duplicate of the data $\mathcal{D}_f$ from the previous data owner $f$ and watermarks it with $\mu_j = j$ before including it into the augmented retain set. For licensing information on individual papers in the `WaterDrum-Ax` dataset, see https://arxiv.org/help/license.

To create the `WaterDrum-TOFU` dataset, we started off with the TOFU dataset (MIT License) and followed their construction in (Maini et al., 2024) by considering just two data owners 0 and 1 with the retain and forget sets, respectively. The retain set is watermarked with key $\mu_0 = 0$, while the forget set is watermarked with key $\mu_1 = 1$. For the experimental settings where duplicate data is considered as discussed above, data owner 0 owns the duplicate of the forget set from data owner 1 by watermarking it with $\mu_0 = 0$ before including it into the augmented retain set.

As a disclaimer, note that as part of `Waterfall`'s watermarking process, the original text data is paraphrased using an LLM. Although efforts have been made to ensure that the watermarked text retains a high semantic similarity with the original text (see the work of Lau et al. (2024) and `https://github.com/aoi3142/Waterfall`), we cannot guarantee the faithful reproduction of all content from the original text nor the factual correctness of the watermarked texts. In practical unlearning applications, additional (automated or manual) checks can be performed on the watermarked text to ensure accuracy and consistency to the original text as described in Lau et al. (2024), such as using Reflexion (Shinn et al., 2023). To reduce the computational cost, we have omitted these steps from our watermarking process. Despite this, similar to the fully fictitious TOFU dataset introduced by the work of Maini et al. (2024), `WaterDrum-Ax` and `WaterDrum-TOFU` still serve as suitable datasets when used for the purpose of evaluating unlearning metrics and algorithms where the factuality of the content in the dataset is not relied upon.

### C.4 PRACTICAL DEPLOYMENT PIPELINE FOR WATERDRUM WITH WATERFALL FOR LLM UNLEARNING EVALUATION

The watermarking process of `WaterDrum` is **lightweight** and incurs very little computational cost. This makes the watermarking process simple and **convenient for data and model owners during real-world deployment**:

- Data owners can quickly watermark their data before sharing them with model owners or releasing important data publicly. This not only facilitates unlearning verification but also allows them to detect whether their data has been used by model owners without authorization (Lau et al., 2024; Maini et al., 2024).

- No changes are required by the model owners who can continue training closed-source LLMs, provide API access, or release open-source models.

- Data owners can detect whether their data has been used for fine-tuning (even in closed-source LLMs) based only on the LLM's text outputs. After submitting an unlearning request, they can verify the extent of unlearning via `WaterDrum`.

- In comparison, other LLM unlearning metrics face severe deployment barriers, such as requiring to reference a retrained LLM (Secs. 1 and 2), which is infeasible even for cooperative model owners due to the computational cost.

The overhead of watermarking the training data is minimal compared to the cost of retraining the entire LLM. Watermarking the `WaterDrum-TOFU` dataset using an implementation of `Waterfall` with

the vLLM library (Kwon et al., 2023) on GPU takes only 10 seconds per 1000 data points and is performed only once when data is first contributed. In contrast, the cost of retraining the entire LLM is around 1h 30min, which is $100\times$ higher than the cost of watermarking the training data. Furthermore, retraining has to be repeated for every unlearning request.

In addition, our framework also reduces the verification costs. Verification of the `Waterfall`'s watermark is very efficient (Lau et al., 2024), requiring about 3 seconds per 1000 query outputs on CPU. In contrast, the computation of ROUGE using the rouge-score library[5] takes around 170 seconds per 1000 query outputs, which is two orders of magnitude slower.

A **limitation** is that `WaterDrum` requires watermarking the data before training and cannot be applied retroactively to data that has already been released. However, this practical concern will likely diminish with time and be mitigated due to the following reasons:

- Recalling already released data may be possible in our unlearning setting as data owners have the rights to their data and can control their use, as discussed in Secs. 1 and 2. Therefore, data owners of released data can still exercise the rights to their data by telling the model owner that they would (i) require their updated watermarked data to be used in the LLM instead, or (ii) consent to the continued use only if watermarking is to be part of the data processing step in the next LLM release. In either case, the model owner must comply with laws and regulations such as the GDPR and copyright laws.

- Even without recalling historical and released data, the data owners can demand that watermarks be applied going forward in future LLMs. They can expect newly generated data to be watermarked, hence facilitating future practical LLM unlearning evaluations. For example, news agencies can start watermarking their news articles and these recent articles may be more important in training future LLMs.

As awareness of privacy and security in LLM training grows, we expect proactive watermarking of data before release to become a common practice among data owners and the **applicability of `WaterDrum` to grow over time**.

A similar concept called image watermarking in the domain of computer vision has been widely studied and adopted for image data copyright protection (Cox et al., 2008). We note that when applications of image watermarking are proposed, such as data backdoors to verify unlearning (Thaker et al., 2024), they also face the same constraint that image watermarking can only be applied to unreleased data (and cannot be retroactively embedded in historical data). The community has accepted the constraint and appreciated the potential benefits going forward. Thus, there are strong reasons to believe in the potential for wider adoption of text watermarking and its applications.

## D    RESILIENCE DESIDERATUM

In Remark 2 in Sec. 4, we raise the following question: Is `WaterDrum` still an effective unlearning metric if (i) the model owner tries to reduce its LLM's aggregate metric value without directly performing unlearning or copyright its LLM via watermarking for a model owner (Kirchenbauer et al., 2023) or (ii) some data owner(s) try to report inflated aggregate metric values to understate the unlearning performed by the model owner?

In this section, we explain when the answer is yes. In particular, the underlying watermarking framework should additionally satisfy the following desiderata:

**W5  Privacy.** Each data owner $i$'s watermark key $\mu_i$ should be private and unknown to the model owner. Verification of the watermark with the verification operator should require the private watermark key. Moreover, the watermark key should not be easily extractable from the watermarked text. This prevents others without the watermark key (e.g., other data owners, model owner) from verifying whether some text contains the watermark and computing the metric value without owner $i$'s permission.

A private watermark may be required to support **W6**.

---

[5]https://pypi.org/project/rouge-score/.

**W6 Resilience.** The watermark should remain **W1** verifiable in the LLM's text outputs after attacks by the model owner. These attacks should have minimal impact on the semantic similarity of the LLM's text output and should not significantly affect the value of the LLM (as in **W0**) as the model owner is still interested in deploying a usable LLM.

Our adopted watermarking framework, `Waterfall`, satisfies **W5** as the verification operator requires the private key and the watermark key cannot be extracted from the watermarked text (Lau et al., 2024). In contrast, other watermarking frameworks do not satisfy **W5**. Frameworks that use invisible Unicode watermark key (Lu et al., 2025; Por et al., 2012; Sato et al., 2023) have their watermark keys plainly exposed in the watermarked text, while frameworks using synonym replacements (Qiang et al., 2023; Yang et al., 2022; Yoo et al., 2023) have watermarks that can be easily extracted from the watermarked text when the verification operator is known. The extraction enables the model owner to compute the metric value and remove or replace the watermark in the LLM's text outputs, which results in these other watermarking frameworks failing to satisfy **W6**.

We analyze whether `Waterfall` satisfies **W6** in this section. We consider attacks, which include (a) the model owner intercepting the LLM's text outputs based on a proxy indicator $SS$ such as semantic similarity with forget set (App. D.1), (b) watermarking for a model owner being applied to the LLM during its auto-regressive generation (App. D.3), and (c) other modifications being made to the LLM's text outputs after generation (App. D.3).

We also consider that under **W5**, data owner(s) can falsely try to report inflated aggregate metric values to understate the unlearning performed by the model owner in App. D.2.

### D.1 DECOY MODEL TO REDUCE AGGREGATE METRIC VALUE WITHOUT DIRECTLY PERFORMING UNLEARNING

Here, we consider the setting where the data owners requesting to erase their data from the LLM are unaware of or cannot control how the model owner performs unlearning. Instead, they can only evaluate the unlearning by querying the updated model. Moreover, the model owner's interests may not align with those of the data owners. As performing unlearning to produce the unlearned model $\widetilde{\varphi}$ can be more costly, the model owner may want to avoid performing unlearning while still appearing to fulfil the data owners' unlearning requests. The model owner can attempt to reduce the aggregate metric value by using a *decoy model* $\breve{\varphi}$ instead.[6]

**Decoy model.** The model owner implements the decoy model $\breve{\varphi}$, which involves using a gating function to intercept any query $q_d$ received. For any metric $M$ that the model owner can compute exactly, the model owner would intercept queries that result in large metric values, indicating that the original LLM $\varphi_\mathcal{T}$ is greatly influenced by the forget set $\mathcal{D}_\mathcal{F}$ (e.g., queries $q_d$ where $M(\varphi_\mathcal{T}(q_d), f) > \kappa$ for some $f \in \mathcal{F}$), and replace the LLM's text output $\varphi_\mathcal{T}(q_d)$ with some text $g(q_d, \mathcal{D}_\mathcal{F})$ that reduces the metric value. For metrics that the model owner cannot compute exactly (e.g., metrics that require some information that is private to the data owner), the model owner can only resort to a proxy indicator $SS$ that measures how similar the LLM's text output $\varphi_\mathcal{T}(q_d)$ is to the text from the forget set $\mathcal{D}_\mathcal{F}$. The decoy model is defined as follows:

$$\breve{\varphi}(q_d) = \begin{cases} g(q_d, \mathcal{D}_\mathcal{F}) & \text{if } \exists d_f \in \mathcal{D}_\mathcal{F} \ \ SS(\varphi_\mathcal{T}(q_d), d_f) > B \ , \\ \varphi_\mathcal{T}(q_d) & \text{otherwise} \ ; \end{cases} \tag{4}$$

with a threshold value $B$ determined by the model owner. Note that a small $B$ would intercept more queries and replace more of the LLM's text outputs. This may be more costly, reduce the overall LLM's performance, and may essentially be comparable to a full unlearning algorithm. *How would using the decoy model affect the effectiveness of various unlearning metrics?*

For metrics that do not depend on information private to the data owners (e.g., ROUGE-L), the model owner can compute them directly. So, the model owner can directly use the metric as $SS$, set the threshold $B$ to match the (learned) threshold $\kappa$ from **D1**, and optimize the replacement text $g(q_d, \mathcal{D}_\mathcal{F})$ that reduces the metric value. As $SS$ and $B$ can be more easily set, the model owner's decoy model

---

[6]Note that the model owner would have no incentive to do so when benchmarking different unlearning algorithms (Sec. 5.1). During benchmarking, the model owner (instead of the data owner) controls the training data and assesses the unlearning effectiveness. The model owner would not deceive itself about the performance of their unlearning algorithms.

attack will be more successful and less costly (from fewer interceptions). Thus, the metric is less resilient to the decoy model attack.

For metrics that depend on information private to the data owners (e.g., `WaterDrum` using `Waterfall` that satisfies **W5**), the model owner cannot compute them directly. Instead, the model owner can only define $SS$ based on some proxy indicator of similarity between the LLM's text output and data from the forget set $\mathcal{D}_{\mathcal{F}}$. The model owner would also have to tune $B$. A lower $B$ would reduce the aggregate `WaterDrum` value (giving the impression of unlearning) but come at a high cost of replacing more of the LLM's text outputs and hence poor model utility/performance. When using `Waterfall`, the model owner can only generate replacement text $g(q_d, \mathcal{D}_{\mathcal{F}})$ with lower metric values by using unwatermarked text from other sources (e.g., another LLM). This may further reduce the decoy model's performance. Thus, `WaterDrum` is more resilient to the decoy model attack.

**Empirical Evaluation of `WaterDrum`.** We prepare data $\overline{\mathcal{D}}$ to form a set $\mathcal{Q} := \{q_d\}_{d \in \overline{\mathcal{D}}}$ of queries that result in the aggregate `WaterDrum` value being above a threshold $\kappa$, i.e., $M'_{d \in \overline{\mathcal{D}}}(\varphi_{\mathcal{T}}(q_d), \mathcal{F}) > \kappa$. To prevent the model owner from recognizing and intercepting the queries easily, data $\overline{\mathcal{D}}$ are similar to but not directly based on the forget set $\mathcal{D}_{\mathcal{F}}$. For example, $\mathcal{D}_{\mathcal{F}}$ is a set of arXiv paper abstracts from the `math.PR` category, and $\overline{\mathcal{D}}$ consists of other such math paper abstracts not in $\mathcal{D}_{\mathcal{F}}$. The model owner can only use the proxy indicator of the STS score as $SS$ and must choose the threshold $B$. Fig. 8 plots the aggregate `WaterDrum` value against the percentage of $\mathcal{Q}$ being intercepted as the threshold $B$ decreases. As the model owner decreases $B$, it potentially reduces the aggregate `WaterDrum` value via two effects: (i) diluting the aggregate `WaterDrum` value by replacing the LLM's text output (with watermark signal) with text from unwatermarked data sources, and (ii) the remaining unintercepted LLM's text outputs are semantically more dissimilar to the original watermarked $\mathcal{D}_{\mathcal{F}}$. It can be observed that the aggregate `WaterDrum` value decreases almost linearly with the percentage of intercepted queries, implying that the model owner only relies on effect (i) with no help from effect (ii), i.e., the model owner can only reduce the aggregate `WaterDrum` value significantly by intercepting most queries whose resulting LLM's text outputs are semantically similar to $\mathcal{D}_{\mathcal{F}}$. This makes it very costly for the model owner to carry out the attack. For example, reducing the aggregate `WaterDrum` value to $0.2$ requires intercepting about $70\%$ of the queries in $Q$, so the model owner may favor performing actual unlearning instead.

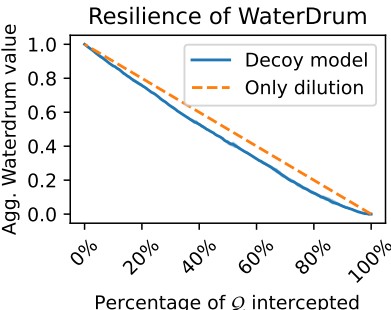

Figure 8: Plot of aggregate `WaterDrum` value of the forget set on the original LLM $\varphi_{\mathcal{T}}$ against the $\%$ of queries in $\mathcal{Q}$ being intercepted as the model owner decreases the threshold $B$ in the decoy model. An ideal unlearning metric would have its aggregate metric value decrease only proportionally with the $\%$ of intercepted queries (dashed orange diagonal line). `WaterDrum` achieves a similar performance, implying that the decoy model needs to intercept a large proportion of queries to reduce the aggregate `WaterDrum` value significantly. The aggregate `WaterDrum` values are scaled such that the value is $1.0$ when there is no intercepted query.

## D.2 DATA OWNER REPORTING INFLATED AGGREGATE WATERDRUM VALUES

When the watermarking framework satisfies **W5**, it is possible that a data owner refuses to acknowledge that unlearning has been done and reports an inaccurate, higher than measured aggregate `WaterDrum` value.

To resolve this issue, a trusted third party can certify the aggregate `WaterDrum` value reported by the data owner. To do so, the model owner provides access to the unlearned LLM $\widetilde{\varphi}$ and the data owner $i$ provides the watermark key $\mu_i$ to the trusted third party for verification. Verification should only be performed on a predefined set of queries with the agreement of both model and data owners. This prevents the model owner from repeatedly querying for the `WaterDrum` value to directly optimize the watermark removal instead of actual unlearning and from building a better decoy model in App. D.1. The trusted third party can then certify to the model owner that the measured aggregate `WaterDrum` value aligns with that reported by the data owner.

### D.3 DILUTION OF WATERDRUM'S WATERMARKS WITH WATERMARKING FOR A MODEL OWNER AND TEXT MODIFICATIONS

In certain scenarios, the model owner might want to watermark the newly generated text outputs of the fine-tuned LLM. This can either be for the legitimate purpose of copyrighting its LLM (i.e., being able to identify text outputs generated from its LLM) or in an adversarial attempt to dilute the watermarks in the training data. In this subsection, we empirically analyze whether `Waterfall` satisfies **W6** and `WaterDrum` is still an effective unlearning metric (i.e., satisfying the separability desideratum **D1** and calibration desideratum **D2**) under these scenarios. We consider the model owner (1) using watermarking and (2) perturbing the LLM's text output after output generation.

As discussed in App. A.3, *watermarking methods for a model owner* may embed a watermark into the LLM's newly generated text outputs during inference time by changing the sampling distribution during the autoregressive generation process. These methods can be further categorized into (i) during logits generation, such as (Kirchenbauer et al., 2023), or (ii) during token sampling, such as (Kuditipudi et al., 2024). `Waterfall`'s watermarking mechanism (when adapted as a watermarking method for a model owner) is similar to those in category (i), resulting in watermarks embedded in the LLM's logits distribution, and hence shares similar properties with other methods in category (i). For model owners to effectively remove category (i)'s watermarks without destroying text fidelity, they would need to apply the right distortions to cancel out the forget owner's watermark. However, since the forget owner's watermark key is private to the data owners (**W5**), it is practically impossible for the model owner to extract the key with limited samples from the forget owner. The model owner can only apply random watermark signals, which result in random distortions on average. `Waterfall` is designed to be robust to such random distortions, as we will show empirically below.

The work of Lau et al. (2024) has empirically shown that `Waterfall` is robust to various attacks, such as the watermark replacement attack (i.e., attack $\mathbb{A}3$ in (Lau et al., 2024)), where the original text watermark remains detectable even after the text has been watermarked again with a different text watermark. We also empirically evaluate the effectiveness of `WaterDrum` when the model owner uses various watermarking methods for a model owner.

First, we adapt the underlying watermarking operator of `Waterfall`'s text watermarking framework for data owners to serve as a watermarking method for a model owner. Our experiments are performed using LLMs fine-tuned on the `WaterDrum-Ax` or `WaterDrum-TOFU` dataset and applying a watermark strength of $\kappa = 2^7$ (i.e., same as that used during text watermarking (App. C.3)) with key $\mu_{20} = 20$, which differs from the data owners' watermark keys (i.e., $\mu_i = i$ for $i = 0, \ldots, 19$). We choose the same watermarking method as the data owner to perform watermarking for a model owner because using a different watermarking method reduces the likelihood that the model owner can exactly cancel out the watermark signal embedded by the data owner. The strongest attack by the model owner would be to apply the exact same watermark method with a 'destructive interference' signal that introduces perturbations opposite to those of the data owner. However, in practice, the model owner cannot do so with high probability given **W5**.

As the adaptation of `Waterfall` falls under category (i), we expect similar results from other watermarking methods for a model owner in this category. Nevertheless, we additionally experiment with a commonly used category (i) benchmark watermarking method for a model owner called KGW (Kirchenbauer et al., 2023) with the default watermark strength of $\delta = 2$ and green-list ratio of $\gamma = 0.5$. It can be observed from the results in Table 5 that the watermark signals are preserved post-attack. Note that due to differences in the underlying methodology of `Waterfall` and KGW,

---

[7]Note that this $\kappa$ is the watermark strength, as defined in `Waterfall` (Lau et al., 2024), and not the same as the separability threshold defined in the separability desideratum **D1**.

the watermark strength of $\kappa = 2^7$ in `Waterfall` and $\delta = 2$ in KGW are not directly comparable, and the performance difference cannot be solely attributed to whether one watermarking method for a model owner is "stronger" in affecting `WaterDrum` than the other. Additionally, while stronger watermarking for a model owner might potentially cause a larger drop in the text watermark's verifiability, the work of Kirchenbauer et al. (2023) has shown that strong watermarking for a model owner also negatively affects the quality of the LLM's newly generated text outputs. Thus, it is not in the model owner's interest to significantly degrade the LLM's performance simply to affect the data owners' ability to verify their watermarks.

As for category (ii) (i.e., also known as *distortion-free* watermarks), a seeded pseudo-random sampler is used in place of an unseeded random sampler during the generation of the LLM's text outputs (Kuditipudi et al., 2024). In this case, the text outputs are still generated from the token distribution of the underlying LLM and therefore practically indistinguishable from that of an unwatermarked LLM. This implies that the underlying token distribution, which contains `WaterDrum`'s watermarks, remains undistorted, hence preserving the watermark verifiability. In the experiments in our paper, no watermarking for a model owner is performed and we use the default unseeded random sampler and generate 10 text outputs to each LLM's query (App. E.2). As a result, we would not expect text outputs from distortion-free watermarking for a model owner to differ significantly from those produced by random sampling. Table 5 reports the results of inverse transform sampling (ITS) watermarking from (Kuditipudi et al., 2024) and again shows that the watermark signals are preserved post-attack.

In Table 5, the AUROC and $R^2$ decrease only slightly with the dilution from watermarking for a model owner. Thus, `WaterDrum` still preserves the separability desideratum **D1** and the calibration desideratum **D2** well.

Table 5: Resilience of `WaterDrum` when the model owner applies various watermarking methods for a model owner.

| Watermarking for a Model Owner | **D1** Separability (AUROC) | **D2** Calibration ($R^2$) |
|---|---|---|
| No dilution | 0.964 | 0.963 |
| Adapted `Waterfall` (Lau et al., 2024) | 0.923 | 0.936 |
| KGW (Kirchenbauer et al., 2023) | 0.944 | 0.948 |
| ITS (Kuditipudi et al., 2024) | 0.955 | 0.965 |

**Perturbing the LLM's text outputs after generation.** The model owner can use word level edits, such as insertion, deletion, and synonym substitution attacks, or passage level edits, such as paraphrasing and applying another text watermark to the LLM's newly generated text outputs. Note that such attacks are not very practical or realistic for the model owner: Word level edits can greatly affect the fidelity of the generated text (Lau et al., 2024), while passage level edits are computationally costly for the model owner, increase latency of the LLM's text outputs to queries, and would not support streaming of the LLM's text outputs. Nonetheless, `Waterfall` has been demonstrated to be robust to these attacks (Lau et al., 2024) and the same robustness would translate to `WaterDrum`'s watermarks.

All in all, this subsection shows that **`Waterfall` satisfies W6 and `WaterDrum` is thus resilient to attacks by the model owner.**

## E  DETAILS ON EXPERIMENTAL SETUP

We conduct our experiments on NVIDIA L40 and H100 GPUs. The experimental results are averaged across 3 random seeds 41, 42, and 43. Text generation from the different LLMs uses temperature = 1, top-p = 1, and top-k left as the LLM's vocabulary size. We use `sentence-transformers/all-mpnet-base-v2` as the STS model to evaluate STS scores. More details on our experimental setup are presented below.

### E.1 TRAINING HYPERPARAMETERS

**Models fine-tuned on `WaterDrum-Ax` dataset.** We fine-tune the bfloat16-pretrained Llama-2-7B model from Hugging Face[8] using LoRA (Hu et al., 2022) (based on $r = 8$ and $\alpha = 32$) with batch size of 128, 20 training epochs, and learning rate $10^{-3}$. Additionally, we fine-tune the bfloat16-pretrained Phi-1.5 model (detailed in App. F.3) under the same settings. Following the model choices in (Maini et al., 2024), we have considered these two models as they are representative of the recent LLMs and differ in terms of architectural details and scale.

**Models fine-tuned on `WaterDrum-TOFU` dataset.** We fine-tune the bfloat16-pretrained Llama-2-7B-chat model from Hugging Face[9] using LoRA (Hu et al., 2022) (based on $r = 8$ and $\alpha = 32$) with batch size of 128, 10 training epochs, and learning rate $10^{-4}$.

Subsequently, for unlearning, we use a batch size of 32. While we conduct our experiments using LoRA as in other LLM unlearning works (Maini et al., 2024; Shi et al., 2025), we also demonstrate in App. F.2 that `WaterDrum` applies to full parameter fine-tuning.

### E.2 QUERIES FOR FINE-TUNED LLM

For the `WaterDrum-Ax` dataset, we simulate a completion task by querying the LLM with the first 50 tokens of the training data for the LLM to complete the text. For the `WaterDrum-TOFU` dataset, we simulate a QA task by querying the LLM with the questions formatted according to the LLM's prompt format. We generate 10 text outputs to each LLM's query and calculate an unlearning metric by taking the mean of their resulting values. We generate up to a maximum of 200 tokens for each LLM's query.

### E.3 BASELINE UNLEARNING METRICS

- **ROUGE-L** measures the longest common subsequence between the LLM's text output and a reference text. This serves as a surrogate for the generation quality for the `WaterDrum-Ax` dataset and the question answering (QA) accuracy for the `WaterDrum-TOFU` dataset. To calculate the metric value, we follow the works of Maini et al. (2024); Shi et al. (2025) by computing the ROUGE-L recall scores (Lin, 2004) to compare the LLM's text outputs with the reference text in the training data. Specifically, we evaluate the ROUGE-L metric value when measured on the text output $\varphi_\bullet(q_d)$ to the query $q_d$ formed by each reference text data point $d$ and comparing $\varphi_\bullet(q_d)$ with $d$, i.e., $P(\varphi_\bullet(q_d), \mathcal{F}) = \text{ROUGE-L}(\varphi_\bullet(q_d), d)$.

- **Truth Ratio** measures the probability of generating a correct vs. wrong answer as an indicator of whether the LLM still memorizes the knowledge to be unlearned on the `WaterDrum-TOFU` dataset. Following the work of Maini et al. (2024), for each given question, the ratio is computed using the average probability over multiple wrong answers divided by the probability of a paraphrased correct answer.

- **KnowMem** measures the ROUGE-L recall scores between the LLM's text outputs vs. reference text answers to the questions in the QA pairs based on the training data, so as to reflect the LLM's memorization of the knowledge on the `WaterDrum-Ax` dataset. Specifically, following the work of Shi et al. (2025), we use GPT-4 to create a QA evaluation set with 8000 QA pairs based on the abstracts in the `WaterDrum-Ax` dataset and measure the ROUGE-L recall score between the LLM's text output to each question/query and the corresponding reference text answer.

- **MIA** measures the separability of the forget set that might still influence the LLM (e.g., due to imperfect unlearning (Sec. 2)) vs. the holdout set that does not (since it is not used to train the LLM). We use the holdout set given in the `WaterDrum-TOFU` dataset, and collect additional 2000 abstracts (100 for each subject category) as the full holdout set for the `WaterDrum-Ax` dataset. In our experiments in Sec. 5 (specifically, under 'Calibration desideratum **D2**'), to avoid imbalance, we randomly sample from the full holdout set to form holdout sets of different sizes that match the varying subset sizes $k$ of the forget set (App. E.5). To perform MIA on LLMs, we adopt the MIN-K% PROB attack with

---

[8] https://huggingface.co/meta-llama/Llama-2-7B-hf.
[9] https://huggingface.co/meta-llama/Llama-2-7B-chat-hf.

$\kappa = 40$ (Shi et al., 2024). Specifically, we use the AUROC to compare the MIN-$\kappa$% PROB values of the LLMs' text outputs to queries formed by the forget set vs. the holdout set where the MIN-$\kappa$% PROB value is defined as the average log-likelihood of the $40\%$ output tokens with the lowest probabilities. The AUROC of MIA on a perfectly unlearned LLM is close to $0.5$. To align our evaluation with other baselines (where lower aggregate metric values indicate better unlearning), we report the aggregate metric value for MIA as $M_{d \in \mathcal{D}_{\mathcal{F}}}(\varphi_\bullet(q_d), \mathcal{F}) := 1 - 2 \times \text{AUROC}$, which is close to $1$ for no unlearning and $0$ for perfect unlearning, the latter of which corresponds to the target AUROC of $0.5$.

- **WaterDrum** (ours) value is computed using the LLM's text output and excluding the query (i.e., without the first $50$ tokens for the completion task and without the prompt for the QA task). Note that in our experiments on the `WaterDrum-Ax` dataset, multiple watermarks are present due to multiple data owners watermarking their data with unique watermarks. Each data owner $i \in \mathcal{F}$ would send some query $q_d$ to the LLM formed by its watermarked data point $d \in \mathcal{D}_i$ and verify its watermark in the LLM's text output using its watermark key $\mu_i$.

With the exception of MIA, we use the uniform average over all text data points $d \in \mathcal{D}_{\mathcal{F}}$ to compute the aggregate metric value, i.e., $M_{d \in \mathcal{D}_{\mathcal{F}}}(\varphi_\bullet(q_d), \mathcal{F}) := \sum_{d \in \mathcal{D}_{\mathcal{F}}} M(\varphi_\bullet(q_d), \mathcal{F}) / |\mathcal{D}_{\mathcal{F}}|$ over all $d \in \mathcal{D}_{\mathcal{F}}$. The aggregate metric value for MIA is already defined above.

### E.4 DUPLICATE DATA

As discussed in Sec. 5 (specifically, under 'Robustness to similar data **D4**'), we examine three representative settings where there exists data $\mathcal{D}_s$ (i.e., injected into $\mathcal{D}_{\mathcal{R}}$) similar to $\mathcal{D}_{\mathcal{F}}$ but with different $SS$: (a) **Exact duplicate**: data points in $\mathcal{D}_s$ are exact copies of those in $\mathcal{D}_{\mathcal{F}}$ (i.e., $\mathcal{D}_s = \mathcal{D}_{\mathcal{F}}$). This marks the highest similarity with mean STS $= 1.00$ and ROUGE-L $= 1.00$. (b) **Semantic duplicate**: data points in $\mathcal{D}_s$ are paraphrased versions of those in $\mathcal{D}_{\mathcal{F}}$ with similar semantic meaning (i.e., $\mathcal{D}_s \simeq \mathcal{D}_{\mathcal{F}}$). We use GPT-4 to paraphrase all text data points in $\mathcal{D}_{\mathcal{F}}$ to obtain $\mathcal{D}_s$. In this setting, $\mathcal{D}_s$ has mean STS $= 0.97$ and ROUGE-L $= 0.69$ for the `WaterDrum-Ax` dataset, and mean STS $= 0.96$ and ROUGE-L $= 0.60$ for the `WaterDrum-TOFU` dataset. While the 'exact duplicate' setting can be less common in reality, it clearly illustrates and clarifies the limitations of different unlearning metrics. The 'semantic duplicate' setting happens and is a better measure of realistic unlearning capabilities (see real-world examples in App. B.4). We also consider the conventional setting where (c) **no duplicate** of any data point in $\mathcal{D}_{\mathcal{F}}$ is used to augment $\mathcal{D}_{\mathcal{R}}$ (i.e., $\mathcal{D}_s = \emptyset$). Specific watermarking details on the duplicate data are described in App. C.3. For each setting, the LLM is fine-tuned on $\mathcal{D}_{\mathcal{R}}^s = \mathcal{D}_s \bigcup \mathcal{D}_{\mathcal{R}}$. During unlearning, we aim to remove the influence of $\mathcal{D}_{\mathcal{F}}$ while retaining the influence of $\mathcal{D}_{\mathcal{R}}^s$.

### E.5 CALIBRATION DESIDERATUM **D2**

In our experiments in Sec. 5 (specifically, under 'Calibration desideratum **D2**'), we simulate varying sizes $k$ of subsets $\mathcal{D}_\ominus$ of the forget set $\mathcal{D}_{\mathcal{F}}$ by partitioning $\mathcal{D}_{\mathcal{F}}$ sequentially into 10 partitions, and incrementally include the partitions (together with the retain set $\mathcal{D}_{\mathcal{R}}$) into the training data for retraining the LLMs, i.e., using the first $0\%, 10\%, 20\%, \dots, 100\%$ of $\mathcal{D}_{\mathcal{F}}$ as $\mathcal{D}_\ominus$ for retraining the LLMs on $\mathcal{D}_{\mathcal{R}} \bigcup \mathcal{D}_\ominus$. It can be observed from Fig. 3 and Table 3 that `WaterDrum` satisfies the calibration desideratum **D2** under this way of partitioning, and we think it would generally be satisfied in expectation over randomly sampled subsets (with a fixed size of $k$) of $\mathcal{D}_{\mathcal{F}}$, as empirically verified in App. F.1.

### E.6 BENCHMARKING UNLEARNING ALGORITHMS

In Sec. 5.1, we consider the setting of a model owner comparing the aggregate `WaterDrum` values on different unlearned LLMs $\widetilde{\varphi}$ resulting from different unlearning algorithms. Such a setting can arise in a practical scenario where the model owner evaluates the performance of different unlearning algorithms on a small dataset before selecting the algorithm to fulfil data owners' unlearning requests after deployment. The model owner can also be a researcher developing new unlearning algorithms. Under this setting, we assume that the training data, LLM, and watermark keys are all under the full control of the model owner. The model owner can use watermarked data such as the `WaterDrum-Ax`

or `WaterDrum-TOFU` dataset with known watermark keys and can compute the metric values of the retain and forget sets directly without the restrictions discussed in App. D. A perfect unlearning algorithm would produce (i) a high aggregate metric value of the retain set $\mathcal{D}_{\mathcal{R}}$ on the unlearned LLM $\widetilde{\varphi}$ (equivalently, the retrained LLM $\varphi_{\mathcal{R}}$) that is ideally close to that on the original LLM $\varphi_{\mathcal{T}}$ (i.e., no unlearning), i.e., $M_{d \in \mathcal{D}_{\mathcal{R}}}(\widetilde{\varphi}(q_d), \mathcal{R}) \approx M_{d \in \mathcal{D}_{\mathcal{R}}}(\varphi_{\mathcal{T}}(q_d), \mathcal{R})$ since the retain set still influences both the retrained and original LLMs, and (ii) a low aggregate metric value of the forget set $\mathcal{D}_{\mathcal{F}}$ on the unlearned LLM $\widetilde{\varphi}$, i.e., $M_{d \in \mathcal{D}_{\mathcal{F}}}(\widetilde{\varphi}(q_d), \mathcal{F}) \approx 0$ since the forget set does not influence the retrained LLM; retraining (i.e., perfect unlearning) is at the bottom right corner in Fig. 4.

In our experiments, we have benchmarked the following popular baseline unlearning algorithms:

- **Retraining**: The base LLM is fine-tuned only on the retain set $\mathcal{D}_{\mathcal{R}}$ for the same number of epochs as when fine-tuning the original LLM $\varphi_{\mathcal{T}}$ to obtain the retrained LLM $\varphi_{\mathcal{R}}$. The retrained LLM usually serves as the gold standard for other unlearning algorithms to produce their unlearned LLMs.

- **Gradient Descent (GD)**: The original LLM $\varphi_{\mathcal{T}}$ is fine-tuned using gradient descent on the retain set $\mathcal{D}_{\mathcal{R}}$ for 1 or several epochs. GD assumes that the influence of the forget set is naturally removed from the LLM as training progresses on the retain set. In our experiments, we fine-tune for 1 epoch.

- **KL Minimization (KL)** (Maini et al., 2024): The original LLM $\varphi_{\mathcal{T}}$ is updated by simultaneously maximizing the predictive loss on the forget set and minimizing the Kullback-Leibler (KL) distance between the predictive distributions/likelihoods of the unlearned vs. original LLMs' text outputs to queries formed by the retain set for 5 unlearning epochs.

- **SCRUB** (Kurmanji et al., 2024): The original LLM $\varphi_{\mathcal{T}}$ is updated by maximizing the Kullback-Leibler distance between the predictive distributions/likelihoods of the unlearned vs. original LLMs' text outputs to queries formed by the forget set, while minimizing the predictive loss and KL distance on the retain set. The optimization process alternates between the maximization steps and minimization steps. In our experiments, we run 3 maximization and minimization epochs.

- **Direct Preference Optimization (DPO)** (Maini et al., 2024): For QA tasks, the original LLM $\varphi_{\mathcal{T}}$ is updated to encourage responses such as "I don't know" on the forget set, while simultaneously minimizing the predictive loss on the retain set. Note that DPO is not compatible with completion tasks and is omitted from benchmarking for the `WaterDrum-Ax` dataset. We run 5 unlearning epochs for DPO.

- **Task Vector (TV)** (Ilharco et al., 2023): We follow the implementation in (Shi et al., 2025). Firstly, the original LLM $\varphi_{\mathcal{T}}$ is further fine-tuned on the forget set to obtain a reinforced LLM $\varphi_{\text{reinforce}}$. Next, we take the difference in parameters by subtracting the parameters of the $\varphi_{\mathcal{T}}$ from that of $\varphi_{\text{reinforce}}$. Lastly, the unlearned LLM is obtained by subtracting this difference from the parameters of $\varphi_{\mathcal{T}}$. In the experiments, $\varphi_{\text{reinforce}}$ is fine-tuned from $\varphi_{\mathcal{T}}$ on the forget set for 5 epochs.

In the benchmarking of unlearning algorithms, we exclude Gradient Ascent on the forget set from the original LLM (Maini et al., 2024) as it has been shown to perform poorly in the other works where the LLM's text outputs become gibberish or random words (Maini et al., 2024).

### E.7 IMPLEMENTING WATERDRUM WITH OTHER ADAPTED WATERMARKING METHODS FOR A MODEL OWNER

As discussed in Sec. 5.2, to assess the performance of `WaterDrum` implemented with other adapted watermarking methods for a model owner, we implement `WaterDrum` with the adapted KGW (Kirchenbauer et al., 2023), Synth-ID (Dathathri et al., 2024), and EXP-edit (Kuditipudi et al., 2024), which are popular **watermarking methods for a model owner** that have been described in App. A.3. However, based on just the design and experimental support of these methods, it is not clear that all of the watermarking desiderata would be met, as summarized in Table 6. Additionally, we still need further adaptations to them and additional experiments to establish that the watermarking desiderata are met:

Table 6: Comparison of `Waterfall` vs. other watermarking methods for a model owner based on whether they satisfy our proposed watermarking desiderata in Sec. 4. We use ^ to denote that the assessment is off-the-shelf as it requires nontrivial modification to their code to support the desideratum. We use * to denote that the assessment is based on our results in Table 4 after adaptation of the methods. Future work may nontrivially improve the design of the various watermarking methods to better satisfy the desiderata than what our current results offer.

| Watermarking Method | W0 | W1 | W2 | W3 | W4 |
|---|---|---|---|---|---|
| `Waterfall` (Lau et al., 2024) | Yes | Yes | Yes | Yes | Yes |
| KGW (Kirchenbauer et al., 2023) | No^ | Yes* | Not shown | Yes | No^ |
| Synth-ID (Dathathri et al., 2024) | No^ | No* | Not shown | Yes | No^ |
| EXP-edit (Kuditipudi et al., 2024) | No^ | No* | Not shown | Yes | Yes |

- For example, to satisfy **W4**, the watermarking methods for data owners that embed watermarks into training text data need to support insertion and verification of watermarks from *different/multiple* data owners. In contrast, watermarking methods for a model owner are designed to embed the watermark of the *single* owner into the LLM's newly generated text outputs.

- The requirements of **W2** are also not explored, but it is an important requirement for these watermarking methods for a model owner to be used in our implementation of `WaterDrum`. On the other hand, Sec. 4.3 in (Lau et al., 2024) has established this property of being able to recover *multiple different* watermark signals in the text outputs generated by LLMs trained on text data watermarked by *many data owners* in their experiments, giving us more confidence in using `Waterfall` for demonstrating the effectiveness of `WaterDrum`.

Nonetheless, we attempt to adapt KGW, Synth-ID, and EXP-edit to satisfy the watermarking desiderata. Specifically, we first add a paraphrasing prompt following the approach in `Waterfall`. As KGW and Synth-ID are designed for watermarking (of the LLM's newly generated text outputs) for a single model owner, their codebases have explicitly fixed single watermark IDs into the internal components of the source code—we have to modify it for them to accept different watermarks and support multiple data owners. We use the same Llama-3.1-8B-Instruct model to apply text watermarking on the original arXiv dataset, except for Synth-ID, for which we use Gemma-7b-it as the provided codebase only supports Gemma and GPT-2 models. The results have been shown in Table 4 in Sec. 5.2.

# F ABLATION STUDIES

## F.1 CALIBRATION DESIDERATUM **D2**

In Eq. (2), we define the calibration desideratum **D2** as the expectation of the aggregate metric value over random subsets $\mathcal{D}_\bullet \subseteq \mathcal{D}_\mathcal{F}$ (with a fixed size of $|\mathcal{D}_\bullet| = k$) to be proportional to $k$. To verify that `WaterDrum` satisfies this relationship, we perform an experiment by randomly sampling three subsets (with a fixed size of $k$) of $\mathcal{D}_\mathcal{F}$ to yield a total of four different subsets (including the subset obtained by sequential partitioning, as detailed in App. E.5) for every proportion $k/|\mathcal{D}_\mathcal{F}| = 0\%, 20\%, 40\%, 60\%, 80\%, 100\%$. Fig. 9 shows the calibration curve for `WaterDrum` (i.e., mean aggregate `WaterDrum` value) enclosed by the minimum and maximum aggregate `WaterDrum` values, as well as the corresponding best-fit line under the 'no duplicate' setting for the `WaterDrum-Ax` dataset. It can be observed that the linear proportional relationship holds with the mean aggregate `WaterDrum` values achieving a high $R^2$ value of $0.960$ (i.e., close to the value of $0.963$ reported in Table 3). Moreover, each subset has an aggregate `WaterDrum` value that is close to the mean of the four subsets, i.e., there is a narrow range between the minimum and maximum aggregate `WaterDrum` values.

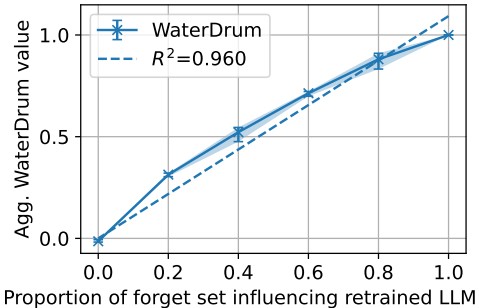

Figure 9: Calibration curve for `WaterDrum` (i.e., mean aggregate `WaterDrum` value) w.r.t. proportion $k/|\mathcal{D}_{\mathcal{F}}|$ of forget set influencing retrained LLM (solid) and its best-fit line ($R^2 = 0.96$) through origin (dotted) under 'no duplicate' setting for `WaterDrum-Ax` dataset. Each $\times$ marker indicates the mean aggregate `WaterDrum` value over four subsets of the same size $k$ and is enclosed by error bars indicating the minimum and maximum aggregate `WaterDrum` values.

### F.2 FULL PARAMETER FINE-TUNING

The experiments in Sec. 5 were conducted using LoRA (Hu et al., 2022), following the setting in the other LLM unlearning works (Maini et al., 2024; Shi et al., 2025). To show that `WaterDrum` is also applicable when used for full parameter fine-tuning, we conduct experiments for the separability desideratum **D1** and calibration desideratum **D2** under different levels of data similarity for the `WaterDrum-Ax` dataset.

For full parameter fine-tuning, we use a learning rate of $10^{-4}$ and train for 10 epochs. Note that due to the high computational cost of full parameter fine-tuning, we only report the results for one seed, while the results for LoRA are averaged across three different seeds.

Tables 7 and 8 show that `WaterDrum` performs better than the other unlearning metrics and better satisfies **D1** and **D2** for both LoRA and full parameter fine-tuning.

Table 7: AUROC of various unlearning metrics under different levels of data similarity for the `WaterDrum-Ax` dataset. `WaterDrum`'s AUROC remains near 1.0 even when similar data exists.

| Data Similarity | | ROUGE | KnowMem | WaterDrum |
|---|---|---|---|---|
| Exact Duplicate | Full | 0.335 | 0.497 | 0.990 |
| | LoRA | 0.334 | 0.492 | 0.957 |
| Semantic Duplicate | Full | 0.965 | 0.447 | 0.990 |
| | LoRA | 0.960 | 0.450 | 0.963 |
| No Duplicate | Full | 0.984 | 0.481 | 0.991 |
| | LoRA | 0.974 | 0.491 | 0.965 |

### F.3 OTHER LLMS

We have also evaluated the performance of `WaterDrum` using the Phi-1.5 model.[10] Figs. 10a and 10b show, respectively, the ROC and calibration curves for `WaterDrum` under the 'no and exact duplicate' settings for the `WaterDrum-Ax` dataset. The high AUROC and $R^2$ values achieved by `WaterDrum` agree with our main empirical findings in Sec. 5 using the Llama2-7B model and show that `WaterDrum` satisfies our proposed effectiveness desiderata **D1**-**D2**. This validates our `WaterDrum`'s adaptability to different LLMs, which increases its real-world applicability.

Beyond the Llama2-7B model evaluated in our main experiments, we further investigate the performance of `WaterDrum` using a larger Llama2-13B model. Figs. 11a and 11b show,

---

[10]https://huggingface.co/microsoft/phi-1_5.

Table 8: $R^2$ values for the best-fit lines of various unlearning metrics under different levels of data similarity for `WaterDrum-Ax` dataset. `WaterDrum` achieves the highest $R^2$ values that are closest to 1 and is hence a well-calibrated metric.

| Data Similarity | | ROUGE | KnowMem | MIA | WaterDrum |
|---|---|---|---|---|---|
| Exact Duplicate | Full | -5059 | -981.5 | -4.774 | 0.984 |
| | LoRA | -37.47 | -498.1 | -285.6 | 0.987 |
| Semantic Duplicate | Full | 0.545 | -139.2 | -35.57 | 0.989 |
| | LoRA | 0.693 | -276.5 | -14.52 | 0.991 |
| No Duplicate | Full | 0.850 | -103.8 | -3.937 | 0.940 |
| | LoRA | 0.650 | -252.9 | 0.677 | 0.963 |

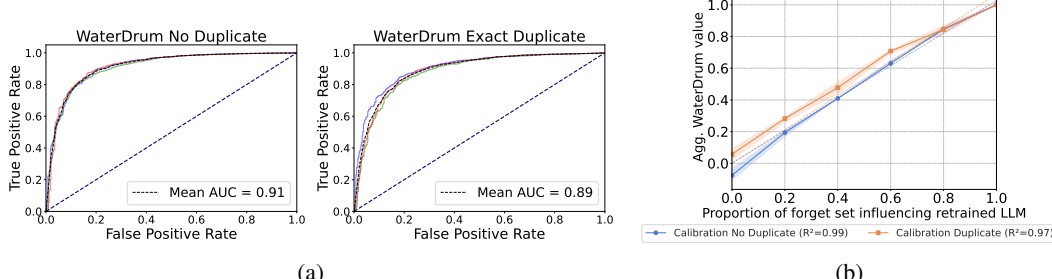

(a)                                                                    (b)

Figure 10: (a) ROC curves and (b) calibration curves for `WaterDrum` w.r.t. proportion $k/|\mathcal{D}_{\mathcal{F}}|$ of forget set influencing the retrained Phi-1.5 model (solid) and their best-fit lines through origin (dotted) under 'no and exact duplicate' settings for `WaterDrum-Ax` dataset. (a) `WaterDrum` achieves good separability due to high AUROC values and hence satisfies **D1**. (b) `WaterDrum` is well-calibrated due to high $R^2$ values for both the 'no duplicate' setting ($R^2 = 0.99$) and the 'exact duplicate' setting ($R^2 = 0.97$) and satisfies **D2** with its best-fit lines closely following its aggregate metric values.

respectively, the ROC and calibration curves for `WaterDrum` under the 'no duplicate' setting for the `WaterDrum-TOFU` dataset. The even higher AUROC and $R^2$ values achieved by `WaterDrum` demonstrate that it satisfies our proposed effectiveness desiderata **D1-D2** to a larger extent on larger LLMs. This further validates `WaterDrum`'s adaptability to different LLM sizes, which increases its real-world applicability.

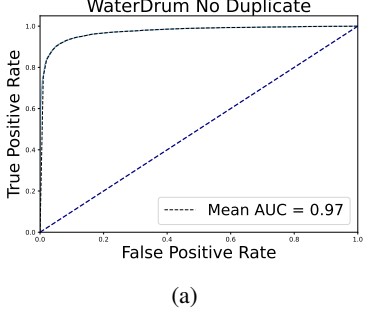
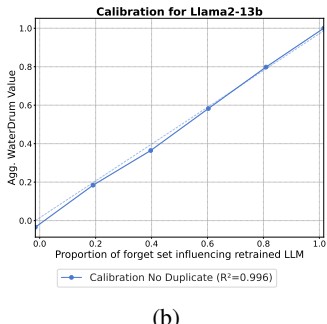

(a)                                                                    (b)

Figure 11: (a) ROC curve and (b) calibration curve for `WaterDrum` w.r.t. proportion $k/|\mathcal{D}_{\mathcal{F}}|$ of forget set influencing the retrained Llama2-13B model (solid) and its best-fit line through origin (dotted) under 'no duplicate' setting for `WaterDrum-Ax` dataset. (a) `WaterDrum` achieves good separability due to a high AUROC value and hence satisfies **D1**. (b) `WaterDrum` is well-calibrated due to a high $R^2$ value for the 'no duplicate' setting ($R^2 = 0.996$) and satisfies **D2** with its best-fit line closely following its aggregate metric values.

## F.4 EVALUATING BASELINE UNLEARNING METRICS ON LLMS FINE-TUNED ON WATERMARKED DATA

We have shown in Sec. 5 that the baseline unlearning metrics perform poorly when similar data exists in both the retain and forget sets. This is largely due to the baseline metrics being unable to ensure a sufficient separability between the similar copies/versions of data in both the retain and forget sets. These baseline metrics are evaluated on LLMs that are fine-tuned on the unwatermarked versions $\mathcal{D}_\mathcal{T}$ in the `WaterDrum-Ax` and `WaterDrum-TOFU` datasets. In this subsection, to study the effects of watermarking on these baseline metrics, we evaluate them by fine-tuning LLMs on the watermarked versions $\mathcal{D}'_\mathcal{T}$ in the `WaterDrum-Ax` and `WaterDrum-TOFU` datasets instead.

The watermarking process **P1** contributes to performance gains in the separability desideratum **D1** (i.e., AUROC) for the baseline unlearning metrics such as ROUGE as well, especially for the 'exact and semantic duplicate' settings, as it makes data less similar by embedding different watermarks unique to each data owner. ROUGE exhibits some improvement in the separability desideratum **D1** for the 'exact and semantic duplicate' settings due to the de-duplication done with watermarking (Table 9). However, **D1** alone is not sufficient as the threshold $\kappa$ for separating the LLM's text outputs to queries formed by the forget set vs. the retain set would be unknown in practice. **D2** is needed. It can be observed from Table 10 that unlike `WaterDrum`, the baseline unlearning metrics still achieve low $R^2$ values and hence fail to satisfy the calibration desideratum **D2**. In particular, the calibration curves for the baseline unlearning metrics do not pass through the origin, so they cannot be used to determine when $\mathcal{D}_\mathcal{F}$ has been perfectly unlearned.

Table 9: AUROC ($\pm$ across 3 seeds) of various unlearning metrics evaluated on LLMs fine-tuned on watermarked text data under different levels of data similarity for the `WaterDrum-TOFU` and `WaterDrum-Ax` datasets. The AUROCs of both `WaterDrum` and ROUGE remain near 1.0 even when similar data exists.

| | WaterDrum-TOFU | | | WaterDrum-Ax | | |
|---|---|---|---|---|---|---|
| Data Similarity | ROUGE | Truth Ratio | WaterDrum | ROUGE | KnowMem | WaterDrum |
| Exact Duplicate | **0.926±0.051** | 0.509±0.002 | **0.926±0.027** | **0.979±0.004** | 0.444±0.007 | **0.957±0.008** |
| Semantic Duplicate | **0.977±0.001** | 0.515±0.003 | **0.954±0.001** | **0.979±0.000** | 0.466±0.008 | **0.963±0.001** |
| No Duplicate | **0.980±0.005** | 0.727±0.000 | **0.928±0.026** | **0.983±0.000** | 0.474±0.003 | **0.965±0.002** |

Table 10: $R^2$ values of various unlearning metrics evaluated on LLMs fine-tuned on watermarked text data under different levels of data similarity for the `WaterDrum-TOFU` and `WaterDrum-Ax` datasets. Only the $R^2$ value of `WaterDrum` remains near 1.0 even when similar data exists.

| | WaterDrum-TOFU | | | | WaterDrum-Ax | | | |
|---|---|---|---|---|---|---|---|---|
| Data Similarity | ROUGE | Truth Ratio | MIA | WaterDrum | ROUGE | KnowMem | MIA | WaterDrum |
| Exact Duplicate | -7.624 | -261.2 | -1.005 | **0.889** | 0.774 | -23.52 | -5.375 | **0.987** |
| Semantic Duplicate | -16.31 | -229.2 | 0.541 | **0.947** | 0.677 | -16.121 | -6.629 | **0.991** |
| No Duplicate | 0.511 | -13.71 | -1.277 | **0.923** | 0.758 | -21.72 | -0.437 | **0.963** |

To summarize, using watermarked data (**P1**) may contribute to some performance gains in the separability desideratum **D1** for the baseline unlearning metrics. However, **P3** and WaterDrum are essential to satisfy **D2** under **D3** and **D4** (i.e., all proposed desiderata).

## G  ADDITIONAL EXPERIMENTAL RESULTS

### G.1  QUANTITATIVE EVIDENCE THAT WATERMARKING WITH WATERFALL DOES NOT DEGRADE LLM'S PERFORMANCE

`WaterDrum` lays out watermarking desiderata for compatible watermarking methods (Sec. 4), including the fidelity desideratum **W0**. We choose to use `Waterfall` (Lau et al., 2024) as this work has already presented extensive empirical results showing that its watermarking process has minimal degradation on the LLM's performance (see App. H.3 in (Lau et al., 2024)).

Nonetheless, we evaluate `Waterfall`'s fidelity by comparing the LLMs' performance when fine-tuned on watermarked vs. unwatermarked data using Truth Ratio (Maini et al., 2024) (App. E.3), which computes each LLM's probability of generating the correct answer compared to a set of wrong answers perturbed from the correct one.

Our results show that the mean Truth Ratio metric values when fine-tuning the LLMs on watermarked vs. unwatermarked versions of the `WaterDrum-TOFU` dataset are very similar at $0.5121$ and $0.5192$, respectively.

### G.2 SIMILARITY OF TEXT OUTPUTS TO QUERIES FORMED BY DUPLICATE DATA

Following the setup discussed in Sec. 5 (specifically, under 'Robustness to similar data **D4**'), under the 'exact and semantic duplicate' settings, we will verify here that when the LLM is fine-tuned on the augmented retain set $\mathcal{D}_{\mathcal{R}}^s$, it generates similar text outputs to queries formed by duplicate data points $d \in \mathcal{D}_s$ and $d_f \in \mathcal{D}_{\mathcal{F}}$ such that $d \simeq d_f$.

We empirically verify the similarity by evaluating the STS scores between the LLM's text outputs to queries formed by duplicate data points $d \in \mathcal{D}_s$ and $d_f \in \mathcal{D}_{\mathcal{F}}$. It can be observed from Table 11 that the mean STS scores are $0.96$ and $0.87$ under the 'exact and semantic duplicate' settings, respectively. For reference, the STS scores between the LLM's text outputs to queries formed by data points in the same academic subject category (i.e., `math.PR` in arXiv) in the `WaterDrum-Ax` dataset only result in a mean STS score of $0.67$. This shows that the LLM's text outputs to queries formed by duplicate data are very similar, much more so than that formed by data in the same academic subject category.

Table 11: Mean semantic text similarity (STS) score between the LLM's text outputs to queries formed by duplicate data points $d \in \mathcal{D}_s$ and $d_f \in \mathcal{D}_{\mathcal{F}}$ under the 'exact and semantic duplicate' settings for the `WaterDrum-Ax` dataset. For reference, the mean STS score between the LLM's text outputs to queries formed by data points in the same academic subject category (i.e., `math.PR` in arXiv) is $0.67$.

| Data Similarity | Mean STS score |
| --- | --- |
| Exact Duplicate | 0.96 |
| Semantic Duplicate | 0.87 |

We further verify that data points in $\mathcal{D}_s$ and $\mathcal{D}_{\mathcal{F}}$ with similar semantic meaning will have similar metric values. We use `WaterDrum` to measure the metric values of data points in $\mathcal{D}_s$ and $\mathcal{D}_{\mathcal{F}}$ (respectively, $M(\varphi_{\mathcal{T}}(q_d), \mathcal{R})$ for any $d \in \mathcal{D}_s$ and $M(\varphi_{\mathcal{T}}(q_d), \mathcal{F})$ for any $d \in \mathcal{D}_{\mathcal{F}}$) for the LLM $\varphi_{\mathcal{T}}$ fine-tuned on the full dataset of $\mathcal{D}_{\mathcal{F}} \bigcup \mathcal{D}_{\mathcal{R}}^s$ (note that the retain set is augmented with $\mathcal{D}_s$). Fig. 12 shows histograms of the `WaterDrum` values of data points in $\mathcal{D}_s$ and $\mathcal{D}_{\mathcal{F}}$ with similar semantic meaning, which verifies that the `WaterDrum` values of data points in $\mathcal{D}_s$ and $\mathcal{D}_{\mathcal{F}}$ indeed have a similar distribution.

### G.3 VERIFICATION PERFORMANCE OF WATERFALL WITH DIFFERENT GENERATED TOKEN LENGTHS

In Sec. 5, we generate up to 200 tokens for each LLM's query when evaluating each unlearning metric (App. E.2). This token length roughly translates to around 5 sentences.

The work of Lau et al. (2024) has shown that the verification performance of `Waterfall` improves with more tokens in longer text. To verify this claim, we consider the LLM fine-tuned on the `WaterDrum-Ax` dataset. With a false positive rate of $1\%$, the true positive rate reaches $50\%$ at 13 tokens generated and $90\%$ at 59 tokens generated.

## H ADDITIONAL EXPERIMENTAL RESULTS ON LLM UNLEARNING EVALUATION

In this section, we provide additional experimental results to evaluate `WaterDrum` and the baseline unlearning metrics on both `WaterDrum-Ax` and `WaterDrum-TOFU` datasets using the same

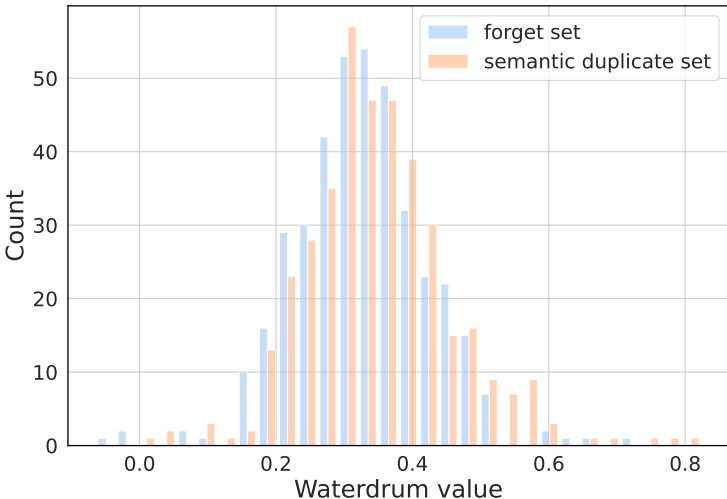

Figure 12: Histograms of the `WaterDrum` values of data points in $\mathcal{D}_s$ and $\mathcal{D}_{\mathcal{F}}$ (with similar semantic meaning) when unlearning 1 category of the `WaterDrum-Ax` dataset, which shows that the `WaterDrum` values of data points in $\mathcal{D}_s$ and $\mathcal{D}_{\mathcal{F}}$ have a similar distribution.

experimental setup described in Sec. 5 (unless stated otherwise), as well as benchmark unlearning algorithms for the cases with multiple data owners and different levels of data similarity on the new `WaterDrum-Ax` dataset.

## H.1 LLM UNLEARNING EVALUATION ON WATERDRUM-AX DATASET

### H.1.1 RELAXATION OF FEASIBILITY DESIDERATUM **D3**

In Sec. 5 (specifically, under 'Calibration desideratum **D2**' and Fig. 3), we demonstrate whether the unlearning metrics are calibrated well or poorly without referencing the retrained LLM $\varphi_{\mathcal{R}}$. Here, we relax the feasibility constraint and allow the baseline unlearning metrics (i.e., ROUGE, KnowMem, and MIA) to reference $\varphi_{\mathcal{R}}$ although doing so infeasibly requires retraining for every forget set being considered.

Specifically, we reference the retrained LLM $\varphi_{\mathcal{R}}$ (i.e., achieved by perfect unlearning) by subtracting its aggregate metric value from that on the unlearned model $\widetilde{\varphi}$ to yield an 'offset' aggregate metric $M_{d \in \mathcal{D}_{\mathcal{F}}}^{-}(\widetilde{\varphi}(q_d), \mathcal{F}) := M_{d \in \mathcal{D}_{\mathcal{F}}}(\widetilde{\varphi}(q_d), \mathcal{F}) - M_{d \in \mathcal{D}_{\mathcal{F}}}(\varphi_{\mathcal{R}}(q_d), \mathcal{F})$.

Fig. 13 and Table 12 show, respectively, the calibration curves for the various unlearning metrics (using the 'offset' aggregate metric values) and the $R^2$ values for the corresponding best-fit lines under different levels of data similarity for the `WaterDrum-Ax` dataset. The results show that, under the relaxed feasibility constraint by referencing $\varphi_{\mathcal{R}}$, the baseline metrics are generally better calibrated. Notably, ROUGE achieves a good calibration across different levels of data similarity even though it underperforms in the 'exact duplicate' setting. In contrast, our `WaterDrum` consistently demonstrates strong calibration with high $R^2$ values across all settings. Nonetheless, it is important to emphasize that the retrained LLMs are not available in practical scenarios and their availability would eliminate the need to perform unlearning in the first place.

Table 12: $R^2$ values for the best-fit lines (dotted in Fig. 13) of various unlearning metrics (using the 'offset' aggregate metric values) under different levels of data similarity for the `WaterDrum-Ax` dataset.

| Data Similarity | ROUGE | KnowMem | MIA | WaterDrum |
|---|---|---|---|---|
| Exact Duplicate | 0.923 | -0.331 | 0.273 | 0.994 |
| Semantic Duplicate | 0.997 | 0.101 | -0.011 | 0.995 |
| No Duplicate | 0.998 | 0.006 | 0.990 | 0.957 |

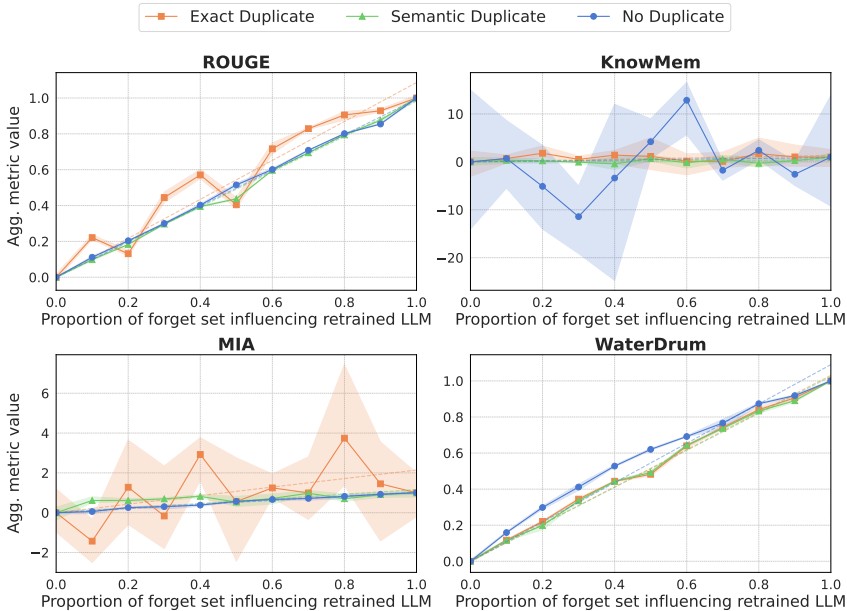

Figure 13: Calibration curves for various unlearning metrics (using the 'offset' aggregate metric values $M^-_{d\in\mathcal{D}_\mathcal{F}}(\widetilde{\varphi}(q_d),\mathcal{F})$) w.r.t. proportion $k/|\mathcal{D}_\mathcal{F}|$ of the forget set influencing the retrained LLM (solid) and their best-fit lines (see associated $R^2$ in Table 12) through the origin (dotted) under different levels of data similarity for the WaterDrum-Ax dataset. The 'offset' aggregate metric values are offset by referencing the retrained LLMs and scaled by referencing the original LLMs such that the values are 0.0 and 1.0 when the proportions are 0.0 and 1.0 respectively.

## H.2 LLM UNLEARNING EVALUATION ON WATERDRUM-TOFU DATASET

As a supplement to the main experiments, we present additional experimental results here for the WaterDrum-TOFU dataset. As described in Sec. 5 (specifically, under 'Robustness to similar data **D4**'), we consider the 'exact duplicate', 'semantic duplicate', and 'no duplicate' settings, and fine-tune the LLMs on the WaterDrum-TOFU dataset. While Sec. 5 (specifically, under 'Separability desideratum **D1**') discusses results on the separability desideratum **D1** under different levels of data similarity, we report below the results to evaluate WaterDrum and the baseline unlearning metrics in the calibration desideratum **D2** and the relaxed feasibility desideratum **D3** under different levels of data similarity.

### H.2.1 CALIBRATION DESIDERATUM **D2**

Fig. 14 and Table 13 show, respectively, the calibration curves for the various unlearning metrics and the $R^2$ values for the corresponding best-fit lines under different levels of data similarity for the WaterDrum-TOFU dataset. Similar to the results in Sec. 5 (specifically, under 'Calibration desideratum **D2**'), our WaterDrum outperforms the baseline metrics by ensuring $M'_{d\in\mathcal{D}_\mathcal{F}}(\varphi_\mathcal{R}(q_d),\mathcal{F})$ to be close to 0 at $k=0$ and maintaining strong calibration with high $R^2$ values without referencing retrained LLMs across all settings.

Table 13: $R^2$ values for the best-fit lines (dotted in Fig. 14) of various unlearning metrics under different levels of data similarity for the WaterDrum-TOFU dataset. WaterDrum achieves the highest $R^2$ values that are closest to 1 and is hence a well-calibrated metric.

| Data Similarity | ROUGE | Truth Ratio | MIA | WaterDrum |
|---|---|---|---|---|
| Exact Duplicate | -1263.951 | -6444.874 | -6.201 | 0.889 |
| Semantic Duplicate | -20.560 | -1416.284 | -0.813 | 0.947 |
| No Duplicate | -0.287 | -11.741 | -1.059 | 0.923 |

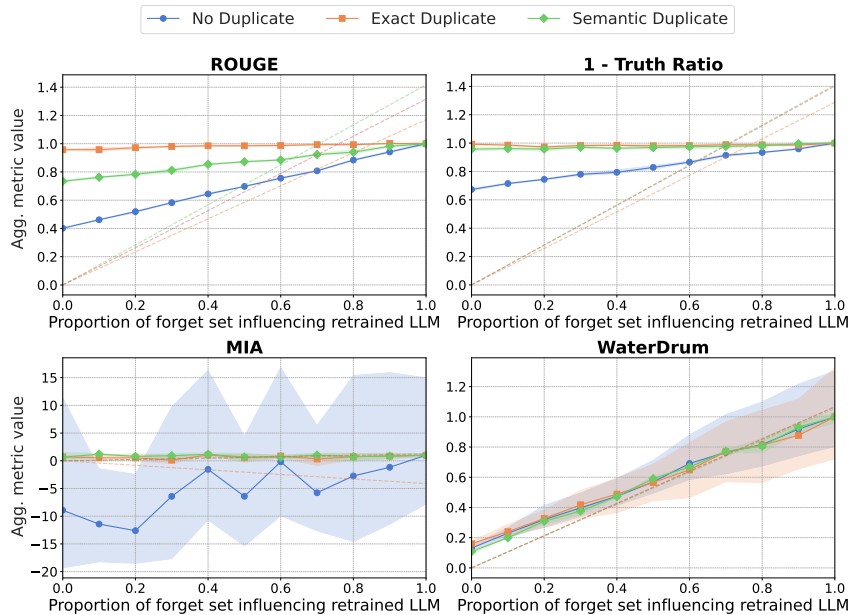

Figure 14: Calibration curves for various unlearning metrics w.r.t. proportion $k/|\mathcal{D}_{\mathcal{F}}|$ of the forget set influencing the retrained LLM (solid) and their best-fit lines (see associated $R^2$ in Table 13) through the origin (dotted) under different levels of data similarity for the `WaterDrum-TOFU` dataset. Only `WaterDrum` is well-calibrated and satisfies **D2** with its best-fit lines closely following its aggregate metric values.

### H.2.2 RELAXATION OF FEASIBILITY DESIDERATUM **D3**

Similar to App. H.1.1, we relax the feasibility constraint here and allow the baseline unlearning metrics (i.e., ROUGE, Truth Ratio, and MIA) to reference the retrained LLM $\varphi_{\mathcal{R}}$ although doing so infeasibly requires retraining for every forget set being considered.

Fig. 15 and Table 14 show, respectively, the calibration curves for the various unlearning metrics (using the 'offset' aggregate metric values defined in App. H.1.1) and the $R^2$ values for the corresponding best-fit lines under different levels of data similarity for the `WaterDrum-Ax` dataset. The results are similar to that in App. H.1.1 and show that, under the relaxed feasibility constraint by referencing $\varphi_{\mathcal{R}}$, the baseline metrics are generally better calibrated. Unlike Truth Ratio and MIA, our `WaterDrum` and ROUGE consistently demonstrate strong calibration with high $R^2$ values across all settings. Nonetheless, it is important to emphasize again that the retrained LLMs are not available in practical scenarios and their availability would eliminate the need to perform unlearning in the first place.

Table 14: $R^2$ values for the best-fit lines (dotted in Fig. 15) of various unlearning metrics (using the 'offset' aggregate metric values) under different levels of data similarity for the `WaterDrum-TOFU` dataset.

| Data Similarity | ROUGE | Truth Ratio | MIA | `WaterDrum` |
|---|---|---|---|---|
| Exact Duplicate | 0.991 | -0.586 | -0.018 | 0.997 |
| Semantic Duplicate | 0.998 | 0.854 | -0.435 | 0.996 |
| No Duplicate | 0.999 | 0.995 | 0.608 | 0.997 |

### H.3 BENCHMARKING UNLEARNING ALGORITHMS ON NEW `WATERDRUM-AX` DATASET FOR MULTIPLE DATA OWNERS AND DIFFERENT LEVELS OF DATA SIMILARITY

In addition to the experimental results in Sec. 5.1, Figs. 16 and 17 illustrate the use of `WaterDrum` in benchmarking the unlearning algorithms under the respective 'no duplicate' and 'exact duplicate'

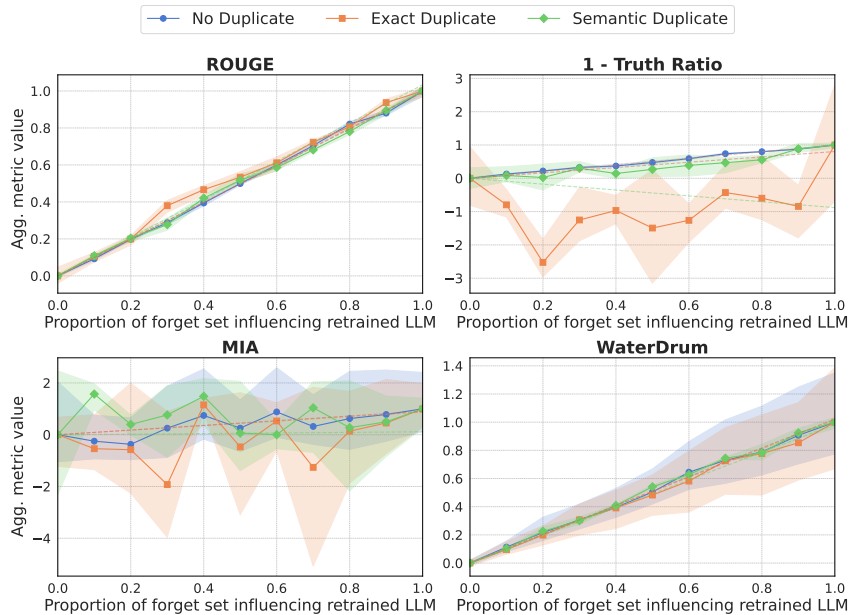

Figure 15: Calibration curves for various unlearning metrics (using the 'offset' aggregate metric values) w.r.t. proportion $k/|\mathcal{D}_{\mathcal{F}}|$ of the forget set influencing the retrained LLM (solid), scaled by referencing the retrained and original LLMs, and their best-fit lines (see associated $R^2$ in Table 14) through the origin (dotted) under different levels of data similarity for the `WaterDrum-TOFU` dataset.

settings of data similarity (i.e., previously described in Sec. 5, specifically, under 'Robustness to similar data **D4**') for the `WaterDrum-Ax` dataset where the forget set consists of data from 1, 3, and 5 data owners (out of a total of 20 data owners) with 1 category of paper abstracts per owner (App. C.3).

Similar to the results in Sec. 5.1, it can be observed from Fig. 16 (Fig. 17) that the unlearning algorithms achieve aggregate `WaterDrum` values still far from that achieved by retraining: KL and TV generally produce unlearned models that unlearn the forget set $\mathcal{D}_{\mathcal{F}}$ very well but cannot preserve the influence of $\mathcal{D}_{\mathcal{R}}$ (or $\mathcal{D}_s$) much, the latter of which compromises their overall utility. GD and SCRUB tend to produce unlearned models that preserve some influence of $\mathcal{D}_{\mathcal{R}}$ (or $\mathcal{D}_s$) but do not unlearn the forget set $\mathcal{D}_{\mathcal{F}}$ well. However, both GD and SCRUB require fine-tuning on the (augmented) retain set ($\mathcal{D}_{\mathcal{R}}^s = \mathcal{D}_s \bigcup \mathcal{D}_{\mathcal{R}}$), which incurs a significant amount of computational resources as the (augmented) retain set is likely to be significantly larger than the forget set and almost similar in size to the full dataset. Typically, LLM fine-tuning only involves very few epochs (Touvron et al., 2023). The computational cost of fine-tuning the LLM for a few epochs on the (augmented) retain set can be almost as expensive as that of retraining.

It can also be observed from Fig. 16 (Fig. 17) that when the forget set consists of data from 5 data owners, the aggregate `WaterDrum` value of the watermarked retain set $\mathcal{D}_{\mathcal{R}}$ in `WaterDrum-Ax` on the retrained LLM (only on the (augmented) retain set) increases slightly beyond 1.0. We hypothesize that this is due to the forget set constituting a larger proportion of the entire dataset (i.e., 5 out of a total of 20 data owners). As a result, the (augmented) retain set used for retraining becomes smaller in proportion relative to the full dataset $\mathcal{D}_{\mathcal{T}}$, which can result in the retrained LLM becoming more specialized in this smaller (augmented) retain set and in turn a larger aggregate `WaterDrum` value. The same reasoning applies to explain why the aggregate `WaterDrum` value of $\mathcal{D}_s$ on the retrained LLM (only on the (augmented) retain set) also increases slightly beyond 1.0 in Fig. 17.

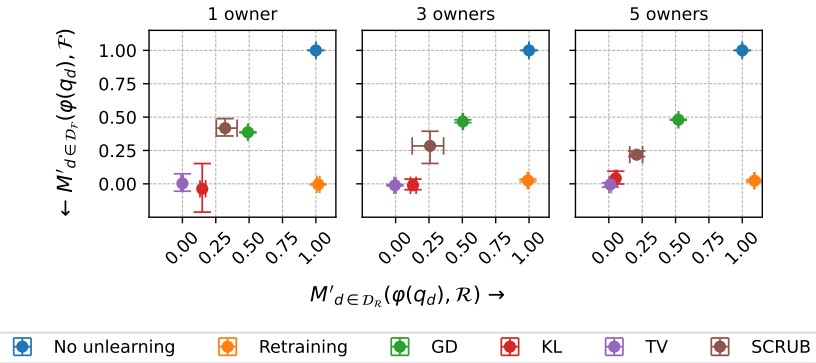

Figure 16: Benchmarking the unlearning algorithms with `WaterDrum` under the 'no duplicate' setting of data similarity for the `WaterDrum-Ax` dataset where the forget set consists of data from 1, 3, and 5 data owners with 1 category of paper abstracts per owner (App. C.3).

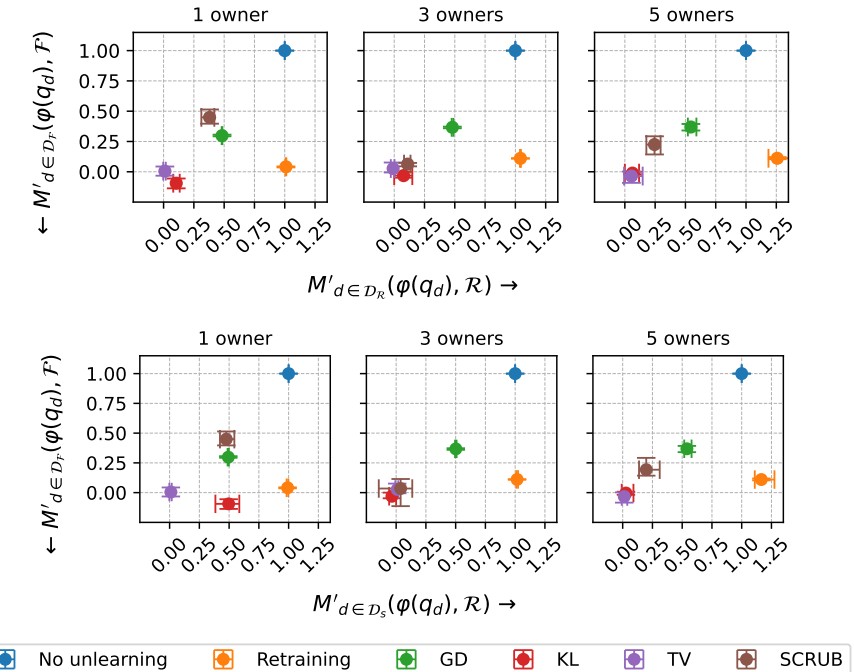

Figure 17: Benchmarking the unlearning algorithms with `WaterDrum` under the 'exact duplicate' setting of data similarity (i.e., previously described in Sec. 5, specifically, under 'Robustness to similar data **D4**') for the `WaterDrum-Ax` dataset where the forget set consists of data from 1, 3, and 5 data owners with 1 category of paper abstracts per owner (App. C.3).

## I OTHER QUESTIONS AND LIMITATIONS

1. **What is the difference of `WaterDrum` with existing watermark-based unlearning metrics?** Existing watermark-based unlearning metrics are mostly for image classification models, as opposed to our proposed unlearning metric for text-based generative LLMs. See discussion on watermark-based metrics in App. A.1 for more details.

2. **Existing works (Liu et al., 2025; Lynch et al., 2024) have already identified similar limitations about existing unlearning metrics. What is the novelty of our work?** We formally define clear desiderata and propose a non-retraining-based metric that works despite

a higher level of data similarity between the forget vs. retain sets and the generalization ability of LLMs. See more discussion in App. A.

3. **Why do we only run experiments on TOFU and `WaterDrum-Ax` datasets instead of other datasets such as WMDP?** TOFU and `WaterDrum-Ax` datasets cover both QA and completion tasks, which are representative of LLM tasks. WMDP is different from TOFU and `WaterDrum-Ax` in nature because it is specifically for knowledge editing and only contains test data instead of training data. As our work considers a data-centric view of unlearning, we are concerned with the unlearning of specific data owners' contributed datasets (with potential similar data instances across data owners), rather than indiscriminately unlearning certain (harmful) knowledge.

4. **Can our conclusion be generalized to other LLMs?** Results on the Phi-1.5 model (see App. F.3) show that our conclusion can be generalized to other LLMs as well. The two LLMs considered in our paper are representative of recent LLMs, different in terms of model architectural details, and span different model scales. These two LLMs are also the only LLMs considered in (Maini et al., 2024; Wang et al., 2025).

5. **Beyond unlearning effectiveness, can our proposed metric be used to measure utility preservation/retention?** As shown in Sec. 5.1, our `WaterDrum` can be used to verify that the aggregate `WaterDrum` value of the retain set on the retrained LLM (i.e., perfect unlearning) is similar to that on the original LLM (i.e., no unlearning). Hence, our `WaterDrum` can also quantify the extent of undesirable removal of the retain set's influence and evaluate the effects of catastrophic forgetting.

6. **Practical significance of unlearning of fine-tuning data vs. pre-training data.** In real-life applications, fine-tuning of an LLM is performed to enhance the LLM in specific downstream tasks, which is more likely to make use of task-specific datasets. These datasets are more concerned with privacy/safety issues and are hence more significant for unlearning than public datasets used in pre-training.

7. **What are the limitations of `WaterDrum` and this work?** The limitations are that (a) the desiderata may not be exhaustive, (b) the `WaterDrum` value (via the watermark's verification score in Eq. (3)) may not exhaustively capture all possible ways of measuring unlearning effectiveness, and (c) `WaterDrum` requires the training data to be watermarked unlike existing metrics.

   We believe that for now, (a) and (b) are acceptable as our work is an important *first* step towards designing and developing more effective and practical unlearning metrics and algorithms, and deriving theoretical results for them. Future work can conduct a more comprehensive and systematic evaluation of existing LLM unlearning algorithms and adapt theoretical insights from the watermarking community to analyze the LLM unlearning metrics based on the new connection that we have established in this work.

   In Remark 1 (Sec. 4) and App. C.4, we explain why watermarking is lightweight, easy to use, and would be a more common practice in the future. Thus, the applicability of `WaterDrum` would increase and limitation (c) would diminish over time. Moreover, limitation (c) is reasonable as the benefits, such as satisfying our desiderata, outweigh the slight inconvenience and cost.

8. **What new insights can be gained from the proposed `WaterDrum`?** (a) We have shown that existing metrics fail to satisfy our necessary desiderata (Sec. 3), prompting caution in the design of unlearning metrics. (b) Using `WaterDrum` to benchmark LLM unlearning algorithms (Sec. 5.1) shows that they perform poorly on unlearning and retaining performance. `WaterDrum` can serve as an optimization criterion for future LLM unlearning algorithms. (c) By emphasizing practicality in our proposed desiderata (Sec. 3), `WaterDrum` encourages future LLM unlearning algorithms to consider realistic settings/constraints.

9. **Why do we not consider other desiderata?** Our work focuses on the most essential desiderata (effectiveness desiderata) and more practical/realistic settings/constraints. We find these desiderata to be the most relevant and necessary criteria for effective and practical unlearning metrics, though they are not meant to be exhaustive nor by themselves sufficient to guarantee unlearning. We see our work as complementary to other compatible frameworks.

