# OpenReview forum: "WaterDrum: Watermark-based Data-centric Unlearning Metric"
_ICLR.cc/2026/Conference — ICLR 2026 Poster_

### Official Review · Reviewer_ftSk · 2025-10-16

**Soundness:** 3
**Presentation:** 3
**Contribution:** 3
**Rating:** 6
**Confidence:** 3

**Summary:**

This paper proposes WaterDrum, the first data-centric unlearning metric for LLMs, which leverages robust text watermarking to overcome the limitations of existing utility-centric metrics that fail to accurately measure the degree of unlearning in real-world scenarios.

**Strengths:**

1. The idea of evaluating unlearning from the watermarking perspective is novel and insightful.
2. The introduction of the new benchmark WaterDrum-Ax provides a useful foundation for future research in this area.

**Weaknesses:**

1. The proposed method has an inherent limitation, it requires the training data to be watermarked prior to model training.
2. The methodology section is somewhat verbose and repetitive, which makes the experimental part appear less substantial in comparison.

**Questions:**

1. Apart from Waterfall, are there other text watermarking methods that could potentially meet WaterDrum’s watermarking requirements? Have you considered conducting compatibility tests with alternative watermarking schemes?
2. For datasets that have already been released without watermarks, are there any remedial or alternative approaches that would allow WaterDrum to still be applied for unlearning evaluation?
3. Can WaterDrum be extended to assess unlearning during the pre-training stage, rather than only in fine-tuning? If so, would it require adjustments to the watermarking strategy or validation process?
4. Beyond the LLaMA2-7B model, have you evaluated WaterDrum on other model families or larger-scale models? If so, how do the experimental results differ?

---

> ### Author Response · Authors · 2025-11-26
> **Response to Reviewer ftSk (part 1/4)**
>
> Thank you for your reviews and questions! We will address your concerns below.
>
> ---
>
> > [W1] The proposed method has an inherent limitation, it requires the training data to be watermarked prior to model training.
>
> > [Q2] For datasets that have already been released without watermarks, are there any remedial or alternative approaches that would allow WaterDrum to still be applied for unlearning evaluation?
>
> As discussed in Appendix C.4, the practical concern that the data needs to be watermarked prior to training will likely diminish with time:
>
> 1. Recalling **already released data** may be possible in our unlearning setting as data owners **have the rights to their data and can control their use**, as discussed in Sec. 1 (lines 27-31) and Sec. 2 (lines 73-75). Therefore, data owners of released datasets can still exercise the rights to their data by telling the model owner that they would (i) require their updated watermarked data to be used in the LLM instead, or (ii) consent to the continued use only if watermarking is to be part of the data processing step in the next LLM release. In either case, the model owner must comply with laws and regulations such as the GDPR and copyright laws.
>
> 2. Even without recalling historical and released data, the data owners can demand that watermarks be applied going forward in future LLMs. They can **expect newly generated data to be watermarked, hence facilitating future practical LLM unlearning evaluations**. For example, news agencies can start watermarking their news articles, and these recent articles may be more important in training future LLMs. Here are more justifications why applying watermarks is often feasible in practice:
>     - As the awareness of privacy and security in LLM training data grows, we expect that proactive watermarking of data before release is likely to become a common practice in the future for data owners who are concerned about privacy and security. Therefore, the applicability of WaterDrum could also grow over time.
>     - The watermarking process is lightweight and incurs very little computational cost. This makes the watermarking process simple and convenient for the data owners and model owners. The watermarking process can be part of the data processing step.
>
> 3. A similar concept called **image watermarking** in the domain of computer vision has been widely studied and adopted for image data copyright protection [1]. We note that when applications of image watermarking are proposed, such as data backdoors to verify unlearning [2], they also face the same constraint that image watermarking can only be applied to unreleased data (and cannot be retroactively embedded in historical data). The community has accepted the constraint and appreciated the potential benefits going forward. **Thus, there are strong reasons to believe in the potential for wider adoption of text watermarking and its applications**.
>
> **References**
>
> [1] Cox, Ingemar J., et al. (2008). "Digital watermarking." Morgan Kaufmann Publishers 54: 56-59.
>
> [2] Thaker, Pratiksha, et al. (2025). "Position: Llm unlearning benchmarks are weak measures of progress." 2025 IEEE Conference on Secure and Trustworthy Machine Learning (SaTML).
>
> ---
>
> > [W2] The methodology section is somewhat verbose and repetitive, which makes the experimental part appear less substantial in comparison.
>
> We would like to gently clarify that the structure of the methodology section is intended to make the paper accessible to readers with different backgrounds and preferences. The detailed formulations in the methodology section are to **ensure there is no ambiguity in the descriptions** and are catered to those readers who prefer a more rigorous formulation. Meanwhile we have also ensured that this level of detail does not compromise the presentation or depth of the experimental section.
>
> To clarify, we have included **the most crucial empirical results** in the main paper to substantially support our claims and demonstrate how WaterDrum satisfies the desiderata in Sec. 4 and compare against baseline unlearning metrics. Additional experiments and analyses are also provided in the appendix for completeness:
> - Appendix F provides ablation studies, including calibration as the expectation of the aggregate metric value (F.1); results on full parameter LLMs (F.2); results on other LLMs (F.3); results for other metrics on LLMs finetuned on watermarked data (F.4).
> - Appendix G provides additional analysis results, including WaterFall's fidelity (G.1); similarity of outputs to queries formed by duplicate data (G.2); WaterFall on different token lengths (G.3).
> - Appendix H provides additional results of WaterDrum, including the results when we relax D3 Feasibility (H.1, H.2.2); additional results on WaterDrum-TOFU (H.2.1); results for benchmarking unlearning algorithms on WaterDrum-Ax with duplicate data (H.3).
>
> ---
>
> &#8595; &#8595; &#8595; **Continued below** &#8595; &#8595; &#8595;

---

> ### Author Response · Authors · 2025-11-26
> **Response to Reviewer ftSk (part 2/4)**
>
> > [Q1] Apart from Waterfall, are there other text watermarking methods that could potentially meet WaterDrum’s watermarking requirements? Have you considered conducting compatibility tests with alternative watermarking schemes?
>
> Thank you for your suggestion on testing with alternative watermarking schemes. We have discussed and considered several alternative watermarking methods and how they fail to satisfy our desiderata in Appendix A.3, C.2. Meanwhile, it is not a central claim/contribution to use Waterfall in our WaterDrum framework; rather, WaterDrum is designed to work with any text watermarking scheme that satisfies the desiderata outlined in Sec. 4. In principle, any other text watermarking schemes that satisfy our desiderata would be compatible.
>
> In our paper, we focused on text watermarking from the data owners' perspective where different data owners want to independently verify if their data still has influence on the LLM. However, there also exists a separate body of work on model watermarking that considers watermarking from the model owner's perspective, which we described in App. D.3. Model watermarking injects a watermark into the LLM’s output text and tokens typically used by a single model owner to distinguish whether a piece of text is generated by their model. We described this in Appendix D.3. We considered some popular model watermarking methods, including KGW [3], Synth-ID [4], and EXP-edit [5]. However, based on just the designs and experimental support provided in references [3,4,5], it was not clear that all of the watermarking desiderata would be met. Additionally, we still needed further adaptations to [3,4,5] and additional experiments to be run to establish that the watermarking desiderata were met:
> - For example, to satisfy our **W4 Unique key**, the text watermarking schemes that inject watermarks into training texts need to support insertion and verification of watermarks from *different/multiple* data owners. In contrast, model watermarking schemes [3,4,5] designed for watermarking a single LLM's generated text typically only requires a *single* watermark to serve the single model owner.
> - The requirements of **W2 Overlap verifiability** were also not explored, but is an important requirement for model watermarking schemes to be used in WaterDrum. On the other hand, Sec. 4.3 of Waterfall [6] had established this property of being able to recover *multiple different* watermark signals in text generated by LLMs trained on text watermarked by *many data owners* in their experiments, giving us more confidence in using it for demonstrating the effectiveness of our WaterDrum unlearning metric.
>
> ||W0|W1|W2|W3|W4|
> |-|-|-|-|-|-|
> |Waterfall [6]|Yes|Yes|Yes|Yes|Yes|
> |KGW [3]|No^|Yes*|Not shown|Yes|No^|
> |Synth-ID [4]|No^|No*|Not shown|Yes|No^|
> |EXP-edit [5]|No^|No*|Not shown|Yes|Yes|
>
> ^ No off-the-shelf, requires modification to their code to support the desideratum
>
> \* Based on our results shown below after adaptation of the schemes
>
>
> Nonetheless, it may be possible to try adapting the schemes to satisfy the desiderata. To run our experiments below, we have tried adapting the schemes [3,4,5] to meet the watermarking desiderata, though we believe that future work might be able to nontrivially improve the design of the different watermarking schemes and satisfy the desiderata to a better extent than our implementations below and than WaterFall:
>
> - We adapted [3,4,5] and ran additional experiments by adding a paraphrasing prompt following the approach in WaterFall [6]. As [3,4] were designed for "watermarking (of LLM-generated text) for a model owner" (as discussed above), their codebase had explicitly fixed single watermark IDs into the internal components of the source code -- we had to modify it for them to accept different watermarks and support multiple data owners.
> - We used the same "Llama-3.1-8B-Instruct" LLM to apply text watermarking on the ArXiv dataset, except for Synth-ID for which we used "Gemma-7b-it" as the provided codebase only supports Gemma and GPT-2 models.
>
> ---
>
> &#8595; &#8595; &#8595; **Continued below** &#8595; &#8595; &#8595;

---

> ### Author Response · Authors · 2025-11-26
> **Response to Reviewer ftSk (part 3/4)**
>
> **Contunuing previous response**
>
> ---
>
> The table below shows results of the adapted [3,4,5] and Waterfall in D1 Separability and D2 Calibration desiderata, along with the average time required to verify a single text sample and whether a GPU is required for verification.
> - It is promising to see that the adapted KGW performs very well in D2 Calibration with an $R^2$ value of almost $1$, though it has worse performance than Waterfall in D1 Separability and also requires significantly more verification time and GPU resources. This can possibly be attributed to D1 Separability requiring the W2 Overlap verifiability desideratum, which KGW may not be designed to achieve due to its "watermarking (of LLM-generated text) for a model owner". In contrast, Waterfall was designed for the "watermarking (of training text) for multiple data owners" and hence had different watermark ID signals designed to be orthogonal to each other, allowing for greater distinction among IDs and hence satisfying W2 and D1, respectively. Its design may also have resulted in its slightly lower $R^2$ value in the D2 Calibration desideratum than adapted KGW (i.e., $0.963$ vs. $0.996$).
>     - Hence, in the setting where there is only $1$ data owner/forget set with the rest of the retain set being public data, satisfying D2 is more important than satisfying D1, and it is practical to use significantly more verification time and GPU resources, it may be better to use the adapted KGW as the watermarking method in WaterDrum since the W2 desideratum may be relaxed in this case. However, for general settings, it may still be better to use Waterfall given its better performance in D1 and D2.
> - Unfortunately, [4,5] did not perform well, likely because they were not designed for satisfying W1 and W2.
> - In addition, the adapted methods [3,4,5] either require a GPU or a few orders of magnitude longer time than Waterfall to verify whether a single text sample contains the watermark. This places practical constraints on when these methods can be used in WaterDrum.
>
> ||D1 Separability|D2 Calibration|Verification time|Require GPU|
> |-|-|-|-|-|
> |Waterfall [6]|0.965|0.963|0.015s|No|
> |adapted KGW [3]|0.8710|0.996|0.336s|Yes|
> |adapted Synth-ID [4]|0.5488|-16.951|0.386s|Yes|
> |adapted EXP-edit [5]|0.7891|-17.079|165.5s|No|
>
> We will incorporate your suggestion and add this meaningful comparison to our revised paper.
>
> **References**
>
> [3] Kirchenbauer, John, et al. (2023). "A watermark for large language models." In Proc. ICML.
>
> [4] Dathathri, Sumanth, et al. (2024). "Scalable watermarking for identifying large language model outputs." Nature 634.8035: 818-823.
>
> [5] Kuditipudi, Rohith, et al. (2024). "Robust Distortion-free Watermarks for Language Models." TMLR.
>
> [6] Lau, Gregory Kang Ruey, et al. (2024). "Waterfall: Scalable Framework For Robust Text Watermarking and Provenance for LLMs." In Proc. EMNLP.
>
> ---
>
> &#8595; &#8595; &#8595; **Continued below** &#8595; &#8595; &#8595;

---

> ### Author Response · Authors · 2025-11-26
> **Response to Reviewer ftSk (part 4/4)**
>
> > [Q3] Can WaterDrum be extended to assess unlearning during the pre-training stage, rather than only in fine-tuning? If so, would it require adjustments to the watermarking strategy or validation process?
>
> Pre-training would likely involve a much larger number of data owners than fine-tuning. Nonetheless, **WaterDrum can be seamlessly applied to assess unlearning during the pre-training stage**. Text watermarks designed to satisfy W2 Overlap verifiability and W4 Unique key desiderata would be able to cater to the large number of data owners involved in pre-training. Waterfall [6] supports W4 up to 10^130274 data owners. Furthermore, [6] showed that when more data owners are added, degradation of W2 plateaus beyond 20 data owners, a trend that can be expected to hold for a much larger number of data owners. In addition, even if unwatermarked public datasets are included in the pre-training data along with watermarked data, for text watermarks that support W1 Verifiability, we do not expect the verifiability of the watermarked training data to be affected. [6] demonstrated that watermarked text can be clearly distinguished from unwatermarked text with watermark verification.
>
> As pre-training frequently uses data scraped from the internet, data owners should pre-emptively watermark all their data before releasing them to the internet, no matter whether they expect their data to be used for LLM training or not. In addition, due to the large number of potential data owners watermarking their data, the watermark key should be chosen randomly (rather than defaulting to common picks such as 42) to minimize the chances of collision with other data owners. Otherwise, in general, **we do not expect any other adjustments to be required for the watermarking strategy or the validation process**.
>
> We focused on the fine-tuning stage in our main paper because in real-life applications, LLM fine-tuning is performed to enhance the LLM in specific downstream tasks, which is more likely to make use of task-specific datasets. **These datasets are more concerned with privacy/safety issues compared to the broad and data used in pre-training**, and are hence more significant for unlearning than those public datasets.
>
>
> ---
>
> > [Q4] Beyond the LLaMA2-7B model, have you evaluated WaterDrum on other model families or larger-scale models? If so, how do the experimental results differ?
>
> In our paper, we have evaluated WaterDrum on another model family, Phi-1.5, as detailed in Appendix F.3. The results show high AUROC and $R^2$ values. To supplement this, we ran an additional experiment on a larger-scale model, Llama2-13B, on the WaterDrum-TOFU dataset. The results are presented in the table below, which, together with our Phi-1.5 experiment, demonstrates that WaterDrum performs well across different model families and sizes. We will add the experiment results to our revised paper.
>
> |            | Separability | Calibration |
> |------------|--------------|-------------|
> | Phi-1.5    |    0.913     |    0.991    |
> | Llama2-7B  |    0.928     |    0.923    |
> | Llama2-13B |    0.974     |    0.993    |
>
>
> ---
>
> Thank you again for reviewing our work. We hope that our clarifications and additional experiments will improve your evaluation of our paper. We would be happy to provide further clarification.

---

### Official Review · Reviewer_mg5c · 2025-10-31

**Soundness:** 2
**Presentation:** 1
**Contribution:** 3
**Rating:** 4
**Confidence:** 2

**Summary:**

The paper first points out that practical unlearning is not really well tested by existing unlearning benchmarks. Existing evaluation protocols 1) don’t test on semantically similar data, 2) need to train a “retrained LLM” (I like to call this the “ideal LLM”) and 3) need better ways to elicit the inclusion of a data point.

The main contribution of this work is a new metric for unlearning, that measures the model’s ability to unlearn similar data (?) and a new dataset based off of arxiv, which forms a new unlearning benchmark. Results show that the WaterDrum metric is superior and matches the desiderata laid out in the paper.

**Strengths:**

S1. This paper raises both interesting and important point that are neglected in unlearning benchmarks:

1) existing benchmarks (like TOFU) evaluate degradation of general model capabilities after unlearning, which does not truly evaluate whether the data has been forgotten and retain the desired data.

2) unlearning benchmarks need to think closely about how to elicit whether a model has forgotten or retained a datapoint.

3) practically, the forget set and retain sets may be semantically close to each other. if you’re just evaluating model capabilities, you would miss a more fine grained notion of whether the model has forgotten.

Because the focus is interesting, I wouldn’t mind if this paper was accepted.

**Weaknesses:**

W1. The presentation of this paper is not ideal, in my opinion. While I can see the authors obviously spent a lot of time in clarifying the writing, I think the notation is extremely dense for a paper that is not theoretical in nature.

W2. I think the focus on “desiderata” and how WaterDrum achieves them is not the best way to present this topic. While I appreciate the sincere thought about what an unlearning benchmark should be, I feel like there’s a more straightforward presentation of your findings and contributions by **highlighting gaps with current benchmarks and showing how your benchmark provides more sensitivity.** I feel this is a more sensible approach, given that some of the desiderata are plainly intuitive e.g. separability — I don’t really need a whole mathematical definition of what is basically captured by AUC ROC.

The paper focuses on interesting points, but I think the paper’s impact would be much better if it is presented in a more narrative way. In fact, I don’t really understand what is going on with most of the results as they point to all different aspects of the authors’ framework which I think is overtheorized.

W3. Finally, the last weakness is regarding the watermarking aspect. From what I gather, there needs to be a way to elicit whether an unlearned model forgets or retains data points. Watermarking could be a nice way to trace that, however, it seems that watermarking and detecting those watermarks would only focus on membership information. It does not seem to me to necessarily capture the higher level concepts within the forget set that is actually intended to be forgotten.

**Questions:**

n/a

---

> ### Author Response · Authors · 2025-11-26
> **Response to Reviewer mg5c (part 1/2)**
>
> Thank you for your review! We will address your concerns below.
>
> ---
>
> > [W1] The presentation of this paper is not ideal, in my opinion. While I can see the authors obviously spent a lot of time in clarifying the writing, I think the notation is extremely dense for a paper that is not theoretical in nature.
>
> > [W2] I think the paper’s impact would be much better if it is presented in a more narrative way. In fact, I don’t really understand what is going on with most of the results as they point to all different aspects of the authors’ framework which I think is overtheorized.
>
> We thank the reviewer for noticing our effort in clarifying the writing and raising the concern about the dense notations. We would like to gently point out to the reviewer that the presentation of the paper is intended to cater to readers with different backgrounds and preferences. The detailed theoretical formulations are for readers who prefer a more rigorous formulation, as these notations can ensure **no ambiguity in the descriptions**, hence allowing them to assess the correctness of our work and technical contributions.
>
> Meanwhile, we have also included **intuitive explanations** in the main paper: For example, Fig. 1 demonstrates AUROC (D1 Separability) with examples of WaterDrum and Truth Ratio, while Appendix B provides **intuitive illustrations** including Figs. 5, 6, 7 for the desiderata mentioned in Sec. 3. We have referenced them in the main paper.
>
> Here, we provide a short and more intuitive version of our experimental results:
> - The results in Table 2 correspond to our D1 Separability (as explained in lines 377-409), which measures the AUROC of the unlearning metrics and assesses whether they can distinguish the retain set vs. the forget set. We consider different levels of data similarity to verify D4 as well.
> - The results in Fig. 3 correspond to our D2 Calibration (as explained in lines 410-458). The plot shows the metric values at different proportions of forget set still influencing the unlearned LLM (i.e., extent of imperfect unlearning) and compares whether they match the best-fit line, considering different levels of data similarity for D4. We also report the $R^2$ value in Table 3 to quantify how well the metric matches the best-fit line.
> - In Fig. 4, we use our WaterDrum to benchmark the existing baseline unlearning algorithms (as explained in lines 460-478). Unlearning algorithms that are closer to retraining (bottom right corner) are better.
>
> We would appreciate it if the reviewer can point out results in the main experiments that are still unclear, so we can improve them.
>
> ---
>
> > [W2] While I appreciate the sincere thought about what an unlearning benchmark should be, I feel like there’s a more straightforward presentation of your findings and contributions by highlighting gaps with current benchmarks and showing how your benchmark provides more sensitivity.
>
> Thank you for this valuable feedback. We agree that our contribution can be highlighted through the gaps with current benchmarks. In fact, this is the intention when **we discussed the limitation of other existing benchmarks** in Sec. 2, and contrasted with existing works in Sec. 5 in our main paper. Our idea of "sensitivity" consists of two aspects: (1) An unlearning benchmark metric should better separate forget and retain set data points (D1), and (2) the metric value should depend on the fraction of the forget set present (D2). These are demonstrated in Fig. 1 and Fig. 3 as well. We will re-emphasize these gaps of current benchmarks in our revised paper.
>
> ---
>
> &#8595; &#8595; &#8595; **Continued below** &#8595; &#8595; &#8595;

---

> ### Author Response · Authors · 2025-11-26
> **Response to Reviewer mg5c (part 2/2)**
>
> > [W3] From what I gather, there needs to be a way to elicit whether an unlearned model forgets or retains data points. Watermarking could be a nice way to trace that, however, it seems that watermarking and detecting those watermarks would only focus on membership information. It does not seem to me to necessarily capture the higher level concepts within the forget set that is actually intended to be forgotten.
>
> Thank you for this question! We agree that WaterDrum is designed to primarily focus on membership information -- whether specific datasets have been unlearned.
>
> We would like to first point out that WaterDrum focuses on the setting of unlearning for data owners (also known as "data influence removal" in [1]) rather than concept unlearning (also known as "knowledge unlearning" in [2, 3]). This distinction is important when the forget data might contain concepts that are also present in the retain data, such as in our exact and semantic duplicate experimental settings. In Sec. 2, we described our setting where even if one copy of the forget data is to be unlearned, the concepts should remain in the model when other copies with similar concepts exists in the retain data. This is in contrast to knowledge unlearning where concepts from the forget set are to be removed no matter whether they are also present in the retain data. In this latter case, other metrics that directly assess the model's ability to answer those concepts would better measure the success of unlearning, e.g., asking the model about those concepts.
>
> - To illustrate the difference, consider the semantic duplicates setting where datasets $D_1$ and $D_2$ only contain concept $C$. In concept unlearning, the goal is to unlearn concept $C$, implying that $D_1$ and $D_2$ should both be forgotten. For data unlearning, if the goal is to unlearn $D_1$, $D_2$ may still be in the retain set and hence the model may still keep knowledge of concept $C$.
> - As we described in Appendix A.2, we would like to emphasize that **WaterDrum is not meant to replace all other metrics**, but can be used in complement with other types of existing LLM unlearning evaluation metrics. We acknowledge that there are many settings that require different types and levels of unlearning, and we consider it as a meaningful future direction to measure other aspects of unlearning.
> - Notably, if the goal is to measure concept unlearning, WaterDrum can act as a rapid and informative first-pass evaluation to verify the removal of data influence. This can subsequently be supplemented by other more computationally intensive metrics that measure other aspects of unlearning, such as high-level concepts.
>
> However, we would also like to highlight that WaterDrum's indication of a dataset's removal does include the removal of the concepts associated with that dataset when no other datasets contains similar concepts. For example, if there is only dataset $D_1$ with concept $C$, and dataset $D_1$ is unlearned, then $C$ could also be considered to have been unlearned.
> - We explicitly validate this with a proxy evaluation experiment to check concept removal. Specifically, in the setting of no duplicate data such that each dataset has its own content, we check whether the WaterDrum metric indicating that dataset $D_i$ is unlearned implies that the concepts in $D_i$ are also unlearned, by asking the model paraphrased questions based on $D_i$ and evaluating the aggregate metric score of the text outputs. Although the experiment setting may contain noise (e.g, bad paraphrased questions), the resulting AUROC is $0.8664$, indicating that *WaterDrum can also identify concept removal* under the no duplicate setting.
>
>  **References**
>
> [1] Liu, Sijia, et al. (2025). "Rethinking machine unlearning for large language models." Nature Machine Intelligence: 1-14.
>
> [2] Si, Nianwen, et al. (2023). "Knowledge Unlearning for LLMs: Tasks, Methods, and Challenges." CoRR.
>
> [3] Jang, Joel, et al. (2023). "Knowledge unlearning for mitigating privacy risks in language models." In Proc. ACL.
>
> ---
>
>
> We hope that our clarifications and additional results will help improve your evaluation of our paper. Thank you very much once again for carefully reviewing our work and providing insightful comments.

---

> > ### Comment · Reviewer_mg5c · 2025-11-27
> > **Great work!**
> >
> > Thank you for your rebuttal, I know writing rebuttals is a lot of work. Sorry, I maintain my score, but I’ll reduce my confidence.
> >
> > On the weaknesses I raised on the writing, I think it was addressed and the authors have a fair point that these formulations may appeal to different audiences. One of the reviewers for instance, explicitly mentioned that the formulations were interesting to think about.
> >
> > For the area chair, I want to reiterate that I don’t mind if this paper is accepted, because the focus of the paper is important. I defer to other reviewers on whether this paper is technically sound / interesting enough to be accepted.

---

### Official Review · Reviewer_mahr · 2025-10-31

**Soundness:** 2
**Presentation:** 3
**Contribution:** 3
**Rating:** 6
**Confidence:** 3

**Summary:**

The paper proposes a framework to evaluate how well a dataset has been unlearned from a LLM. The paper focuses on a setting where one has only a black-box  access to the proposed "unlearned" LLM, and no access to a baseline retrained LLM. As such, I consider this to be first and foremost a paper about auditing black box model.

The authors formulate four main properties their auditing system:
1. separability, i.e, the metric is able to beat a random guesser on whether an llm answer was influenced by data belonging to the unlearned set or not)
2. calibration, to take into account imperfect unlearning in practice
3. practicality, i.e non-reliance on a retrained model
4. robustness to similar data between the unlearned dataset and the retained dataset

From these desired properties, the authors conclude that the use of a watermarking system is well-suited to design a metric fulfilling these properties.

They further go on to specify the desired properties of a watermarking scheme useful for the auditing task -- which I won't describe since they are pretty straightforward properties of most watermarking systems contrary to what the authors claim.

Finally, the auditing system is described as follow: content from each owner is watermarked using a specific key, unique to the owner but fixed across datapoints of this owner. The verification system then detects the presence of the watermark from a the proposed LLM output using a query formed using data points belonging in the unlearned dataset.

Finally, the authors instantiate their auditing system using the WaterFall watermarking scheme and validate the performance of their method against other metric such as ROUGE, Truth Ratio, KnowMem and some membership inference attack.

I also note that the authors propose a new dataset for their benchmark based on the collection of arXiv abstract.

**Strengths:**

**Clarity**: This might be the greatest strength of the paper. Every desired property is extremely well-defined, with a clear formalism and pedagogical explanations. The idea is ell motivated and simple in its implementation. Similarly, the experimental metrics look well chosen and motivated to my non-expert view (with one exception that I will touch in the questions). Consequently, the experiments were convincing to me.

**Originality**: I am not qualified to judge on this point as I am not an expert on unlearning evaluation. However, the use of watermarking for tracking the use of data across LLM training sets is definitely not new, for example in the landmark paper [1] (strangely not being cited !).

**Significance**: I admit to be impressed by the elegant solution this paper propose to black-box auditing. The idea of **not** relying on data for the auditing part allows some very impressive results and flexibility not available without watermarking and I such I find this work quite significant when watermarking is actually available in practice.

**Weaknesses:**

**Some claims are not substantiated**: To be precise, the necessity of watermarking for unlearning metrics, as claimed in Remark 1, is not obvious to me from the desiderata. I would be hard pressed to find a better solution fulfilling both the robustness and separability aspects, but the author do not provide a proof either. Since the onus is on them, I would either retract from such claims or provide actual proof.

**Over-reliance on a specific, watermarking scheme**: Another important unsubstantiated claim is the uniqueness of the WaterFall algorithm in meeting the desiderata. The paper feels heavily skewed towards promoting this specific framework, which appears to be a very recent paper from 2024, possibly from the same research group.  I found it quite surprising that from the classical text watermarking schemes such as [2,3,4] only KGW was (quickly) considered in the appendix (I). I don't understand the reason for disregarding them, especially since the "radioactivity" [1] demonstrated that one can use them and recover a watermark signal in text generated by LLM trained on such schemes. I looked at the Waterfall paper and could not find such a discussion the main body either.  To be precise, the paper dismisses several schemes in the appendix by claiming they fail W5, but this seems like a post-hoc justification to select Waterfall. It's not clear why other schemes couldn't be adapted.

**Lacking references**: This is somewhat of a follow up to the previous weakness. I cannot assess this for the unlearning part, but the choice of references for watermarking is somewhat bizarre. Kirchenbauer is incorrectly referred to as "model watermarking": it is not post-hoc indeed, but does not watermark the model. It watermarks the generated text by modifying the sampling distribution. Aaronson [3] is not cited, despite being the first distortion-free scheme. The authors seems to have focused on post-hoc schemes, but I don't understand why. Furthermore, I don't see any reference to the use of watermark as a tracking technology for training data despite -- e.g. [1] using it exactly for this.  The paper feels, at times, more like an application paper for Waterfall than a comprehensive exploration of watermarking for unlearning evaluation.

**Questions:**

**Questions**:
- Most importantly, I would advise the author to refrain from promoting Waterfall this much without providing better arguments as to why it is necessary or unique within other text watermarking in achieving the desiderata. Either the author should provide a better discussion/experimental proofs that other schemes do not meet the desiderata or tone down the claims towards Waterfall.

- I don't understand the possibility of forgetting data when **exact** duplicates can be found between the retain and forged set. Maybe I am misunderstanding something in the experimental design. I understand the paraphrased data will be different thanks to the use of different keys. But since the **original** data is still the same, what is the point ? Is it only for illustration purposes of the failure of classic unlearning algorithms. I would appreciate the authors make this point clearer in the paper.

- I feel that the paper lacks a calibration study for other watermarking scheme such as KGW and Aaronson. I don't ask for a whole benchmark, but given that these schemes have been shown to contaminate output data for LLM in [1], I believe such a study would at least make the choice of Waterfall a bit more meaningful.

**Recommendation**: I am somewhat torn about this paper. On one hand, I really like the idea of leveraging watermarking capabilities to allow black-box auditing. Furthermore, the paper is very meticulous in describing what it wants to achieve and I found it to be a joy to read in that regard. On the the other hand, the lack of engagement with rest of the field and the huge promotion of Waterfall makes me somewhat suspicious. I am ready to give a *borderline accept* thanks to the attractive idea but would gladly increase my score if the authors can engage with the rest of the art and **really** demonstrate both theoretically and empirically the superiority of Waterfall compared to other schemes. Note that, although I can be considered an expert on **watermarking**, I am definitely not an expert in **unlearning** and I might be missing some insights in the current literature on the well-foundedness of the approach.

**References**:

- [1] Sander, Tom, et al. "Watermarking makes language models radioactive." Advances in Neural Information Processing Systems 37 (2024): 21079-21113.
- [2] Kirchenbauer, John, et al. "A watermark for large language models." International Conference on Machine Learning. PMLR, 2023.
- [3] Aaronson, S. My AI Safety Lecture for UT Effective Altruism, November 2022. URL https://scottaaronson.blog/?p=6823.
- [4] Dathathri, Sumanth, et al. "Scalable watermarking for identifying large language model outputs." Nature 634.8035 (2024): 818-823.

---

> ### Author Response · Authors · 2025-11-26
> **Response to Reviewer mahr (part 1/5)**
>
> Thank you for your helpful review of our work and recognizing its clarity and significance! We will address your comments below.
>
> ---
>
> > [W2] Over-reliance on a specific, watermarking scheme
>
> Regarding the possible over-reliance on a specific watermarking scheme, we thank you for this comment and would like to first clarify that we did not intend to claim the "uniqueness of the Waterfall framework in meeting the desiderata" nor promote this specific framework:
>
> - As this work is primarily on proposing the new unlearning metric called WaterDrum, we focused our experiments on benchmarking with other unlearning metrics rather than ablating over the best watermarking method to be used in WaterDrum. Hence, we conducted our experiments on the watermarking method that directly met our requirements and was easily implemented to suit our purpose without needing to adapt much.
> - That also explained why we explicitly laid out clear watermarking desiderata required for WaterDrum, in the hope of helping spur future work to propose other better suited watermarking methods to be used in WaterDrum. We had highlighted this hope in bold in App A.3, line 757.
>
> We sincerely thank the reviewer for pointing out the impression of just a single watermarking method being promoted in the paper, and agree with the reviewer that including more discussions and experiments would help resolve this issue -- we will add them into the revised paper.
> - We will also explicitly state in the revised paper that while we demonstrated the effectiveness and practicality of our Waterdrum unlearning metric using WaterFall in this paper, any other watermarking schemes satisfying our desiderata can be adopted as well. In fact, we hope that better watermarking methods can be used so that WaterDrum can improve its performance even further, beyond what has already been demonstrated with Waterfall that has outperformed existing metrics.
>
> We provide further clarifications, including details on the discussions and experiments for the various additional watermarking schemes, below.
>
> ---
>
> &#8595; &#8595; &#8595; **Continued below** &#8595; &#8595; &#8595;

---

> ### Author Response · Authors · 2025-11-26
> **Response to Reviewer mahr (part 2/5)**
>
> > [W3] Lacking references
> > ... incorrectly referred to as "model watermarking".
> > ... focused on post-hoc schemes...
> > ... comprehensive exploration of watermarking for unlearning evaluation.
>
> We thank the reviewer for highlighting the various watermarking references [1,2,3,4] -- we will discuss all of them in our revised paper, along with [5] which describes in its Sec. 2.5 the watermarking method from [3].
>
> We would like to clarify that our paper adopts a different focus and classification of watermarking works. We summarize our (non-exhaustive) classification in the list below:
>
> 1. "Model watermarking" (Single model owner protects its LLM's generated text)
>
>     1a. Token-wise during autoregressive generation, e.g., [2,3,4,5]
>
>     1b. Model weights, e.g., [6]
>
>     1c. Post hoc (using text watermarking right after generation), e.g., [7]
>
> 2. "Text watermarking" (Multiple data owners each protect their existing text)
>
>     2a. LLM aided paraphrasing, e.g., [8]
>
>     2b. Word/Synonym replacement, e.g., [9]
>
>     2c. Invisible Unicode, e.g., [10]
>
> The reviewer may be thinking of watermarking from the classical model owner's perspective (1) with methods such as watermarking the generated text by changing the sampling distribution (1a), watermarking the model's weights (1b), and post-hoc watermarking the LLM-generated text (1c). We refer to all of these approaches as "model watermarking". However, we focus on  watermarking from the data owners' perspective (2) where different data owners want to independently verify if their data still has influence on the LLM. This involves watermarking the training data via LLM approaches (2a) or non-LLM approaches (2b, 2c, mentioned in App. A). WaterFall falls under 2a and relies on using LLM paraphrasing to generate the watermarked training data.
>
> - Our definitions of "model watermarking" and "text watermarking" were presented in App. D.3. We defined "model watermarking" (1) as techniques that a model owner implements to **inject a watermark into the LLM’s text outputs and tokens** typically used by a single model owner to distinguish whether a piece of text is generated by their model. This includes those that perform watermarking during the LLM's autoregressive generation process (1a) as in [2,3,4], which is referred to as "LLM watermarking" in [1].
> - In contrast, we define "text watermarking" (2) as techniques that **data owners use to watermark their existing text**, such as human-written articles, in order to trace downstream usage of their data, such as in model training. Unlike "model watermarking" (1), the primary concerns in setting (2) differ, such as stricter preservation of semantic content of the text (W0 Fidelity) and scalablility to support a large number of data owners concurrently.
>
>
> In our revised paper, we can distinguish them as "watermarking for a model owner" vs. "watermarking for multiple data owners" instead. We will also further clarify and contrast the differences between these two modes of watermarking, in terms of who uses them (i.e., model owner vs. data owner), input (i.e., generic LLM query vs. existing source text), and scalability (i.e., supporting single model owner vs. multiple data owners). We thank the reviewer for pointing out [1] which will help contextualize the new discussions and we will add for the other suggested watermarking references.
>
> **References**
>
> [1] Sander, Tom, et al. (2024). "Watermarking makes language models radioactive." In Proc. NeurIPS, pages 21079-21113.
>
> [2] Kirchenbauer, John, et al. (2023). "A watermark for large language models." In Proc. ICML.
>
> [3] Aaronson, S. (2022). My AI Safety Lecture for UT Effective Altruism, URL https://scottaaronson.blog/?p=6823.
>
> [4] Dathathri, Sumanth, et al. (2024). "Scalable watermarking for identifying large language model outputs." Nature 634.8035: 818-823.
>
> [5] Kuditipudi, Rohith, et al. (2024). "Robust Distortion-free Watermarks for Language Models." TMLR.
>
> [6] Li, Linyang, et al. (2023). Watermarking LLMs with Weight Quantization. In Proc. EMNLP.
>
> [7] Hao, Jifei, et al. (2025). Post-Hoc Watermarking for Robust Detection in Text Generated by Large Language Models. In Proc. COLING.
>
> [8] Lau, Gregory Kang Ruey, et al. (2024). "Waterfall: Scalable Framework For Robust Text Watermarking and Provenance for LLMs." In Proc. EMNLP.
>
> [9] Qiang, Jipeng, et al. (2023). Natural language watermarking via paraphraser-based lexical substitution. Artificial Intelligence, 317, 103859.
>
> [10] Sato, Ryoma, et al. (2023) Embarrassingly simple text watermarks. In Proc. CoRR.
>
> ---
>
> &#8595; &#8595; &#8595; **Continued below** &#8595; &#8595; &#8595;

---

> ### Author Response · Authors · 2025-11-26
> **Response to Reviewer mahr (part 3/5)**
>
> > [Q1] Either the author should provide a better discussion/experimental proofs that other schemes do not meet the desiderata or tone down the claims towards Waterfall.
>
> > [Q3] ...calibration study for other watermarking scheme such as KGW and Aaronson ... I believe such a study would at least make the choice of Waterfall a bit more meaningful.
>
> In addition to adjusting the phrasing in our paper to avoid the impression of Waterfall being the only watermarking method that can be used in WaterDrum, we will provide both discussions and experiments on how various other watermarking methods meet/do not meet the various desiderata.
>
> ### **Watermarking desiderata**
>
> We summarize the differences in the original watermarking schemes without adaptations in the table below. In place of [3], we used [5], a published distortion-free "watermarking for a model owner" paper where the approach of watermarking via exponential minimum sampling (EXP) is from [3].
> - As mentioned above, we adopted Waterfall as our watermarking method because it explicitly focused on the watermarking of training text data directly, unlike [2,4,5] that focused their watermarking design and experiments specifically to the setting of watermarking LLM-generated text during LLM inference.
> - Based on just the designs and experimental support provided in references [2,4,5], it was not clear that all of the watermarking desiderata would be met. The work of [1] pointed out by the reviewer would help provide some support, though we still needed further adaptations to [2,4,5] and additional experiments to be run to establish that the watermarking desiderata were met:
>     - For example, to satisfy our **W4 Unique key**, the watermarking schemes that inject watermarks into training texts need to support insertion and verification of watermarks from *different/multiple* data owners. In contrast, schemes [2,4,5] designed for watermarking a single LLM's generated text typically only requires a *single* watermark to serve the single model owner.
>     - The requirements of **W2 Overlap verifiability** were also not explored in [1], but is an important requirement for watermarking schemes to be used in WaterDrum. On the other hand, Sec. 4.3 of Waterfall [8] had established this property of being able to recover *multiple different* watermark signals in text generated by LLMs trained on text watermarked by *many data owners* in their experiments, giving us more confidence in using it for demonstrating the effectiveness of our WaterDrum unlearning metric.
>
> ||W0|W1|W2|W3|W4|
> |-|-|-|-|-|-|
> |Waterfall [8]|Yes|Yes|Yes|Yes|Yes|
> |KGW [2]|No^|Yes*|Not shown|Yes|No^|
> |Synth-ID [4]|No^|No*|Not shown|Yes|No^|
> |EXP-edit [5]|No^|No*|Not shown|Yes|Yes|
>
> ^ No off-the-shelf, requires modification to their code to support the desideratum.
>
> \* Based on our results shown below after adaptation of the schemes.
>
> ---
>
> &#8595; &#8595; &#8595; **Continued below** &#8595; &#8595; &#8595;

---

> ### Author Response · Authors · 2025-11-26
> **Response to Reviewer mahr (part 4/5)**
>
> **Contunuing previous response**
>
> ---
>
> ### **Experiments with different watermarking schemes**
>
> However, although some watermarking schemes were not originally designed to satisfy the watermarking desiderata directly, we agree that it may be possible to try adapting the schemes to satisfy the desiderata. To run our experiments below, we have tried adapting the schemes [2,4,5] to meet the watermarking desiderata, though we believe that future work might be able to nontrivially improve the design of the different watermarking schemes and satisfy the desiderata to a better extent than our implementations below and than WaterFall:
> - We adapted [2,4,5] and ran additional experiments by adding a paraphrasing prompt following the approach in WaterFall [8]. As [2,4] were designed for "watermarking (of LLM-generated text) for a model owner" (as discussed above), their codebase had explicitly fixed single watermark IDs into the internal components of the source code -- we had to modify it for them to accept different watermarks and support multiple data owners.
> - We used the same "Llama-3.1-8B-Instruct" LLM to apply text watermarking on the ArXiv dataset, except for Synth-ID for which we used "Gemma-7b-it" as the provided codebase only supports Gemma and GPT-2 models.
>
> The table below shows results of the adapted [2,4,5] and Waterfall in D1 Separability and D2 Calibration desiderata, along with the average time required to verify a single text sample and whether a GPU is required for verification.
> - It is promising to see that the adapted KGW performs very well in D2 Calibration with an $R^2$ value of almost $1$, though it has worse performance than Waterfall in D1 Separability and also requires significantly more verification time and GPU resources. This can possibly be attributed to D1 Separability requiring the W2 Overlap verifiability desideratum, which KGW may not be designed to achieve due to its "watermarking (of LLM-generated text) for a model owner". In contrast, Waterfall was designed for the "watermarking (of training text) for multiple data owners" and hence had different watermark ID signals designed to be orthogonal to each other, allowing for greater distinction among IDs and hence satisfying W2 and D1, respectively. Its design may also have resulted in its slightly lower $R^2$ value in the D2 Calibration desideratum than adapted KGW (i.e., $0.963$ vs. $0.996$).
>     - Hence, in the setting where there is only $1$ data owner/forget set with the rest of the retain set being public data, satisfying D2 is more important than satisfying D1, and it is practical to use significantly more verification time and GPU resources, it may be better to use the adapted KGW as the watermarking method in WaterDrum since the W2 desideratum may be relaxed in this case. However, for general settings, it may still be better to use Waterfall given its better performance in D1 and D2.
> - Unfortunately, [4,5] did not perform well, likely because they were not designed for satisfying W1 (even though as the reviewer pointed out, based on [1], it might turn out to be empirically, like for the adapted KGW) and W2.
> - In addition, the adapted methods [2,4,5] either require a GPU or a few orders of magnitude longer time than Waterfall to verify whether a single text sample contains the watermark. This places practical constraints on when these methods can be used in WaterDrum.
>
> ||D1 Separability|D2 Calibration|Verification time|Requires GPU|
> |-|-|-|-|-|
> |Waterfall [8]|0.965|0.963|0.015s|No|
> |adapted KGW [2]|0.8710|0.996|0.336s|Yes|
> |adapted Synth-ID [4]|0.5488|-16.951|0.386s|Yes|
> |adapted EXP-edit [5]|0.7891|-17.079|165.5s|No|
>
>
> We thank the reviewer once again for suggesting the discussions and experiments. While our paper is primarily on proposing an unlearning metric, we agree with the reviewer that providing such results and discussions will enrich the paper and potentially further inspire future work on how better watermarking methods can be used to further improve WaterDrum, which is what we sincerely hope for. We will include these results in our revised paper.
>
>
> ---
>
> > [W1] The necessity of watermarking for unlearning metrics, as claimed in Remark 1, is not obvious to me from the desiderata.
>
> Thank you for pointing this out. We had only intended to highlight that among existing unlearning metrics, the watermarking approach (i.e., WaterDrum) was the only one that met all desiderata, and "using watermarked data" was necessary to achieve that. We will adjust the language to avoid the impression of asserting that we have proven the necessity of watermarking to meet the desiderata:
> - Remark 1. Using watermarked data is both (i) beneficial and important for identifying practical and effective unlearning metrics...
>
> ---
>
> &#8595; &#8595; &#8595; **Continued below** &#8595; &#8595; &#8595;

---

> ### Author Response · Authors · 2025-11-26
> **Response to Reviewer mahr (part 5/5)**
>
> > [Q2] I don't understand the possibility of forgetting data when exact duplicates can be found between the retain and forget set.  But since the original data is still the same, what is the point? Is it only for illustration purposes of the failure of classic unlearning algorithms. I would appreciate the authors make this point clearer in the paper.
>
> In real-world settings, the data owners supplying data for LLM training can be data brokers who themselves source data from the creators of the content. The same piece of data might be collated by multiple different data brokers, and these data brokers can later retract or request for their data to be unlearned from the LLM.
>
> We do however agree with the reviewer that the 'exact duplicate' setting can be less common. We introduce the 'exact duplicate' setting to clearly illustrate and clarify the limitations of existing unlearning metrics, whereas our WaterDrum performs well even in this extreme setting.
> - In practice, it is more likely that our 'semantic duplicate' setting happens and is **a better measure of realistic unlearning capabilities** (with real-world examples discussed in Appendix B.2). We will clarify the purpose of the 'exact duplicate' setting clearly in our revision.
> - The 'exact duplicate' setting in reality might involve data owners possibly with subsets of duplicate data. In those cases, each data owner may not be aware of such duplicate data from other data owners and would still appreciate indicators of whether all their data has been removed.
>
>
> ---
>
> Thank you again for reviewing our work and providing valuable feedback. We sincerely hope that our clarifications and additional comparisons with other watermarking baselines will improve your evaluation of our paper. We would be happy to provide further clarification.

---

> > ### Comment · Reviewer_mahr · 2025-11-27
> >
> > I sincerely thank and command the authors for the comprehensive rebuttal. I will most certainly increase my score. Given all the work that has been performed to address my gripes, I believe the paper to be quite deserving -- please remember to update the revision if it hasn't been done already. I won't change my score otherwise.
> >
> > I would like to highlight the fact that you took the time to adapt other watermarking algorithms for your solution, which is no small feat in such a small amount of time.
> >
> > I have a few remaining questions:
> >
> > > The reviewer may be thinking of watermarking from the classical model owner's perspective (1) with methods such as watermarking the generated text by changing the sampling distribution (1a), watermarking the model's weights (1b), and post-hoc watermarking the LLM-generated text (1c). We refer to all of these approaches as "model watermarking". However, we focus on watermarking from the data owners' perspective (2) where different data owners want to independently verify if their data still has influence on the LLM. This involves watermarking the training data via LLM approaches (2a) or non-LLM approaches (2b, 2c, mentioned in App. A). WaterFall falls under 2a and relies on using LLM paraphrasing to generate the watermarked training data.
> >
> > Yes, this exactly the distinction over which I have a difficulty. Since (2) comprises (post-hoc) paraphrasing by an LLM, methods from (1a) can definitely be used for watermarking itself, no ? Isn't it exactly what you did for the new experiments ? In this case, isn't the only difference how the keys/id are managed ? I am sorry if I am missing something here.
> >
> > > [Q2] ...
> >
> > Your answer is clear and mostly what I understood when reading the paper. My actual question was, how relevant is that setting. If you only unlearned my watermarked data, but that the underlying, non watermarked data is an exact duplicate between brokers, did I really unlearn anything ? I somewhat see this as a main difference between a data-centric approach and a data-agnostic approach such as yours, where the unlearning is done on specific *watermarked* data-points, but not the original one.

---

> > > ### Author Response · Authors · 2025-12-03
> > > **Further Response to Reviewer mahr (part 1/2)**
> > >
> > > Thank you very much for your acknowledgement of our rebuttal and proposing to raise your score. We have updated the revised paper accordingly, as you’ve requested. We will address your further comments below.
> > >
> > > ---
> > >
> > >
> > > > Yes, this exactly the distinction over which I have a difficulty. Since (2) comprises (post-hoc) paraphrasing by an LLM, methods from (1a) can definitely be used for watermarking itself, no ? Isn't it exactly what you did for the new experiments ? In this case, isn't the only difference how the keys/id are managed ? I am sorry if I am missing something here.
> > >
> > >
> > > Yes, you are right that making (1a) able to accept different watermarking keys/ids is a change that we had to make to apply them for (2). The major distinction between (1a) and (2a) is that the watermarking methods designed for (1a) focus on the application scenario with **a single model owner** while those for (2a) need to **support a large number of data owners**. This is exactly the reason why methods like [2,4] had a fixed single watermark ID hard-coded into their source code. Therefore, methods from (1a) can be used for watermarking with the following steps and caveats:
> > >
> > > - Step 1: The implementation has to be modified to be **verifiable in the LLM even when data of multiple watermarks are present in the training data**, such that it supports a large number of data owners. As [2,4,5] are not explicitly designed with this in mind, methods from (1a) can have limited ability in satisfying the W2 Overlap verifiability desideratum, which can help explain their poorer performance in D1 Separability.
> > > - Step 2: The implementation needs to **include a paraphrasing prompt along with the source text as the input to the LLM** when watermarking. Although a seemingly trivial change, the paraphrasing prompt plays a major role in the design of (2a), due to the key difference in the measure of "text quality" between (1) and (2). In (1), quality is evaluated as how close the watermarked text matches an LLM's original predictive probabilities given an arbitrary prompt, often measured with perplexity as in [2,4]. On the other hand, (2) requires W0 Fidelity, where the watermarked text has to preserve the semantics of the original source text, often measured with a semantic text similarity score. Simply using (1a) off the shelf without the paraphrasing prompt will result in the LLM generating new text which would likely not retain W0 Fidelity.
> > > - Caveat: As the methods in (1a) are designed with a single model owner in mind, there is no guarantee on the behavior with multiple data owners such as W2 Overlap verifiability. After the adaptation steps, we have to verify them empirically which we have done in the previous response. In contrast, WaterFall was designed to support multiple data owners and hence it was our initial choice.
> > > - Beyond these steps we implemented to adapt (1a) into (2a), there are a few other design choices in Waterfall that make it better suited for (2). For example, Waterfall's implementation includes a weighted scoring mechanism to balance W0 Fidelity against the W1 Verifiability of the watermark. These mechanisms can also be incorporated into the other (1a) schemes when they are adapted for (2a) to be used in Waterdrum.
> > >
> > > ---
> > >
> > > &#8595; &#8595; &#8595; **Continued below** &#8595; &#8595; &#8595;

---

> > > ### Author Response · Authors · 2025-12-03
> > > **Further Response to Reviewer mahr (part 2/2)**
> > >
> > > > [Q2] ...
> > >
> > > We would first like to clarify that in our watermarking setting described in P1 Watermarking setup in Sec. 4, data owners have already watermarked their original data before providing them to the model owner for LLM training. The **LLM is never trained on these original unwatermarked data** and is trained on the paraphrased, watermarked data instead, and unlearning requests from the data owners only involve removing the watermarked data instead of the original one that was never seen by the model owner. Therefore, WaterDrum can **verify the unlearning of the data used for training by verifying the removal of the watermarked signal**.
> > >
> > > > My actual question was, how relevant is that setting. If you only unlearned my watermarked data, but that the underlying, non watermarked data is an exact duplicate between brokers, did I really unlearn anything?
> > >
> > > We interpret your question as: why is WaterDrum unlearning metric still meaningful when multiple data owners have the same/similar unwatermarked data but different watermarked data (i.e., in the real-world setting described in Appendix B.4)? We agree with you that under the exact duplicate setting, when only some data owners request for their data to be unlearned, the semantic content (e.g., of a news article) may still be present in the LLM. However, we aim to address another problem in this setting: **Data owners still want a metric to verify that their specific copy of data has been unlearned** in accordance with privacy, copyright, and retraction laws, regardless of the semantic content or whether the other data owners' copies are in the retain set. WaterDrum is **meaningful** as it is robust to meaningful similar data, allowing individual owners of similar data to verify if their data has been unlearned. As we discussed in the last paragraph of Sec. 2, existing utility-centric metrics (e.g., truth ratio) may not be able to do so.
> > >
> > > We would also like to emphasize our discussion in App. A.2, that WaterDrum is not meant to replace all other metrics, but can be **used in complement** with other types of existing utility-centric LLM unlearning evaluation metrics that measure other aspects of unlearning (e.g., truth ratio can measure general semantic content).
> > >
> > > > I somewhat see this as a main difference between a data-centric approach and a data-agnostic approach such as yours, where the unlearning is done on specific watermarked data-points, but not the original one.
> > >
> > > We like to clarify that WaterDrum is by no means data-agnostic. We described in Sec. 4 that WaterDrum is a data-centric metric that directly tracks the influence of data by actively embedding data-specific signals (unique to the data owner) detectable in the LLM’s text outputs. Fig. 2 explains that WaterDrum is robust to similar data by embedding orthogonal data-specific signals in the LLM’s text outputs that are W1 verifiable. Unlike data-agnostic metrics that focus on global model performance and might not be able to verify the unlearning of specific data, as a data-centric metric, WaterDrum can **verify if the watermarked data from a certain data owner is unlearned**.
> > >
> > > ---
> > >
> > > Thank you once again for your acknowledgement of our rebuttal.

---

### Author Response · Authors · 2025-12-03
**Summary of Rebuttal**

Dear ACs,

We appreciate the time, effort, and valuable feedback from the reviewers and ACs. We summarize the discussions with the reviewers:

I. Contributions and Significance
- We focused on an **interesting and important problem** by analyzing the limitations in existing LLM unlearning metrics ([mg5c](#:~:text=raises,benchmarks), [ftSk](#:~:text=idea%20of%20evaluating,insightful), [mahr](#:~:text=idea%20is,motivated)).
- We formulated **clear and extremely well-defined desiderata** for effective and practical LLM unlearning metrics that challenge key limitations of existing utility-centric metrics ([mahr](#:~:text=Every,explanations)).
- We introduced **WaterDrum, a novel and elegant solution** by applying text watermarking to evaluate LLM unlearning ([ftSk](#:~:text=idea%20of%20evaluating,insightful), [mahr](#:~:text=impressed,solution)).
- Extensive experiments **convincingly** validate the effectiveness of WaterDrum ([mahr](#:~:text=experimental%20metrics,motivated&text=experiments%20were%20convincing)).
- We introduced a new benchmark dataset **WaterDrum-Ax which is a useful foundation for future LLM unlearning evaluation research** ([ftSk](#:~:text=new%20benchmark%20WaterDrum,area)).
- The reviewers recognized the **clarity in our writing** ([mahr](#:~:text=Clarity%3A,paper), [mg5c](#:~:text=spent,writing)). While reviewer mg5c was initially concerned about the necessity of our well-defined and clear formalism, reviewer mahr [praised it as `the greatest strength`](#:~:text=greatest%20strength%20of,.), to which reviewer mg5c [agreed that it appeals to different audiences](#:~:text=it%20was%20addressed,audiences).

II. Key Reviewer Concerns/Questions Addressed

**Reviewer mahr:**
- We [clarified our definitions on "model" vs "text" watermarking](#:~:text=our%20paper%20adopts,below). To address the reviewer's confusion, we changed the terminologies in our paper to watermarking for "a model owner" and "multiple data owners" respectively.
- The reviewer [*acknowledged and highlighted* our efforts in performing additional experiments using watermarking methods](#:~:text=highlight%20the,algorithms) that were not initially designed to satisfy our problem setting. [By modifying them to perform "watermarking for multiple data owners", some of these methods could be viable as components within WaterDrum](#:~:text=Experiments%20with,schemes). We also 1) demonstrated the [superiority of the chosen watermarking method compared to other existing methods](#:~:text=table%20below%20shows,Waterfall) and 2) clarified that WaterDrum can [support new watermarking methods proposed in the future](#:~:text=future%20work%20might,desiderata).
- We further [explained the motivation for the 'exact duplicate' setting](#:~:text=In%20real%2Dworld%20settings,the%20LLM) and how [our metric is more fine-grained than the utility-centric metrics under this setting](#:~:text=WaterDrum%20is%20meaningful,so).

**Reviewer mg5c:**
- The reviewer was [satisfied with our response](#:~:text=it%20was%20addressed,point) to their concern regarding the presentation of the paper, where we stated that we [cater to readers of different backgrounds by including both formal, unambiguous notations as well as intuitive illustrations in our paper](#:~:text=presentation%20of%20the%20paper%20is,preferences).
- We [clarified the distinctions between the setting of unlearning for data owners and concept unlearning](#:~:text=WaterDrum%20focuses,3%5D). While our focus is on the former, we showed that [WaterDrum could also measure the latter under the 'no duplicate' setting](#:~:text=WaterDrum%27s,similar%20concepts).

**Reviewer ftSk:**
- We pointed the reviewer to the appendix addressing their question, where the [practical concern of WaterDrum requiring data to be watermarked before training should diminish over time](#:~:text=As%20discussed%20in,time).
- As suggested, we performed additional experiments of 1) [implementing WaterDrum with other adapted watermarking methods](#:~:text=We%20adapted%20%5B3%2C4%2C5%5D,experiments), and 2) [evaluating WaterDrum on a larger LLM](#:~:text=we%20ran,model).
- We [explained that WaterDrum is applicable to the pre-training stage without adjustments](#:~:text=WaterDrum%20can%20be,stage) to answer the reviewer's question.

Concluding our discussions, reviewer mahr indicated [satisfaction with our responses and has proposed to raise the score](#:~:text=thank%20and,score); reviewer mg5c reduced the confidence but [`wouldn't mind if this paper is accepted` due to the significance of our work](#:~:text=I%20don%E2%80%99t%20mind,important); reviewer ftSk did not indicate further questions.

In our revised paper (edits in blue), we added experiments of adapting watermarking methods for model owner into WaterDrum in Sec. 5.2, App. E.7, evaluation on larger LLMs in App. F.3, and more related works on concept unlearning and text watermarking in App. A.

---

We thank the ACs for evaluating our submission.

Regards,

The Authors.

---

### Meta-Review · Area_Chair_j9oa · 2026-01-07

**Summary:**

The paper presents WaterDrum, a data-centric unlearning metric for LLMs that uses text watermarking to verify whether training data has been unlearned. The reviewers raised several concerns. Reviewer mahr questioned the heavy reliance on the Waterfall watermarking scheme and wanted to see comparisons with alternatives like KGW and Aaronson's method. Reviewers mg5c and ftSk both found the methodology section overly dense with notation, which made the experimental contributions feel less substantial. Reviewer ftSk also pointed out the practical limitation that data must be watermarked before model training. Finally, mg5c raised concerns about whether watermarking can capture higher-level concept unlearning or only membership information. That said, all reviewers found the problem important and the core idea novel, with mahr highlighting the clarity of the desiderata formulation.

**Reviewer Concerns:**

The authors addressed several key concerns effectively during the rebuttal period. The concern about over-reliance on Waterfall was resolved through comprehensive experiments adapting KGW, Synth-ID, and EXP-edit, demonstrating Waterfall's advantages in separability and verification speed. The terminology confusion between "model watermarking" and "text watermarking" was clarified and the paper was updated accordingly. Scalability concerns were addressed through new Llama2-13B experiments showing promising results. The authors also clarified that WaterDrum focuses on data influence removal rather than concept unlearning, making it complementary to existing metrics.

Some concerns remain outstanding. The presentation density issue was maintained by mg5c but the reviewer signalled that they do not oppose accepting the paper. The requirement to watermark before training remains an inherent limitation of the approach but was addressed with reasonable arguments about future adoption trends by the authors.

**Reviewer Scores:**

Reviewer mahr started at 6 and would likely increase to 8, having explicitly stated they would "most certainly increase" their score after the comprehensive rebuttal addressing watermarking comparisons. Reviewer mg5c at 4 remains unchanged but does not oppose acceptance of the paper. Reviewer ftSk (score 6) did not respond after the rebuttal, though all their questions were addressed by the authors with additional experiments.

---

### Decision · Program_Chairs · 2026-01-26

Accept (Poster)